# Sample-Adaptivity Tradeoff in On-Demand Sampling

**Nika Haghtalab**
University of California, Berkeley
nika@berkeley.edu

**Omar Montasser**
Yale University
omar.montasser@yale.edu

**Mingda Qiao**
University of Massachusetts Amherst
mqiao@umass.edu

## Abstract

We study the tradeoff between sample complexity and round complexity in *on-demand sampling*, where the learning algorithm adaptively samples from $k$ distributions over a limited number of rounds. In the realizable setting of Multi-Distribution Learning (MDL), we show that the optimal sample complexity of an $r$-round algorithm scales approximately as $dk^{\Theta(1/r)}/\varepsilon$. For the general agnostic case, we present an algorithm that achieves near-optimal sample complexity of $\widetilde{O}((d + k)/\varepsilon^2)$ within $\widetilde{O}(\sqrt{k})$ rounds. Of independent interest, we introduce a new framework, Optimization via On-Demand Sampling (OODS), which abstracts the sample-adaptivity tradeoff and captures most existing MDL algorithms. We establish nearly tight bounds on the round complexity in the OODS setting. The upper bounds directly yield the $\widetilde{O}(\sqrt{k})$-round algorithm for agnostic MDL, while the lower bounds imply that achieving sub-polynomial round complexity would require fundamentally new techniques that bypass the inherent hardness of OODS.

## 1 Introduction

Modern machine learning pipelines increasingly treat the training set as a mutable resource—adapting the data collection process in response to intermediate learning signals in order to focus effort where it matters most. This adaptivity arises in a range of settings. In multi-distribution learning (see e.g. [BHPQ17, HJZ22]), the on-demand sampling framework allows algorithms to adaptively select domains from which to sample to minimize the worst-case loss. Similarly in multi-armed bandit problems, adaptively selecting which arm to pull next is an important aspect of algorithm design. In practice, pipelines for training models adaptively decide how to reweigh or augment their datasets to improve downstream accuracy [XPD+23, SRC24]. While these paradigms demonstrate—both theoretically and empirically—that adaptive data collection can significantly improve performance and sample efficiency, adaptivity is often an undesirable feature. It requires the ML practitioner to collect data sequentially, slowing down the end-to-end training process and limiting opportunities for parallelism and scalability. This tension raises a central question:

> *to what extent is adaptive sample collection necessary to achieve the observed gains in learning performance? And what are the quantitative tradeoffs between the number of adaptive rounds and sample complexity?*

We study this question in the context of *on-demand sampling* within the framework of *multi-distribution learning* [HJZ22]. Multi-distribution learning extends the classical agnostic learning setting by giving the learner sampling access to $k$ distributions $D_1, \ldots, D_k$, with the goal of learning a single predictor that minimizes the worst-case error across all distributions. This framework has

39th Conference on Neural Information Processing Systems (NeurIPS 2025).

emerged as a central model for studying algorithmic dataset selection, offering both a method of allocating a fixed sampling budget across heterogeneous data sources and a unifying perspective on several recent advances in federated learning, multi-task learning, domain adaptation, and fair and robust machine learning [KNRW18, MSS19, SKHL20, RY21, TH22, HJZ23, ZZC+24].

Prior work on multi-distribution learning has established optimal on-demand sample complexities of $\widetilde{O}((d + k)/\varepsilon)$ in the realizable case [BHPQ17, CZZ18, NZ18] and $\widetilde{O}((\log(|\mathcal{H}|) + k)/\varepsilon^2)$ in the agnostic case [HJZ22], where $d$ is the VC dimension of hypothesis class $\mathcal{H}$. The latter was recently extended to $\widetilde{O}((d+k)/\varepsilon^2)$ for infinite hypothesis classes [ZZC+24, Pen24]. However, these algorithms rely on a large number of adaptive rounds—often polynomial in $1/\varepsilon$ and the complexity of $\mathcal{H}$ and with a mild sublinear dependence on $k$ (see Table 1 for details). In some cases, these algorithms collect a single sample per round, resulting in a number of rounds that is as large as the sample complexity itself! In contrast, the best known fully non-adaptive algorithms incur significantly higher sample complexities of $\widetilde{O}(dk/\varepsilon)$ and $\widetilde{O}(dk/\varepsilon^2)$ in the realizable and agnostic settings, respectively.

Despite this gap, the complexity landscape between the two extremes—full adaptivity and full non-adaptivity—remains largely unexplored. In particular, it is unknown whether a *constant number of adaptive rounds*, or even one that is merely *independent of the accuracy level $\varepsilon$*, could suffice to recover the optimal sample complexities achieved in the fully adaptive setting.

**Our Contributions and Results**    In this work, we formalize the problem of studying the tradeoffs between adaptivity and sample complexity of on-demand sampling algorithms. Specifically, we aim for achieving the optimal sample complexity with a number of adaptive rounds that is nearly independent of $\varepsilon$ and $d$ and with only sublinear dependence on $k$. We refer to the number of adaptive rounds as the *round complexity* of an algorithm.

In the realizable case, we provide a tight characterization of the sample-adaptivity tradeoff. In particular, we prove that a round complexity $r$ allows for a sample complexity of $dk^{\Theta(1/r)}/\varepsilon$, yielding a smooth tradeoff between round complexity and sample complexity. In addition to confirming that $\log k$ rounds are necessary for achieving the optimal sample complexity in the realizable setting, this also indicates that a small constant number of rounds—say, 3 rounds (!)—are sufficient to achieve an $\widetilde{O}(d\sqrt{k}/\varepsilon)$ sample complexity, which is a significant improvement over fully non-adaptive approaches. In the agnostic case, we show that $\widetilde{O}(\sqrt{k})$ rounds of adaptivity is sufficient to achieve the optimal sample complexity of $\widetilde{O}((d + k)/\varepsilon^2)$.

From a technical perspective, we establish the tradeoff between adaptivity and sample complexity through two approaches. In the realizable case, our algorithms are based on a novel application of a variant of the AdaBoost algorithm with a particular notion of margin. In the agnostic setting, we introduce a general and abstract optimization problem called Optimization via On-Demand Sampling (OODS). In this framework, the goal is to optimize a concave function $f$ over $[0, 1]^k$, representing weights over $k$ distributions. There is no notion of sample complexity in this setting; instead, the algorithm can only access value and gradient information about $f$ within a restricted *trust region*, which the algorithm can expand in every round. At a high level, the extent of the trust region serves as a proxy for sample complexity: the more a distribution is sampled, the better we can estimate the performance of predictors on it. The number of times the trust region is expanded before finding the optimum of $f$ corresponds to the round complexity. We establish both upper and lower bounds on the round complexity in the OODS setting.

A strength of the OODS framework is that algorithms developed for OODS naturally transfer to the agnostic multi-distribution learning problem, forming the foundation for our performance guarantees. Additionally, the optimization formulation gives rise to more natural algorithm-independent lower bounds on adaptivity. In particular, we prove $\mathrm{poly}(k)$ lower bounds on the round complexity of the OODS problem. These lower bounds shed light on the challenges of achieving the optimal sample complexity in agnostic multi-distribution learning using a sub-polynomial number of rounds.

## 1.1   Related Work

**Multi-Distribution Learning**    Blum, Haghtalab, Procaccia and Qiao [BHPQ17] introduced the *realizable* setting of multi-distribution learning, for which several $O(\log k)$-round algorithms with near-optimal sample complexity of $\widetilde{O}((d+k)/\varepsilon)$ were given [BHPQ17, CZZ18, NZ18]. On the other

| Setting | Sample Complexity | Round Complexity | Reference |
|---|---|---|---|
| Realizable | $\widetilde{O}((d+k)/\varepsilon)$ | $O(\log k)$ | [BHPQ17, CZZ18, NZ18] |
| Agnostic | $\widetilde{O}((\log(|\mathcal{H}|)+k)/\varepsilon^2)$ | $\widetilde{O}((\log(|\mathcal{H}|)+k)/\varepsilon^2)$ | [HJZ22] |
| Agnostic | $\widetilde{O}(d/\varepsilon^4 + k/\varepsilon^2)$ | $O(\log(k)/\varepsilon^2)$ | [AHZ23] |
| Agnostic | $\widetilde{O}((d+k)/\varepsilon^2) \cdot (\log k)^{O(\log(1/\varepsilon))}$ | $(\log k)^{O(\log(1/\varepsilon))}$ | [Pen24] |
| Agnostic | $\widetilde{O}((d+k)/\varepsilon^2)$ | $O(\log(k)/\varepsilon^2)$ | [ZZC$^+$24] |
| Realizable | $\widetilde{O}((k^{2/r} \cdot d + k)/\varepsilon)$ | $r$ | Theorem 1 |
| Realizable | $\widetilde{\Omega}(k^{1/r} \cdot d/r)$ | $r$ | Theorem 2 |
| Agnostic | $\widetilde{O}((d+k)/\varepsilon^2)$ | $\min\{\widetilde{O}(\sqrt{k}), O(k \log k)\}$ | Propositions 2 and 3 |

Table 1: An overview of sample-adaptivity tradeoff in multi-distribution learning. $k$ is the number of distributions. $\mathcal{H}$ is the hypothesis class and $d$ is its VC dimension. $r$ denotes a tunable round complexity between 1 and $O(\log k)$. $\widetilde{O}(\cdot)$ and $\widetilde{\Omega}(\cdot)$ suppress $\mathrm{polylog}(d, k, 1/\varepsilon, 1/\delta)$ factors.

hand, it is folklore that the sample complexity is $\Omega(dk/\varepsilon)$ without adaptive sampling. For the more challenging *agnostic* setting where a "perfect" predictor may not exist, the optimal sample complexity was shown to be $\widetilde{O}((d+k)/\varepsilon^2)$ in a series of recent work [HJZ22, AHZ23, ZZC$^+$24, Pen24]. Interestingly, all these algorithms have a round complexity of at least $\mathrm{poly}(1/\varepsilon)$ (see Table 1 for details). Other variants of the problem, where some data sources might be adversarial or differently labeled, have also been studied [Qia18, DQ24].

**Power of Adaptivity in Learning and Beyond**  Agarwal, Agarwal, Assadi and Khanna [AAAK17] systematically formulated the tradeoff between adaptivity and sample complexity in several learning problems, including a batched setting of multi-armed bandits that was previously introduced by Jun, Jamieson, Nowak and Zhu [JJNZ16] and subsequently studied in [GHRZ19, JYT$^+$24, JZZ25]. Chen, Papadimitriou and Peng [CPP22] proposed a PAC learning framework of continual learning, and quantified the tradeoff between the number of sequential passes and the memory usage of the learning algorithm. Another recent line of work focused on the adaptivity-query tradeoff in submodular optimization [BS18, FMZ19, EN19, CQ19b, BRS19, CQ19a, ENV19, LLV20].

## 2  Preliminaries

**Multi-Distribution Learning (MDL)**  We follow the formulation of MDL in [HJZ22]. Let $\mathcal{X}$ be the instance space and $\mathcal{Y} = \{0, 1\}$ be the binary label space. Let $\mathcal{H} \subseteq \mathcal{Y}^{\mathcal{X}}$ be a hypothesis class and $d$ be its VC dimension. There are $k$ unknown data distributions $D_1, D_2, \ldots, D_k$ over $\mathcal{X} \times \mathcal{Y}$. In each round, the algorithm draws samples from the $k$ distributions. The number of samples may differ on the $k$ distributions, and may be chosen adaptively based on samples drawn in previous rounds. The *sample complexity* is the total number of samples drawn from all distributions in all rounds. An *r-round algorithm* draws $r$ rounds of samples and has a *round complexity* of $r$.

The goal is to learn a predictor $\hat{h} : \mathcal{X} \to \mathcal{Y}$ that performs well on all $k$ distributions $D_1, \ldots, D_k$. Formally, letting $\mathrm{err}(\hat{h}, D) := \mathbf{Pr}_{(x,y) \sim D}\left[\hat{h}(x) \neq y\right]$ denote the population error of predictor $\hat{h}$ on distribution $D$, an MDL algorithm is $(\varepsilon, \delta)$-*PAC (Probably Approximately Correct)* if

$$\max_{i \in [k]} \mathrm{err}(\hat{h}, D_i) \leq \mathsf{OPT} + \varepsilon \quad \text{where} \quad \mathsf{OPT} := \min_{h \in \mathcal{H}} \max_{i \in [k]} \mathrm{err}(h, D_i)$$

holds with probability at least $1 - \delta$ over the randomness in both the algorithm and the samples.

In the *realizable* setting, the data distributions are promised to satisfy $\mathsf{OPT} = 0$, i.e., there exists a perfect predictor $h^\star \in \mathcal{H}$ such that $\mathrm{err}(h^\star, D_i) = 0$ for every $i \in [k]$. We also refer to the general MDL setting—where $\mathsf{OPT}$ can be non-zero—as the *agnostic* setting.

**MDL via Game Dynamics**   Most previous agnostic MDL algorithms (e.g., [HJZ22, ZZC$^+$24, Pen24]) view the learning problem as a zero-sum game, in which the "min player" chooses a hypothesis (or a mixture of multiple hypotheses) and the "max player" chooses a mixture of the $k$ data distributions. These algorithms solve MDL by simulating the game dynamics when the two players follow certain strategies, e.g., best response or a no-regret online learning algorithm. The analysis then boils down to finding the sample size that suffices for simulating the game dynamics accurately. For instance, simulating a "min player" that best-responds to the "max player" is equivalent to finding a hypothesis that approximately minimizes the error on a given mixture of the $k$ distributions.

## 3   Sample-Adaptivity Tradeoff for Realizable MDL

### 3.1   Overview of Upper Bound

For the realizable setting of MDL, we present an algorithm that establishes a tradeoff between sample complexity and round complexity.

**Theorem 1** (Informal version of Theorem 10). *Algorithm 1 is an $r$-round $(\varepsilon, \delta)$-PAC algorithm for realizable MDL with sample complexity $O(k^{2/r} \log k \cdot \frac{d}{\varepsilon} + \frac{k \log(k) \log(k/\delta)}{\varepsilon})$.*

We highlight two special cases of the general tradeoff above. First, when $r = \log k$, Algorithm 1 is most sample-efficient and recovers the near-optimal sample complexity bound of $\widetilde{O}((d + k)/\varepsilon)$ for realizable MDL [BHPQ17, CZZ18, NZ18]. Second, with a small constant number of adaptive rounds (e.g., $r = 4$), Algorithm 1 has a sample complexity of $\widetilde{O}((d\sqrt{k} + k)/\varepsilon)$. Thus, our result demonstrates that even in the limited-adaptivity regime of $r = O(1)$, we can improve on the $\Omega(dk/\varepsilon)$ sample complexity required in the fully non-adaptive case of $r = 1$.

**Remark 1.** *Using more sophisticated variants of boosting such as boost-by-majority [SF12, Chapter 13] or recursive boosting [Sch90], it is possible to achieve a sample complexity of $\widetilde{O}((d\sqrt{k}+k)/\varepsilon)$ using exactly 3 rounds. Formal claims are deferred to Appendix A.*

**Additional Notations**   To state our algorithm succinctly, we introduce a few more notations. For a distribution $D$, we write $D^{\otimes m}$ as its $m$-fold product distribution, and $S \sim D^{\otimes m}$ is a shorthand for drawing a size-$m$ sample $S$ from $D$. At each iteration $t$ of the algorithm, we maintain weights $q_t(1), q_t(2), \ldots, q_t(k) \geq 0$ that sum up to 1. We also abuse the notation and write $q_t$ as the mixture distribution $\sum_{i=1}^{k} q_t(i) \cdot D_i$. While distributions $D_1, D_2, \ldots, D_k$ are unknown, the algorithm may still sample from the mixture $q_t$ easily: it suffices to first draw a random index $i$ such that $\mathbf{Pr}[i = j] = q_t(j)$ for each $j \in [k]$ and then sample from $D_i$.

---

**Algorithm 1:** Trade-off Multi-Distribution Learning

**Input:** Sample access to $k$ unknown distributions $D_1, \ldots, D_k$, an optimal PAC learner $\mathbb{A}$ for class $\mathcal{H}$, number of rounds $r$, target error $\varepsilon$, and failure probability $\delta$.

1 Set margin $\theta = \frac{r}{2 \log(k)}$, $p = \frac{1}{2} \cdot \left(4k^{2/r}\right)^{-1/(1-\theta)}$, $\tau = \frac{\varepsilon}{1+1/\theta}$, and $\alpha = \frac{1}{2} \ln\left(\frac{1-p}{p}\right)$.

2 Set $\varepsilon_{\mathbb{A}} = \frac{\tau p}{4}$, $\delta_{\mathbb{A}} = \frac{\delta}{2r}$, and $m = O\left(\frac{d + \log(1/\delta_{\mathbb{A}})}{\varepsilon_{\mathbb{A}}}\right)$.

3 Initialize $q_1(j) = \frac{1}{k}$ for each $j \in \{1, \ldots, k\}$, and let $q_1 = \sum_{j=1}^{k} q_1(j) D_j$.

4 **for** $t = 1, 2, \ldots, r$ **do**

5     Call learner $\mathbb{A}$ on a sample $\widetilde{S}_t \sim q_t^{\otimes m}$, and let $h_t$ be the returned predictor.

6     **for** $j = 1, 2, \ldots, k$ **do**

7        Draw a sample $S_{j,t} \sim D_j^{\otimes n}$, where $n = \frac{12}{\tau} \log(2rk/\delta)$.

8        Update:

$$q_{t+1}(j) = \frac{q_t(j)}{Z_t} \times \begin{cases} e^{-\alpha} & \text{if } \mathrm{err}(h_t, S_{j,t}) \leq \frac{\tau}{2} \\ e^{\alpha} & \text{if } \mathrm{err}(h_t, S_{j,t}) > \frac{\tau}{2} \end{cases}$$

       where $Z_t$ is a normalization constant that ensures $\sum_{j=1}^{k} q_{t+1}(j) = 1$.

**Output:** Majority-Vote Predictor $F : x \mapsto \mathbb{1}\left[\frac{1}{r} \sum_{t=1}^{r} h_t(x) \geq 1/2\right]$.

---

**Technical Overview**  We explain here the main ideas behind Algorithm 1; the full proof and analysis is deferred to Appendix A. We run a variant of the classical AdaBoost algorithm [SF12] to maximize a particular notion of "margin" that is defined on distributions $D_1, \ldots, D_k$. Specifically, in each round $1 \leq t \leq r$, Algorithm 1 calls learner $\mathbb{A}$ to learn a predictor $h_t$ that has a low error on $q_t$:

$$\sum_{j=1}^{k} q_t(j) \cdot \mathrm{err}(h_t, D_j) = \mathrm{err}(h_t, q_t) \leq \tau p.$$

Then, by Markov's inequality, $h_t$ also minimizes the fraction of distributions (as weighted by $q_t$) that have error more than $\tau$:

$$\sum_{j=1}^{k} q_t(j) \mathbb{1}\left[\mathrm{err}(h_t, D_j) > \tau\right] \leq p.$$

Subsequently, Algorithm 1 updates the weighted mixture $q_t$ over the $k$ distributions based on the thresholded loss function $\mathbb{1}[\mathrm{err}(h_t, D_j) > \tau]$ (as is done in AdaBoost). After $r$ rounds, the margin-maximization property of AdaBoost guarantees that

$$\frac{1}{k} \sum_{j=1}^{k} \mathbb{1}\left[\frac{1}{r} \sum_{t=1}^{r} \mathbb{1}[\mathrm{err}(h_t, D_j) > \tau] > \frac{1}{2} - \frac{\theta}{2}\right] \leq \prod_{t=1}^{r} 2\sqrt{(1-p)^{1+\theta} p^{1-\theta}}.$$

By choosing the margin parameter $\theta$ and $p$ such that $\prod_{t=1}^{r} 2\sqrt{(1-p)^{1+\theta} p^{1-\theta}} < 1/k$, we are guaranteed that, on each of the $k$ distributions, at least $1/2 + \theta/2$ fraction of the $r$ predictors have error at most $\tau$. Formally, it holds for every $j \in [k]$ that

$$\frac{1}{r} \sum_{t=1}^{r} \mathbb{1}[\mathrm{err}(h_t, D_j) > \tau] \leq \frac{1}{2} - \frac{\theta}{2}.$$

Finally, with this margin property, invoking Lemma 13 implies that the majority-vote predictor will have error at most $(1 + 1/\theta)\tau = \varepsilon$ on all $k$ distributions.

## 3.2  Overview of Lower Bound

The following theorem complements Theorem 1 by showing that a $k^{\Omega(1/r)}$ overhead is unavoidable.

**Theorem 2** (Informal version of Theorem 16). *For every $r = O(\log k)$ and sufficiently large d, every r-round algorithm for realizable MDL has an* $\Omega\left(\frac{dk^{1/r}}{r\log^2 k}\right)$ *sample complexity.*

This sample complexity lower bound nearly matches the $dk^{2/r} \log k$ term in Theorem 1, up to a $\mathrm{poly}(r, \log k)$ factor and a factor of 2 in the exponent of $k^{\Theta(1/r)}$.

We briefly sketch the proof of the $r = 2$ case, i.e., an $\Omega(d\sqrt{k})$ lower bound against two-round algorithms. We consider the class of linear functions over $\mathcal{X} = \mathbb{F}_2^d$, which has a VC dimension of $d$. The ground truth classifier $h^\star$ is drawn uniformly at random from all the $2^d$ linear functions. To construct the $k$ data distributions, we choose $k$ *difficulty levels* $\mathrm{diff}_1, \mathrm{diff}_2, \ldots, \mathrm{diff}_k$ as a uniformly random permutation of: (1) 1 copy of $\Theta(d)$; (2) $\sqrt{k}$ copies of $\Theta(d/\sqrt{k})$; (3) $k - \sqrt{k} - 1$ copies of $\Theta(d/k)$. Note that $\sum_{i=1}^{k} \mathrm{diff}_i \leq d$. Each data distribution $D_i$ is the uniform distribution over a randomly chosen $\mathrm{diff}_i$-dimensional subspace $V_i \subseteq \mathbb{F}_2^d$. Furthermore, the subspaces $V_1, V_2, \ldots, V_k$ are chosen such that they are linearly independent, i.e., $\dim(\mathrm{Span}(V_1 \cup V_2 \cup \cdots \cup V_k)) = \sum_{i=1}^{k} \mathrm{diff}_i$.

Intuitively, $\mathrm{diff}_i$ measures the "effective sample complexity" for learning $D_i$: $\Theta(\mathrm{diff}_i)$ samples are sufficient and necessary to learn an accurate classifier for $D_i$. In addition, since $h^\star$ is randomly chosen and the subspaces $V_1, V_2, \ldots, V_k$ are independent, samples collected from one distribution $D_i$ provide no information about the value of $h^\star$ on $V_j$ (except for the zero vector) for every $j \neq i$. Furthermore, if $\mathrm{diff}_i \in \{\Theta(d/\sqrt{k}), \Theta(d)\}$ and $m \ll d/\sqrt{k}$ samples have been drawn from $D_i$, it holds with high probability that the $m$ vectors in these samples are linearly independent. Then, the learner gains no information for distinguishing whether $\mathrm{diff}_i = \Theta(d/\sqrt{k})$ or $\mathrm{diff}_i = \Theta(d)$.

A *three-round* learner has a simple strategy: (1) In Round 1, draw $\Theta(d/k)$ samples from each distribution, thereby identifying the distributions with $\mathrm{diff}_i = \Theta(d/k)$ as well as learning the value

of $h^\star$ on each $V_i$; (2) In Round 2, draw $\Theta(d/\sqrt{k})$ samples from each of the $\sqrt{k}+1$ remaining distributions, which is sufficient for all distributions except the one with $\text{diff}_i = \Theta(d)$; (3) In Round 3, learn the only remaining distribution using $\Theta(d)$ samples. The resulting sample complexity is $O(d)$.

In contrast, a two-round learner must "skip" one of the three steps. For example, in Round 2 where there are still $\sqrt{k}+1$ "suspects" among which one distribution has difficulty level $\Theta(d)$, the learner could draw $\Theta(d)$ samples from each of them. Alternatively, the learner could draw $\Theta(d/\sqrt{k})$ samples from each distribution in Round 1, so that the distribution with $\text{diff}_i = \Theta(d)$ can be identified and then learned in Round 2. However, both strategies would have an $\Omega(d\sqrt{k})$ sample complexity.

The formal proof (in Appendix B) extends the hard instance construction to all $r = O(\log k)$ by using $r + 1$ different difficulty levels separated by a $k^{1/r}$ factor. We then formalize the intuition that every $r$-round MDL algorithm must "skip" a step and thus incur a $k^{1/r}$ overhead in the sample complexity.

## 4   Sample-Adaptivity Tradeoff for Agnostic MDL

For the agnostic setting, we show that the near-optimal sample complexity of $\widetilde{O}((d+k)/\varepsilon^2)$ can be achieved by a $\text{poly}(k)$-round algorithm.

**Proposition 1** (Corollaries 21 and 22). *There is a* $\min\{\widetilde{O}(\sqrt{k}), O(k \log k)\}$*-round MDL algorithm with sample complexity* $\widetilde{O}((d+k)/\varepsilon^2)$.

**The MDL Algorithm of [ZZC$^+$24]**   Our starting point is the approach of [ZZC$^+$24, Algorithm 1], which we briefly describe below. For brevity, we use $\widetilde{O}(\cdot)$ and $\widetilde{\Theta}(\cdot)$ to suppress $\text{polylog}(k, d, 1/\varepsilon, 1/\delta)$ factors, and let $\text{err}(h, S) \coloneqq \frac{1}{|S|} \sum_{(x,y) \in S} \mathbb{1}[h(x) \neq y]$ denote the empirical error of hypothesis $h : \mathcal{X} \to \mathcal{Y}$ on dataset $S \subseteq \mathcal{X} \times \mathcal{Y}$.

The algorithm maintains $k$ datasets $S_1, S_2, \ldots, S_k$, where each $S_i$ contains training examples drawn from $D_i$. The algorithm runs the Hedge algorithm for $T = \Theta((\log k)/\varepsilon^2)$ iterations starting at $w^{(1)} = (1/k, 1/k, \ldots, 1/k)$. Each iteration $t \in [T]$ consists of the following two steps:

- **ERM step:** For each $i \in [k]$, draw additional samples from $D_i$ and add them to $S_i$ until $|S_i| \geq w_i^{(t)} \cdot \widetilde{\Theta}((d+k)/\varepsilon^2)$. Then, find a hypothesis $h^{(t)} \in \mathcal{H}$ that minimizes the empirical error $\hat{L}(h) \coloneqq \sum_{i=1}^k w_i^{(t)} \cdot \text{err}(h, S_i)$, which is an estimate of the error of $h$ on $\sum_{i=1}^k w_i^{(t)} D_i$.

- **Hedge update step:** For each $i \in [k]$, draw $w_i^{(t)} \cdot \Theta(k)$ *fresh* samples from $D_i$ to obtain an estimate $r_i^{(t)} \approx \text{err}(h^{(t)}, D_i)$. Compute $w^{(t+1)}$ from $w^{(t)}$ and $r^{(t)}$ via a Hedge update.

The crux of the analysis of [ZZC$^+$24] is to show that the dataset sizes in the two steps above are sufficiently large, so that $h^{(t)}$ approximately minimizes the error on mixture $\sum_{i=1}^k w_i^{(t)} D_i$, and the reward vector $r^{(t)}$ is accurate enough for the Hedge update.

A straightforward implementation of the algorithm needs $T = \Theta((\log k)/\varepsilon^2)$ rounds of sampling. In comparison, the $\widetilde{O}(\sqrt{k})$ round complexity in Proposition 1 is lower when $\varepsilon \ll 1/k^{1/4}$.

**Hedge with Lazy Updates**   We prove Proposition 1 by modifying the algorithm of [ZZC$^+$24] so that it draws samples more lazily. The resulting algorithm is termed LazyHedge and formally defined in Algorithm 2. There are two versions of the algorithm—the "box" version and the "ellipsoid" version—that give the $O(k \log k)$ and $\widetilde{O}(\sqrt{k})$ round complexity bounds, respectively.

Similar to the Hedge algorithm, LazyHedge maintains a weight vector $w^{(t)}$ at each iteration $t$. In addition, it maintains a *cap vector* $\overline{w}_i^{(t)}$ as a proxy for the size of dataset $S_i$ at time $t$. At the start of iteration $t$, it checks whether $w^{(t)}$ is "observable" under cap $\overline{w}^{(t-1)}$ in the sense that $w^{(t)} \in \mathcal{O}(\overline{w}^{(t-1)})$. If the condition holds, no additional samples are drawn and the cap vector is left unchanged. Otherwise, the cap $\overline{w}^{(t)}$ is updated to $C$ times the entrywise maximum of all weight vectors so far, and additional samples are drawn so that both $|S_i|$ and $|S_{i,t}|$ match $\overline{w}_i^{(t)}$. Finally, LazyHedge computes the next weight vector $w^{(t+1)}$ from $w^{(t)}$ using the Hedge update rule.

---

**Algorithm 2:** LazyHedge: Hedge with Lazy Updates

---

**Input:** Number of distributions $k$, number of iterations $T = \Theta((\log k)/\varepsilon^2)$, step size $\eta = \Theta(\varepsilon)$, margin parameter $C > 1$.

1 **Box version:** Define $\mathcal{O}(\overline{w}) := \{w \in \Delta^{k-1} : w_i \leq \overline{w}_i, \ \forall i \in [k]\}$.

2 **Ellipsoid version:** Define $\mathcal{O}(\overline{w}) := \{w \in \Delta^{k-1} : \sum_{i=1}^{k} w_i^2/\overline{w}_i \leq 1\}$.

3 Set $w^{(1)} = (1/k, 1/k, \ldots, 1/k)$ and $\overline{w}^{(0)} = (0, 0, \ldots, 0)$.

4 Set $S_i = \emptyset$ for $i \in [k]$ and $S_{i,t} = \emptyset$ for $i \in [k]$ and $t \in [T]$.

5 **for** $t = 1, 2, \ldots, T$ **do**

6     **if** $w^{(t)} \in \mathcal{O}(\overline{w}^{(t-1)})$ **then**

7        Set $\overline{w}^{(t)} = \overline{w}^{(t-1)}$.

8     **else**

9        Set $\overline{w}_i^{(t)} = C \cdot \max\{w_i^{(1)}, w_i^{(2)}, \ldots, w_i^{(t)}\}$ for every $i \in [k]$.

10        Add samples from $D_i$ to $S_i$ until $|S_i| \geq \overline{w}_i^{(t)} \cdot \widetilde{\Theta}((d+k)/\varepsilon^2)$ for every $i \in [k]$.

11        Add samples from $D_i$ to $S_{i,t'}$ until $|S_{i,t'}| \geq \overline{w}_i^{(t)} \cdot \Theta(k)$ for every $i \in [k]$ and $t \leq t' \leq T$.

12     **ERM step:** Set $\hat{h}^{(t)} \in \operatorname{argmin}_{h \in \mathcal{H}} \sum_{i=1}^{k} w_i^{(t)} \cdot \operatorname{err}(h, S_i)$.

13     **Hedge update step:** Set $r_i^{(t)} = \operatorname{err}(\hat{h}^{(t)}, S_{i,t})$. Compute $w^{(t+1)} \in \Delta^{k-1}$ such that

$$w_i^{(t+1)} = \frac{w_i^{(t)} \cdot e^{\eta r_i^{(t)}}}{\sum_{j=1}^{k} w_j^{(t)} \cdot e^{\eta r_j^{(t)}}} \text{ for every } i \in [k].$$

**Output:** Randomized classifier uniformly distributed over $\{\hat{h}^{(1)}, \hat{h}^{(2)}, \ldots, \hat{h}^{(T)}\}$.

---

The correctness and sample complexity of LazyHedge follow from the analysis of [ZZC+24]. At a high level, either version of LazyHedge ensures that using $S_i$ in the ERM step and using $S_{i,t}$ in the Hedge update step lead to low-variance estimates, which allow the analysis of [ZZC+24] to go through. We provide a more detailed analysis in Appendix C.4.

It remains to upper bound the round complexity of LazyHedge, namely, the number of times the cap vector is updated in Line 9. For the box version, we have an $O(k \log k)$ upper bound.

**Proposition 2.** *The box version of* LazyHedge *takes at most* $O(k \log k)$ *rounds.*

*Proof sketch.* If the cap vector is updated in the $t$-th iteration, there exists $i \in [k]$ such that $w_i^{(t)} > \overline{w}_i^{(t-1)}$. We call such index $i$ the *culprit* of this cap update. For index $i$ to be the culprit, the historical high of $w_i^{(t)}$ must have increased by a factor of $C$ since the last cap update. As this historical high is non-decreasing and in $[1/k, 1]$, each index $i$ can be the culprit at most $O(\log_C k)$ times. Thus, the round complexity is at most $k \cdot O(\log_C k) = O(k \log k)$ for any constant $C > 1$. $\qquad\square$

For the ellipsoid version, a more involved analysis gives an $\widetilde{O}(\sqrt{k})$ round complexity bound.

**Proposition 3.** *The ellipsoid version of* LazyHedge *takes at most* $\widetilde{O}(\sqrt{k})$ *rounds.*

The analysis applies the following technical lemma shown by [ZZC+24].

**Lemma 3** (Lemma 3 of [ZZC+24]). *For some choice of* $T = \Theta((\log k)/\varepsilon^2)$ *and* $\eta = \Theta(\varepsilon)$ *in* LazyHedge, *it holds with probability* $1 - \delta$ *that* $\sum_{i=1}^{k} \max_{1 \leq t \leq T} w_i^{(t)} \leq O(\log^8(k/(\varepsilon\delta))) = \widetilde{O}(1)$.

*Proof sketch of Proposition 3.* We classify the cap updates into two types: A "Type I" update is when some coordinate $w_i$ reaches a historical high of $> 1/\sqrt{k}$, and a "Type II" update is one without a significant increase in any coordinate. We show that either type of cap updates happen $\widetilde{O}(\sqrt{k})$ times.

The upper bound for Type I updates follows from Lemma 3, which implies that there are at most $\widetilde{O}(1) \cdot \sqrt{k}$ Type I updates where the coordinate reaches $\approx 1/\sqrt{k}$, at most $\widetilde{O}(1) \cdot \sqrt{k}/2$ Type I updates where the coordinate reaches $\approx 2/\sqrt{k}$, and so on. These upper bounds sum up to $\widetilde{O}(\sqrt{k})$.

The analysis for Type II updates is more involved. Roughly speaking, we say that a coordinate $i \in [k]$ gains a *potential* of $a^2/b$ when the historical high of $w_i$ increases from $b$ to $a$ through the

Hedge dynamics. The $\widetilde{O}(\sqrt{k})$ bound follows from two technical claims: (1) Each Type II update may happen only if a total potential of $\Omega(1)$ is accrued over all $k$ coordinates; (2) The total potential that the $k$ coordinates may contribute to Type II updates is at most $\widetilde{O}(\sqrt{k})$. $\square$

# 5 A General Framework: Optimization via On-Demand Sampling

## 5.1 Problem Setup

In Optimization via On-Demand Sampling (OODS), the goal is to maximize a concave function $f$ over the probability simplex $\Delta^{k-1} := \{w \in \mathbb{R}^k : \sum_{i=1}^k w_i = 1, w_i \geq 0 \ \forall i \in [k]\}$ by "sampling" from the $k$ coordinates. The algorithm does not have full access to $f$; instead, it maintains a *cap vector* $\overline{w} \in [0, 1]^k$ that specifies the *observable region* of the simplex. We focus on two concrete settings of the problem, where the observable region is either a box or an ellipsoid defined by $\overline{w}$.

**Definition 1** (Optimization via On-Demand Sampling). $f : \Delta^{k-1} \to [0, 1]$ is an unknown concave function. In each round $t = 1, 2, \ldots, r$, the algorithm chooses cap $\overline{w}^{(t)} \in [0, 1]^k$ that is lower bounded by $\overline{w}^{(t-1)}$ entry-wise (if $t > 1$). Then, the algorithm makes arbitrarily many queries to a first-order oracle of $f$—which returns the value and a supergradient—at any $w \in \mathcal{O}(\overline{w}^{(t)})$, where $\mathcal{O}(\overline{w}) := \{w \in \Delta^{k-1} : w_i \leq \overline{w}_i, \ \forall i \in [k]\}$ in the box setting, and $\mathcal{O}(\overline{w}) := \{w \in \Delta^{k-1} : \sum_{i=1}^k w_i^2/\overline{w}_i \leq 1\}$ in the ellipsoid setting. The goal is to find $\hat{w} \in \Delta^{k-1}$ such that $f(\hat{w}) \geq \max_{w \in \Delta^{k-1}} f(w) - \varepsilon$ while minimizing the *sample overhead* $\sum_{i=1}^k \overline{w}_i^{(r)}$ and the *round complexity* $r$.

To see how OODS connects to MDL and on-demand sampling in general, we view the $k$ coordinates as distributions $D_1, D_2, \ldots, D_k$ from which the algorithm may sample. Maximizing $f(w)$ can then be viewed as optimizing the mixing weights in mixture $\sum_{i=1}^k w_i D_i$. The cap $\overline{w}_i$ is a proxy for and proportional to the number of samples that have already been drawn from $D_i$. In light of this analogy, the sample overhead $\sum_{i=1}^k \overline{w}_i^{(r)}$ is simply a proxy for the total number of samples that the algorithm draws, while the round complexity $r$ is the number of rounds of on-demand sampling.

The observable region $\mathcal{O}(\overline{w})$ represents the mixing weights $w \in \Delta^{k-1}$ on which $f(w)$ can be accurately estimated using the current dataset specified by $\overline{w}$. In the box setting, the algorithm is only allowed to query $f(w)$ if $w_i \leq \overline{w}_i$ holds for every $i \in [k]$, which can be viewed as a sufficient condition for the algorithm to obtain an accurate estimate for mixture $\sum_{i=1}^k w_i D_i$ using the $\Theta(\overline{w}_i)$ samples collected from each $D_i$. In the ellipsoid setting, we use the more refined condition $\sum_{i=1}^k w_i^2/\overline{w}_i \leq 1$, where the summation is a proxy for the variance in estimating the mixture $\sum_{i=1}^k w_i D_i$ using the datasets. More details on how the box and ellipsoid settings connect to MDL can be found in Appendix C.4.

## 5.2 Overview of Upper Bounds

For the OODS problem, we give a simple algorithm with a $\text{poly}(k)$ round complexity. The algorithm is also termed LazyHedge, as it is almost identical to the agnostic MDL algorithm in Section 4. We formally define the algorithm (Algorithm 3) in Appendix C for completeness. To guarantee a sample overhead of $\widetilde{O}(s)$, the algorithm takes $\widetilde{O}(k/s)$ rounds in the box setting and $\widetilde{O}(\sqrt{k/s})$ rounds in the ellipsoid setting. Here, the $\widetilde{O}(\cdot)$ notation hides $\text{polylog}(k/\varepsilon)$ factors, where $\varepsilon$ is the accuracy parameter in OODS.

**Theorem 4** (Informal version of Theorems 8 and 20). *There is an OODS algorithm with sample overhead $\widetilde{O}(s)$ that takes $\widetilde{O}(k/s)$ rounds in box setting and $\widetilde{O}(\sqrt{k/s})$ rounds in ellipsoid setting.*

**Hedge with Lazy Updates**   We apply the same LazyHedge strategy as in Algorithm 2 for MDL. LazyHedge maintains a weight vector $w^{(t)}$ at each iteration $t$. At the start of iteration $t$, it checks whether $w^{(t)}$ is still in the observable region $\mathcal{O}(\overline{w}^{(t-1)})$ specified by the previous cap $\overline{w}^{(t-1)}$. If so, the cap is left unchanged; otherwise, the cap $\overline{w}^{(t)}$ is set to $C$ times the entrywise maximum of all weight vectors so far. Then, LazyHedge queries the first-order oracle to obtain a supergradient $r^{(t)}$ at $w^{(t)}$, and computes the next weight vector $w^{(t+1)}$ using the Hedge update. While LazyHedge takes

$T = \Theta((\log k)/\varepsilon^2)$ *iterations*, its *round complexity* is the number of times the cap is updated, which can be much lower than $T$.

**Analysis for the Box Setting**   We sketch the analysis for the box setting, and defer the ellipsoid setting to Appendix C. The standard regret analysis of Hedge shows that LazyHedge finds an $O(\varepsilon)$-approximate maximum. We prove the following lemma in Appendix C.1.

**Lemma 5.** LazyHedge *outputs* $\hat{w} \in \Delta^{k-1}$ *such that* $f(\hat{w}) \geq \max_{w \in \Delta^{k-1}} f(w) - O(\varepsilon)$.

Next, we show that the sample overhead of LazyHedge is low. Recall that $C > 1$ is the margin parameter used in LazyHedge for the cap updates.

**Lemma 6.** LazyHedge *has an* $O(C \log^8(k/\varepsilon))$ *sample overhead.*

*Proof.* LazyHedge guarantees that $\overline{w}_i^{(T)} \leq C \cdot \max_{1 \leq t \leq T} w_i^{(t)}$ for every $i \in [k]$. By Lemma 3, the sample overhead is $\sum_{i=1}^k \overline{w}_i^{(T)} \leq C \cdot \sum_{i=1}^k \max_{1 \leq t \leq T} w_i^{(t)} = O(C \cdot \log^8(k/\varepsilon))$. $\qquad\square$

It remains to upper bound the round complexity of LazyHedge in the box setting. The proof of the following lemma resembles and extends that of Proposition 2 in the MDL setting.

**Lemma 7.** LazyHedge *takes* $\min\left\{k, O((k/C) \cdot \log^8(k/\varepsilon))\right\} \cdot O(\log_C k)$ *rounds in the box setting.*

*Proof sketch.* If the cap is updated in the $t$-th iteration of LazyHedge, there exists $i \in [k]$ such that $w_i^{(t)} > \overline{w}_i^{(t-1)}$. We call such index $i$ the *culprit* of this cap update. By the same argument as in Proposition 2, each index $i$ can be the culprit of at most $O(\log_C k)$ cap updates. It remains to bound the number of indices that become the culprit of at least one cap update. This number is trivially at most $k$. Furthermore, since LazyHedge sets $\overline{w}^{(1)} = (C/k, C/k, \ldots, C/k)$ in the first iteration, for index $i$ to become the culprit of a later cap update, $w_i^{(t)}$ must reach $C/k$ for some $t$. By Lemma 3, at most $\widetilde{O}(1)/(C/k) = \widetilde{O}(k/C)$ indices can satisfy this. Thus, the round complexity is at most $\min\{k, \widetilde{O}(k/C)\} \cdot O(\log_C k)$. $\qquad\square$

Combining Lemmas 5, 6 and 7 immediately gives the first part of Theorem 4.

**Theorem 8.** *For any* $C \in [2, k]$, *in the box setting,* LazyHedge *finds an* $O(\varepsilon)$-*approximate maximum with an* $O(C \log^8(k/\varepsilon))$ *sample overhead in* $\min\{O(k \log k), O((k/C) \cdot \log^9(k/\varepsilon))\}$ *rounds.*

### 5.3   Overview of Lower Bounds

The following theorem shows that the $\text{poly}(k/s)$ round complexity in Theorem 4 cannot be avoided when $\varepsilon \leq 1/\text{poly}(k)$. For exponentially small $\varepsilon$, the exponents on $k/s$ also match Theorem 4.

**Theorem 9** (Informal version of Theorems 25 and 28)**.** *If* $\varepsilon \leq O(1/k)$, *every OODS algorithm with sample overhead* $s$ *must take* $\Omega(\sqrt{k/s})$ *rounds in the box setting and* $\Omega((k/s)^{1/4})$ *rounds in the ellipsoid setting. If* $\varepsilon \leq e^{-\Omega(k)}$, *every OODS algorithm with sample overhead* $s$ *must take* $\Omega(k/s)$ *rounds in the box setting and* $\Omega(\sqrt{k/s})$ *rounds in the ellipsoid setting.*

As a corollary, if $\varepsilon \leq O(1/k)$, every OODS algorithm has either a $\text{poly}(k)$ round complexity or a sample overhead that is almost linear in $k$. While these lower bounds do not directly imply lower bounds for agnostic MDL, they show that further improving the round complexity in Proposition 1 requires a substantially different approach. Roughly speaking, the MDL algorithm of [ZZC$^+$24] fits into the OODS framework because: (1) It uses samples in a restricted way: finding an ERM $\hat{h}$ on mixture $\sum_{i=1}^k w_i D_i$ for some weight vector $w$ in an "observable region", and estimating the error of $\hat{h}$ on every $D_i$; (2) By solving the MDL instance, it finds a "hard" mixture $\sum_{i=1}^k \hat{w}_i D_i$ on which the best hypothesis in $\mathcal{H}$ has an error close to the minimax value. The first property ensures that the MDL algorithm requires no more information than what the first-order oracle provides in OODS. The second ensures that the MDL algorithm implicitly solves the OODS problem. Theorem 9 then suggests that every algorithm with the two properties faces an inherent obstacle in achieving a sub-polynomial round complexity.

We sketch the proof of Theorem 9 in the box setting and the $\varepsilon \leq O(1/k)$ regime; the formal proofs are deferred to Appendix D. We consider the objective function $f(w) := \min_{j \in [m]} \{w_{i_j^\star} + j/m^2\}$, where $i_1^\star, i_2^\star, \ldots, i_m^\star \in [k]$ are $m \leq k$ different *critical indices* sampled uniformly at random. The lower bound builds on two observations on $f$: (1) (Lemma 23) If $\varepsilon \leq O(1/k)$, every $\varepsilon$-approximate maximum must put an $\Omega(1/m)$ weight on at least half of the critical indices $i_1^\star, \ldots, i_m^\star$; (2) (Lemma 24) Unless we put a weight of $> 1/m^2$ on each of $i_1^\star, i_2^\star, \ldots, i_j^\star$, the value of $f(w)$ is determined by the first $j$ terms in the minimum. Intuitively, unless we already "know" the first $j$ critical indices, we cannot learn the values of $i_{j+1}^\star, \ldots, i_m^\star$ from the first-order oracle.

There is a natural $m$-round algorithm that solves the instance above. In the first round, we query the uniform weight vector $w = (1/k, 1/k, \ldots, 1/k)$ to learn the value of $i_1^\star$. In the second round, we query $f$ on some $w$ with $w_{i_1^\star} \gg 1/m^2$, thereby learning the value of $i_2^\star$. Repeating this $m$ times recovers all the critical indices. One might hope to be "more clever" and learn many critical indices in a round. For example, the algorithm might put a cap of $\gg 1/m^2$ on several coordinates in the first round, in the hope of hitting more than one indices in $i_1^\star, i_2^\star, \ldots$. However, if the algorithm has a sample overhead of $s$, only $O(m^2 s)$ such guesses can be made. In particular, assuming $m \ll \sqrt{k/s}$, the $O(m^2 s) \ll k$ guesses only cover a tiny fraction of the indices. Thus, over the uniform randomness in $i^\star$, the algorithm learns only $O(1)$ critical indices within each round in expectation.

## 6  Discussion

In this work, we formalized the trade-off between sample and round complexities in multi-distribution learning (MDL). For the realizable case, we obtained a nearly tight characterization: when the learner is allowed $r$ rounds of sampling, the optimal sample complexity is proportional to $k^{\Theta(1/r)}$. In particular, a constant number of rounds suffice to achieve a sublinear dependence on $k$, whereas nearly $\log k$ rounds are necessary to reach near-optimal sample complexity.

For the more general agnostic setting, we introduced the *optimization via on-demand sampling* (OODS) problem as an abstraction of the common approach shared by many recent MDL algorithms. We then leveraged the intuition behind the OODS algorithms to obtain an improved round complexity of $\widetilde{O}(\sqrt{k})$. On the negative side, any MDL algorithm based on the OODS approach must take $\mathrm{poly}(k)$ rounds to match the near-optimal sample complexity of $\widetilde{O}((d+k)/\varepsilon^2)$.

To further understand the landscape of sample-adaptivity trade-off in agnostic MDL, we highlight the following concrete open question:

**Open Question 1.** *Does there exist an agnostic MDL algorithm that simultaneously achieves sample complexity $\widetilde{O}\left(\frac{d+k}{\varepsilon^2}\right)$ and round complexity $\mathrm{polylog}(k/\varepsilon)$?*

Our results on OODS may shed light on both directions. The lower bounds for OODS suggest that any such algorithm must leverage the data in a more sophisticated manner—beyond invoking an ERM oracle or merely evaluating empirical risks. Notably, a recent algorithm of Peng [Pen24] is one such candidate: despite having a high round complexity (cf. Table 1), it uses the collected data to construct a "refined" hypothesis class and might therefore circumvent the OODS lower bounds.

Conversely, to establish a round lower bound for sample-optimal MDL algorithms, a natural approach would be to recast our hard instances for OODS as MDL instances. The key challenge, however, lies in ensuring that *every* MDL algorithm gains no more information from the samples than OODS-based learners do. Developing such a reduction would establish a deeper connection between MDL and OODS, and represent a major step toward a complete understanding of the sample-adaptivity trade-off.

## Acknowledgments and Disclosure of Funding

This work was supported in part by the National Science Foundation under grant CCF-2145898, by the Office of Naval Research under grant N00014-24-1-2159, an Alfred P. Sloan fellowship, and a Schmidt Science AI2050 fellowship. Any opinions, findings, and conclusions or recommendations expressed in this material are those of the author(s) and do not necessarily reflect the views of sponsoring agencies.

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

# A Upper Bound for Realizable MDL

In this section, we prove the following upper bound on sample-adaptivity tradeoff for realizable multi-distribution learning:

**Theorem 10.** *For any $k$ unknown distributions $D_1, \ldots, D_k$, any class $\mathcal{H}$ such that $\mathsf{OPT} = 0$, and any target error $\varepsilon$ and number of rounds $r \leq \log(k)$, the output predictor $F$ of Algorithm 1 satisfies with probability at least $1 - \delta$,*

$$\forall 1 \leq j \leq k : \mathrm{err}(F, D_j) \leq \varepsilon.$$

*Furthermore, Algorithm 1 uses $r$ adaptive rounds and has a sample complexity of*

$$O\left( k^{2/r} \log(k) \frac{d}{\varepsilon} + \frac{k \log(k)}{\varepsilon} \log\left( \frac{k}{\delta} \right) \right).$$

Before proceeding with the proof of Theorem 10, we state a few lemmas that will be useful in the proof. First, instead of using plain ERM which incurs a sample complexity of $O(\frac{d \log(1/\varepsilon) + \log(1/\delta)}{\varepsilon})$, it will be particularly advantageous for us to use an optimal PAC learner that avoids the $\log(1/\varepsilon)$ factor in sample complexity.

**Lemma 11** (Optimal Sample Complexity of PAC Learning, [Han16, Lar23]). *For any class $\mathcal{H}$, there exists an improper learner $\mathbb{A}$, such that for any distribution $D$ where $\inf_{h \in \mathcal{H}} \mathrm{err}(h, D) = 0$, with probability at least $1 - \delta$ over $S \sim D^m$ where $m = O(\frac{d + \log(1/\delta)}{\varepsilon})$, $\mathrm{err}(\mathbb{A}(S), D) \leq \varepsilon$. In particular, as shown by [Lar23], bagging combined with $\mathsf{ERM}$ yields an optimal PAC learner $\mathbb{A}$.*

Next, the lemma below essentially allows us to argue that we can perform weight updates in Algorithm 1 based on empirical error instead of population error.

**Lemma 12** (Empirical Samples and Population Error). *For any fixed predictor $h : \mathcal{X} \to \mathcal{Y}$, any distribution $D$ over $\mathcal{X} \times \mathcal{Y}$, any $\delta \in (0, 1)$, any $\tau \in (0, 1/2)$, with probability at least $1 - \delta$ over $S \sim D^n$ where $n = \frac{12}{\tau} \log(1/\delta)$, the followings holds:*

1. *If $\mathrm{err}(h, D) > \tau$ then $\mathrm{err}(h, S) > \tau/2$, and*

2. *if $\mathrm{err}(h, S) > \tau/2$ then $\mathrm{err}(h, D) > \tau/4$.*

*Proof.* We apply a standard Chernoff bound. $\square$

Finally, the lemma below shows that, when at least a $1/2 + \theta/2$ fraction of predictors achieve error at most $\tau$ on a distribution $D$, the majority-vote achieves error at most $(1 + 1/\theta)\tau$ on $D$.

**Lemma 13** (Weighted-Majority Error Bound). *Let $D$ be an arbitrary distribution over $\mathcal{X} \times \mathcal{Y}$, $\tau \in (0, 1)$, $\theta \in (0, 1)$. For any predictors $h_1, \ldots, h_r$ with corresponding (non-negative) weights $\alpha_1, \ldots, \alpha_r$ that satisfy*

$$\sum_{t=1}^{r} \alpha_t \mathbb{1}[\mathrm{err}(h_t, D) \leq \tau] - \sum_{t=1}^{r} \alpha_t \mathbb{1}[\mathrm{err}(h_t, D) > \tau] \geq \theta \sum_{t=1}^{r} \alpha_t,$$

*it holds that*

$$\Pr_{(x,y) \sim D} \left[ \mathbb{1}\left[ \sum_{t=1}^{r} \alpha_t h_t(x) \geq \frac{1}{2} \sum_{t=1}^{r} \alpha_t \right] \neq y \right] \leq \left( 1 + \frac{1}{\theta} \right) \tau.$$

*Proof.* Let $G_\tau = \{1 \leq t \leq r : \mathrm{err}(h_t, D) \leq \tau\}$, and $B_\tau = [r] \setminus G_\tau$. Without loss of generality, we will consider the normalized weights, i.e., we assume $\sum_{t=1}^{r} \alpha_t = 1$. Let $W_G = \sum_{t \in G_\tau} \alpha_t$ and

$W_B = \sum_{t \in B_\tau} \alpha_t$. Observe that

$$\mathbf{Pr}_{(x,y)\sim D}\left[\mathbb{1}\left[\sum_{t=1}^{r}\alpha_t h_t(x) \geq \frac{1}{2}\right] \neq y\right] \leq \mathbf{Pr}_{(x,y)\sim D}\left[\sum_{t=1}^{r}\alpha_t\mathbb{1}[h_t(x) \neq y] \geq \frac{1}{2}\right]$$

$$\leq \mathbf{Pr}_{(x,y)\sim D}\left[\sum_{t \in G_\tau}\alpha_t\mathbb{1}[h_t(x) \neq y] \geq \frac{1}{2} - W_B\right]$$

$$\leq \frac{\sum_{t \in G_\tau}\alpha_t\mathrm{err}(h_t, D)}{\frac{1}{2} - W_B} \leq \frac{\tau W_G}{\frac{1}{2} - W_B}$$

$$= \tau \cdot \frac{2W_G}{1 - 2W_B} = \tau \cdot \frac{2W_G}{W_G - W_B}$$

$$= \tau \cdot \frac{(W_G + W_B) + (W_G - W_B)}{W_G - W_B}$$

$$= \tau\left(\frac{1}{W_G - W_B} + 1\right) \leq \tau\left(1 + \frac{1}{\theta}\right),$$

where we have used Markov's inequality and the fact that $W_G - W_B \geq \theta$. $\qquad\square$

We are now ready to proceed with the proof of Theorem 10.

*Proof of Theorem 10.* Before analyzing Algorithm 1, with probability at least $1 - \delta/2$ over sampling in Line 5, we have the following PAC learning guarantee

$$\forall 1 \leq t \leq r : \mathrm{err}(h_t, q_t) = \sum_{j=1}^{k} q_t(j)\mathrm{err}(h_t, D_j) \leq \varepsilon_{\mathbb{A}} = \frac{\tau p}{4}. \tag{1}$$

Ideally, we would compute weight updates in a round $t \in [r]$ in Line 8 in Algorithm 1 based on evaluating the population error $\mathrm{err}(h_t, D_j)$ for each $j \in \{1, \ldots, k\}$. However, we only have access to samples from each $D_j$. In the analysis below, we show that we can effectively carry out the weight updates based on samples. This hinges on the following property which follows from invoking Lemma 12: with probability at least $1 - \delta/2$ over sampling in Line 7, we are guaranteed

$$\begin{aligned}\forall 1 \leq t \leq r, \forall 1 \leq j \leq k : \quad &\mathrm{err}(h_t, D_j) > \tau \Rightarrow \mathrm{err}(h_t, S_{j,t}) > \tau/2, \\ &\mathrm{err}(h_t, S_{j,t}) > \tau/2 \Rightarrow \mathrm{err}(h_t, D_j) > \tau/4.\end{aligned} \tag{2}$$

We now proceed with analysis of Algorithm 1 assuming Equations (1) and (2) hold simultaneously, which happens with probability at least $1 - \delta$. The essence of the proof lies in analyzing an appropriate "margin-type" loss function defined based on the majority-vote predictor $F(x) = \mathbb{1}\left[\frac{1}{r}\sum_{t=1}^{r}h_t(x) \geq 1/2\right]$ and the distributions $D_1, \ldots, D_k$. Specifically, for each distribution $D_j$, we will analyze the following 0-1 loss function:

$$L_{\tau,\theta}(F, D_j) = \mathbb{1}\left[\sum_{t=1}^{r}\mathbb{1}\left[\mathrm{err}(h_t, D_j) \leq \tau\right] - \sum_{t=1}^{r}\mathbb{1}\left[\mathrm{err}(h_t, D_j) > \tau\right] \leq \theta \cdot r\right].$$

This loss function considers the "margin" which in this context is the fraction of good predictors achieving error below threshold $\tau$ minus the fraction of bad predictors which have error above $\tau$:

$$\frac{1}{r}\left(\sum_{t=1}^{r}\mathbb{1}\left[\mathrm{err}(h_t, D_j) \leq \tau\right] - \sum_{t=1}^{r}\mathbb{1}\left[\mathrm{err}(h_t, D_j) > \tau\right]\right).$$

The loss function $L_{\tau,\theta}(F, D_j)$ evaluates to 0 if and only if the margin is greater than $\theta$, i.e., the fraction of predictors $h_t$ achieving population error $\mathrm{err}(h_t, D_j)$ below the threshold $\tau$ is greater than $\frac{1}{2} + \frac{\theta}{2}$.

Observe now that when $L_{\tau,\theta}(F, D_j) = 1$, by definition,

$$\sum_{t=1}^{r} \alpha \mathbb{1}\left[\text{err}(h_t, D_j) \leq \tau\right] - \sum_{t=1}^{r} \alpha \mathbb{1}\left[\text{err}(h_t, D_j) > \tau\right] \leq \theta \sum_{t=1}^{r} \alpha$$

$$\iff \sum_{t=1}^{r} \alpha \left(\mathbb{1}\left[\text{err}(h_t, D_j) > \tau\right] - \mathbb{1}\left[\text{err}(h_t, D_j) \leq \tau\right]\right) + \theta \sum_{t=1}^{r} \alpha \geq 0$$

$$\iff \exp\left(\sum_{t=1}^{r} \alpha \left(\mathbb{1}\left[\text{err}(h_t, D_j) > \tau\right] - \mathbb{1}\left[\text{err}(h_t, D_j) \leq \tau\right]\right) + \theta \sum_{t=1}^{r} \alpha\right) \geq 1.$$

Thus, we can bound from above the fraction of distributions $D_j$ for which $L_{\tau,\theta}(F, D_j) = 1$ as follows

$$\frac{1}{k} \sum_{j=1}^{k} L_{\tau,\theta}(F, D_j)$$

$$\leq \frac{1}{k} \sum_{j=1}^{k} \exp\left(\sum_{t=1}^{r} \alpha \left(\mathbb{1}\left[\text{err}(h_t, D_j) > \tau\right] - \mathbb{1}\left[\text{err}(h_t, D_j) \leq \tau\right]\right) + \theta \sum_{t=1}^{r} \alpha\right)$$

$$= \exp\left(\theta \sum_{t=1}^{r} \alpha\right) \sum_{j=1}^{k} \frac{1}{k} \exp\left(\sum_{t=1}^{r} \alpha \left(\mathbb{1}\left[\text{err}(h_t, D_j) > \tau\right] - \mathbb{1}\left[\text{err}(h_t, D_j) \leq \tau\right]\right)\right)$$

$$\leq \exp\left(\theta \sum_{t=1}^{r} \alpha\right) \sum_{j=1}^{k} \frac{1}{k} \exp\left(\sum_{t=1}^{r} \alpha \left(\mathbb{1}\left[\text{err}(h_t, S_{j,t}) > \tau/2\right] - \mathbb{1}\left[\text{err}(h_t, S_{j,t}) \leq \tau/2\right]\right)\right),$$

where the last inequality follows from Equation (2). Observe now that by the update rule in Line 9 in Algorithm 1,

$$q_{r+1}(j) = \frac{q_1(j) \exp\left(\sum_{t=1}^{r} \alpha \left(\mathbb{1}\left[\text{err}(h_t, S_{j,t}) > \tau/2\right] - \mathbb{1}\left[\text{err}(h_t, S_{j,t}) \leq \tau/2\right]\right)\right)}{\prod_{t=1}^{r} Z_t}.$$

Thus, by combining the above, it follows that

$$\frac{1}{k} \sum_{j=1}^{k} L_{\tau,\theta}(F, D_j) \leq \exp\left(\theta \sum_{t=1}^{r} \alpha\right) \prod_{t=1}^{r} Z_t = \prod_{t=1}^{r} e^{\theta\alpha} Z_t, \tag{3}$$

where the normalization constant

$$Z_t = \sum_{j=1}^{k} q_t(j) \exp\left(\alpha \left(\mathbb{1}\left[\text{err}(h_t, S_{j,t}) > \tau/2\right] - \mathbb{1}\left[\text{err}(h_t, S_{j,t}) \leq \tau/2\right]\right)\right).$$

Observe that, by Equation (2), if $\text{err}(h_t, S_{j,t}) > \tau/2$ then $\text{err}(h_t, D_j) > \tau/4$, thus we can bound $Z_t$ from above as follows

$$Z_t \leq \sum_{j=1}^{k} q_t(j) \exp\left(\alpha \left(\mathbb{1}\left[\text{err}(h_t, D_j) > \tau/4\right] - \mathbb{1}\left[\text{err}(h_t, D_j) \leq \tau/4\right]\right)\right) = e^{-\alpha}(1 - p_t) + e^{\alpha} p_t,$$

where $p_t = \mathbf{Pr}_{j \sim q_t}\left[\text{err}(h_t, D_j) > \tau/4\right]$. Next, we bound $p_t$ from above by the parameter $p$. To do this, we invoke Markov's inequality and the PAC learning guarantee in Equation (1),

$$p_t = \mathbf{Pr}_{j \sim q_t}\left[\text{err}(h_t, D_j) > \tau/4\right] \leq \frac{\mathbb{E}_{j \sim q_t}\left[\text{err}(h_t, D_j)\right]}{\tau/4} \leq \frac{\varepsilon_{\mathbb{A}}}{\tau/4} = p. \tag{4}$$

Since $\alpha = \frac{1}{2} \ln\left(\frac{1-p}{p}\right)$ and $1 - p > p$, combined with the above, we get

$$Z_t \leq e^{-\alpha}(1 - p) + e^{\alpha} p = \sqrt{\frac{p}{1-p}}(1 - p) + \sqrt{\frac{1-p}{p}} p = 2\sqrt{p(1 - p)}.$$

Combining Equation (3) with the above bound on $Z_t$, we get

$$\frac{1}{k}\sum_{j=1}^{k} L_{\tau,\theta}(F, D_j) \leq \prod_{t=1}^{r}\left(e^{\theta\alpha}\cdot 2\sqrt{p(1-p)}\right) = \prod_{t=1}^{r} 2\sqrt{(1-p)^{1+\theta}p^{1-\theta}}. \tag{5}$$

Observe that by the bound established in Equation (5), what remains is to solve for suitable values of $\theta$ and $p$ such that

$$\prod_{t=1}^{r} 2\sqrt{(1-p)^{1+\theta}p^{1-\theta}} < \frac{1}{k},$$

because that would imply $\frac{1}{k}\sum_{j=1}^{k} L_{\tau,\theta}(F, D_j) < 1/k$, and hence the majority-vote predictor $F$ has "margin" $\theta$ on all $k$ distributions:

$$\forall 1 \leq j \leq k: \sum_{t=1}^{r} \alpha\mathbb{1}\left[\mathrm{err}(h_t, D_j) \leq \tau\right] - \sum_{t=1}^{r} \alpha\mathbb{1}\left[\mathrm{err}(h_t, D_j) > \tau\right] > \theta\sum_{t=1}^{r} \alpha.$$

Combining this margin guarantee and Lemma 13 implies that $F$ achieves error at most $\varepsilon$ on all $k$ distributions, i.e., for every $j \in [k]$,

$$\mathrm{err}(F, D_j) = \Pr_{(x,y)\sim D_j}\left[\mathbb{1}\left[\frac{1}{r}\sum_{t=1}^{r} h_t(x) \geq \frac{1}{2}\right] \neq y\right] \leq (1+1/\theta)\tau = (1+1/\theta)\frac{\varepsilon}{1+1/\theta} = \varepsilon.$$

We now turn to choosing $p$ and $\theta$ to guarantee the above.

$$\prod_{t=1}^{r} 2\sqrt{(1-p)^{1+\theta}p^{1-\theta}} < \frac{1}{k} \iff \left(4(1-p)^{1+\theta}p^{1-\theta}\right)^{r/2} < \frac{1}{k}$$

$$\iff 4(1-p)^{1+\theta}p^{1-\theta} < \frac{1}{k^{2/r}}.$$

Observe that $(1-p)^{(1+\theta)} \leq 1$ for all $\theta, p \in (0,1)$. Thus, it suffices to choose $p, \theta$ so that

$$\left(\frac{1}{p}\right)^{1-\theta} > 4k^{2/r}, \tag{6}$$

which can be satisfied by setting

$$\frac{1}{p} = 2\cdot\left(4k^{2/r}\right)^{1/(1-\theta)}.$$

Plugging-in these parameters, the total sample complexity is

$$r\cdot\left[O\left(\frac{(1+1/\theta)\left(4k^{2/r}\right)^{1/(1-\theta)}}{\varepsilon}\cdot\left(d+\log\left(\frac{2r}{\delta}\right)\right)\right) + k\cdot O\left(\frac{1+1/\theta}{\varepsilon}\log\left(\frac{2rk}{\delta}\right)\right)\right].$$

To conclude, we optimize the bound above by choosing $\theta = \frac{r}{2\log k}$. This choice ensures that $1+1/\theta = O(\log(k)/r)$. Since $r \leq \log k$, we also have $\theta \leq 1/2$, which implies $1/(1-\theta) = 1 + \theta/(1-\theta) \leq 1 + 2\theta$. Hence, $\left(4k^{2/r}\right)^{1/(1-\theta)} \leq \left(4k^{2/r}\right)\left(4k^{2/r}\right)^{2\theta} = O(k^{2/r})$. This yields the following sample complexity bound:

$$O\left(k^{2/r}\log(k)\frac{d}{\varepsilon} + \frac{k\log(k)}{\varepsilon}\log\left(\frac{k}{\delta}\right)\right).$$

$\square$

We sketch below alternative bounds that can be achieved by employing slightly more sophisticated variants of boosting such as boost-by-majority [SF12, Chapter 13] or recursive boosting [Sch90]. The general idea remains the same as in our application of AdaBoost, where we optimize a particular "margin" loss function defined on the $k$ distributions, but the bounds below can be favorable in the regime where $r$ is small (e.g., constant).

**Claim 14** (Recursive Boosting–Base Case). *For any $p > 0$ and any mixture $q = (w_1, \ldots, w_k)$, suppose that predictors $h_1, h_2, h_3$ satisfy:*

1. $\mathbf{Pr}_{j \sim q} [\mathrm{err}(h_1, D_j) > \tau] \leq p.$

2. $\mathbf{Pr}_{j \sim q_2} [\mathrm{err}(h_2, D_j) > \tau] \leq p$, *where* $q_2 = \frac{1}{2} q_C + \frac{1}{2} q_I$, $q_C$ *is $q$ conditioned on* $\{j \in [k] : \mathrm{err}(h_1, D_j) \leq \tau\}$ *and $q_I$ is $q$ conditioned on* $\{j \in [k] : \mathrm{err}(h_1, D_j) > \tau\}$.

3. $\mathbf{Pr}_{j \sim q_3} [\mathrm{err}(h_3, D_j) > \tau] \leq p$, *where $q_3$ is $q$ conditioned on*
$$\{j \in [k] : \mathbb{1}[\mathrm{err}(h_1, D_j) > \tau] \neq \mathbb{1}[\mathrm{err}(h_2, D_j) > \tau]\}.$$

*Then, the majority vote predictor $F$ obtained from $h_1$, $h_2$, and $h_3$ satisfies*
$$\mathbf{Pr}_{j \sim q} [\mathrm{err}(F, D_j) > 2\tau] \leq 3p^2 - 2p^3.$$

It follows from the claim above as a corollary, by choosing $p = 1/\sqrt{4k}$, that only 3 rounds of adaptive sampling suffice to achieve a sample complexity of $\widetilde{O}(d\sqrt{k}/\varepsilon)$ for realizable MDL.

More generally, by employing a variant of Boost-by-Majority [SF12, Chapter 13, Exercise 13.5], we get the following guarantee:

**Claim 15** (Boost-by-Majority). *For any target $\varepsilon$, for any number of rounds $r \geq 1$, for any $p \in (0, \frac{1}{2})$, and a suitably chosen $\theta \in (0, 1)$, running a modified version of Boost-by-Majority [SF12, Chapter 13, Exercise 13.5] for $r$ rounds making calls to a $\left(\tau := \frac{\varepsilon p}{1+1/\theta}, \delta\right)$-PAC-learner $\mathbb{A}$, outputs a predictor $F : x \mapsto \mathbb{1} \left[\frac{1}{r} \sum_{t=1}^{r} h_t(x) \geq \frac{1}{2}\right]$ that satisfies*
$$\frac{1}{k} \sum_{j=1}^{k} \mathbb{1} \left[\mathrm{err}(F, D_j) > \tau\right] \leq \mathrm{Binom}\left(r, \left(\frac{1+\theta}{2}\right) r, 1 - p\right).$$

To invoke the claim above, one would need to solve for $p$ and $\theta$ such that $\mathrm{Binom}\left(r, \left(\frac{1+\theta}{2}\right) r, 1 - p\right) < 1/k$. See [SF12, Chapter 13, Figure 13.3] for an illustrative comparison between AdaBoost and Boost-by-Majority in terms of performance based on number of rounds $r$.

## B  Lower Bound for Realizable MDL

In this section, we prove the following sample complexity lower bound against $r$-round MDL algorithms. The lower bound nearly matches the upper bound in Theorem 10, and holds even for MDL algorithms that only work in the realizable setting. For brevity, we treat the PAC parameters $\varepsilon$ and $\delta$ as sufficiently small constants (below $1/100$).

**Theorem 16** (Formal version of Theorem 2). *For $k \geq 1$, $r = O(\log k)$ and*
$$d \geq \max\{\Omega(k \log k), \Omega(k^{1-1/r} \log(k) \log(r))\},$$
*there exists a hypothesis class of VC dimension $d$ such that every $r$-round, $(0.01, 0.01)$-PAC algorithm for realizable MDL on $k$ distributions has a sample complexity of*
$$\Omega\left(\frac{dk^{1/r}}{r \log^2 k}\right).$$

In particular, to achieve a near-optimal sample complexity of $O((d + k) \mathrm{polylog}(k))$, the learning algorithm must have a round complexity of $\Omega\left(\frac{\log k}{\log \log k}\right)$.

**Remark 2.** *While the lower bound in Theorem 16 is stated for a constant accuracy parameter (i.e., $\epsilon = 0.01$) for brevity, it is easy to derive an $\Omega(\frac{1}{\epsilon} \cdot \frac{dk^{1/r}}{r \log^2 k})$ lower bound for general $\epsilon$ via a standard argument. We start with the construction for $\epsilon_0 = 0.01$. Then, we obtain a new MDL instance by "diluting" each data distribution by a factor of $100\epsilon$: we scale the probability mass on each example by a factor of $100\epsilon$, and put the remaining mass of $1 - 100\epsilon$ on a "trivial" example $(\perp, 0)$, where $\perp$ is a dummy instance that is labeled with 0 by every hypothesis. Learning this new instance up to error $\epsilon$ is equivalent to learning the original instance (before diluting) up to error $\epsilon/(100\epsilon) = 0.01 = \epsilon_0$. Intuitively, the example $(\perp, 0)$ provides no information, and the learning algorithm has to draw $\Omega(1/\epsilon)$ samples in expectation to see an informative example. This leads to a lower bound with an additional $1/\epsilon$ factor.*

## B.1 Intuition

We start by explaining the intuition behind the construction of the hard MDL instance. For simplicity, we start with the $r = 2$ case and sketch the proof of an $\Omega(d\sqrt{k})$ lower bound.

**Hard Instance against Two-Round Algorithms** We will consider the class of linear functions over $\mathcal{X} = \mathbb{F}_2^d$, namely, the hypothesis class

$$\mathcal{H}_d := \left\{ h_w : w \in \mathbb{F}_2^d \right\}, \text{ where } h_w : x \in \mathbb{F}_2^d \mapsto w^\top x \in \mathbb{F}_2.$$

The ground truth $h^\star$ is drawn uniformly at random from $\mathcal{H}_d$.

To construct the $k$ data distributions, we will randomly choose *difficulty levels* $\mathsf{diff}_1, \mathsf{diff}_2, \ldots, \mathsf{diff}_k$ such that $\sum_{i=1}^k \mathsf{diff}_i \le d$. (We will specify the distribution of $(\mathsf{diff}_i)_{i=1}^k$ later.) Then, we randomly choose subspaces $V_1, V_2, \ldots, V_k \subseteq \mathbb{F}_2^d$ such that: (1) $\dim(V_i) = \mathsf{diff}_i$; (2) $\dim(\mathrm{Span}(V_1 \cup V_2 \cup \cdots \cup V_k)) = \sum_{i=1}^k \mathsf{diff}_i$; (3) The $k$-tuple $(V_1, V_2, \ldots, V_k)$ is uniformly distributed over all possible choices that satisfy Constraints (1) and (2). Finally, each $D_i$ is set to the uniform distribution over $V_i$.

It remains to specify the choice of $\mathsf{diff}_1$ through $\mathsf{diff}_k$. Let $d_0$ and $k_0$ be integers to be chosen later. We will choose $\mathsf{diff}_1, \ldots, \mathsf{diff}_k$ to be a uniform permutation of: (1) 1 copy of $d_0$; (2) $\sqrt{k_0}$ copies of $d_0/\sqrt{k_0}$; (3) $\ge k_0$ copies of $d_0/k_0$. For this to be valid, we need

$$1 + \sqrt{k_0} + k_0 \le k \quad \text{and} \quad 1 \cdot d_0 + \sqrt{k_0} \cdot d_0/\sqrt{k_0} + k \cdot d_0/k_0 \le d,$$

both of which can be satisfied for some $d_0 = \Theta(d)$ and $k_0 = \Theta(k)$. In the following, we will drop the subscripts in $d_0$ and $k_0$ for brevity.

**Two-Round vs. Three-Round Algorithms** Given labeled examples $\{(x_i, y_i)\}_{i \in [m]}$, we can determine the value of $h^\star(x)$ for every $x \in \mathrm{Span}(\{x_1, x_2, \ldots, x_m\})$. Since the prior distribution of $h^\star$ is uniform over $\mathcal{H}_d$, for every $x \in \mathbb{F}_2^d \setminus \mathrm{Span}(\{x_1, x_2, \ldots, x_m\})$, the conditional distribution of $h^\star(x)$ is uniform. Intuitively, this means that, for every such instance $x$, we cannot predict its label better than random guessing. Therefore, to learn a classifier with error $\le 0.01$ on $D_i$, we must observe $\mathsf{diff}_i$ linearly independent instances from $V_i$. In particular, we must draw at least $\mathsf{diff}_i$ samples from $D_i$.

Suppose that the learner were allowed *three* rounds of sampling. The following is a natural algorithm:

- Draw $\approx d/k$ samples from each distribution. This allows us to identify the distributions with $\mathsf{diff}_i = d/k$ as well as to find an accurate classifier for each such distribution.
- From each of the $\sqrt{k} + 1$ remaining distributions, draw $\approx d/\sqrt{k}$ samples. This satisfies all distributions except the one with $\mathsf{diff}_i = d$.
- Finally, draw $\approx d$ samples from the only remaining distribution.

Note that each step draws $O(d)$ samples, so the total sample complexity is $O(d)$.

If only two rounds are allowed, the learner must "skip" one of the three steps above, and use either of the following two strategies, both of which lead to an $\Omega(d\sqrt{k})$ sample complexity:

- Strategy 1: In the first round, draw $\approx d/\sqrt{k}$ samples from each of the $k$ distribution to identify the only distribution with $\mathsf{diff}_i = d$. In the second round, draw $\approx d$ samples from the remaining distribution.
- Strategy 2: As before, draw $\approx d/k$ samples from each distribution to identify the distributions with $\mathsf{diff}_i \ge d/\sqrt{k}$. Then, there are still $\sqrt{k} + 1$ "suspects" among which one distribution has difficulty level $\mathsf{diff}_i = d$. The learner must draw $\approx d$ samples from each of them in the second round.

**Hard Instance against $r$-Round Algorithms** For the general case, we set $\alpha := k^{1/r}$. We extend the construction above such that $\mathsf{diff}_1$ through $\mathsf{diff}_k$ is a random permutation of

- $n_0 = k$ copies of $d_0 = d/(rk)$.
- $n_1 = k/\alpha$ copies of $d_1 = \alpha d/(rk)$.
- $n_2 = k/\alpha^2$ copies of $d_2 = \alpha^2 d/(rk)$.

- $\cdots$
- $n_r = k/\alpha^r = 1$ copy of $d_r = \alpha^r d/(rk) = d/r$.

As long as $r \leq \log_2 k$, we have $\alpha = k^{1/r} \geq 2$, which implies that $n_0, \ldots, n_r$ decreases geometrically. It follows that

$$n_0 + n_1 + \cdots + n_r \leq 2k \quad \text{and} \quad \sum_{i=0}^{r} n_i \cdot d_i = (r+1) \cdot (d/r) \leq 2d.$$

Thus, the above would give a valid instance after scaling every $n_i$ down by a factor of 2.

Again, an $(r+1)$-round learner would spend $O(n_{i-1} \cdot d_{i-1}) = O(d/r)$ samples in the $i$-th round to learn the distributions with difficulty levels $\leq d_{i-1}$. Then, there are only $O(n_i)$ remaining distributions, each of which has a difficulty level $\geq d_i$. The sample complexity is thus $(d/r) \cdot (r+1) = O(d)$.

When the learner is only allowed $r$ rounds of adaptive sampling, intuitively, the learner must "skip" one of the $r + 1$ rounds outlined above. Suppose that, for some $i \in [r]$, the learner decides to draw $\Theta(d_i)$ samples from each of the $\Theta(n_{i-1})$ remaining distributions, in the hope of learning the distributions with difficulty levels both $d_{i-1}$ and $d_i$. Note that the parameters $n_i$ and $d_i$ are chosen such that $n_{i-1} \cdot d_i = dk^{1/r}/r$, so the learner must incur a $k^{1/r}$ blowup in the sample complexity.

## B.2 Towards a Formal Proof

To formalize the intuition outlined above, we slightly modify the construction into a $k$-fold direct sum of a single-distribution version.

**Random MDL Instance $\mathcal{I}$**   Let $\alpha := k^{1/r}$. For some $d_0 \geq k$, let $D^{\mathsf{diff}}$ be the probability distribution over $[d_0]$ such that $D^{\mathsf{diff}}(d_0) = \frac{1}{k}$ and

$$D^{\mathsf{diff}}\left(\frac{d_0}{\alpha^i}\right) = \frac{\alpha^i - \alpha^{i-1}}{k}, \ \forall i \in [r].$$

Here and in the rest of this section, we abuse the notation $D^{\mathsf{diff}}$ for its probability mass function. We also assume for brevity that $d_0/\alpha^i$ is an integer; if not, the same proof would go through after proper rounding. Note that

$$\mathop{\mathbb{E}}_{\mathsf{diff} \sim D^{\mathsf{diff}}} [\mathsf{diff}] \leq \sum_{i=0}^{r} \frac{d_0}{\alpha^i} \cdot \frac{\alpha^i}{k} = (r+1) \cdot \frac{d_0}{k}.$$

Now we define a distribution over MDL instances with $k$ distributions.

**Definition 2** (Multi-distribution instance). Given $d_0, k$, and a sufficiently large $d := \Theta(d_0 \log k)$, we construct an MDL instance $\mathcal{I}$ as follows:

- Independently sample $\mathsf{diff}_1, \mathsf{diff}_2, \ldots, \mathsf{diff}_k \sim D^{\mathsf{diff}}$. Repeat until $\sum_{i=1}^{k} \mathsf{diff}_i \leq d$.

- Sample subspaces $V_1, V_2, \ldots, V_k \subseteq \mathbb{F}_2^d$ uniformly at random, subject to $\dim(V_i) = \mathsf{diff}_i$ and $\dim(\mathrm{Span}(V_1 \cup V_2 \cup \cdots \cup V_k)) = \sum_{i=1}^{k} \mathsf{diff}_i$.

- Let $\mathcal{I}$ be the MDL instance on the class $\mathcal{H}_d$ of linear functions over $\mathbb{F}_2^d$, where the $i$-th data distribution $D_i$ is the uniform distribution over $V_i$, and the ground truth classifier $h^\star \in \mathcal{H}_d$ is chosen uniformly at random.

**Random Single-Distribution Instance $\mathcal{I}'$**   We also consider a closely-related single-distribution version of the problem, where only one difficulty level diff is sampled from $D^{\mathsf{diff}}$.

**Definition 3** (Single-distribution instance). Given $d \geq d_0 \geq 1$ and distribution $D^{\mathsf{diff}}$ over $[d_0]$, we construct the single-distribution learning instance $\mathcal{I}'$ as follows:

- Sample $\mathsf{diff} \sim D^{\mathsf{diff}}$ and a uniformly random diff-dimensional subspace $V \subseteq \mathbb{F}_2^d$.

- Let $\mathcal{I}'$ be the realizable PAC learning instance on the class $\mathcal{H}_d$ of linear functions over $\mathbb{F}_2^d$, where the data distribution $D$ is the uniform distribution over $V$, and the ground truth classifier $h^\star \in \mathcal{H}_d$ is chosen uniformly at random.

**Roadmap** We prove Theorem 16 by combining the following two technical lemmas:

- Lemma 17: An $r$-round, $(0.01, 0.01)$-PAC MDL algorithm with sample complexity $M$ can be transformed into an $r$-round $(0.02, O(1/k))$-PAC algorithm for $\mathcal{I}'$ with sample complexity bound $O((M/k)\log k)$.

- Lemma 18: An $r$-round, $(0.02, O(1/k))$-PAC algorithm for $\mathcal{I}'$ must draw $\Omega(d_0 k^{1/r}/(rk))$ samples in expectation.

**Lemma 17.** *For any $d_0 \geq k \geq 1$ and $1 \leq r \leq O(\log k)$, if there is an $r$-round $(0.01, 0.01)$-PAC MDL algorithm $\mathcal{A}$ with sample complexity $M$ on $k$ distributions and hypothesis classes of VC-dimension $d = \Theta(d_0 \log k)$, there is another $r$-round algorithm $\mathcal{A}'$ such that:*

- *On a random single-distribution instance $\mathcal{I}'$ (Definition 3), $\mathcal{A}'$ learns an $0.02$-accurate classifier with probability at least $1 - \frac{1}{100k}$.*

- *On a random single-distribution instance $\mathcal{I}'$, $\mathcal{A}'$ takes $O((M/k)\log k)$ samples in expectation.*

*The probability and expectation above are over the randomness in instance $\mathcal{I}'$, the learning algorithm $\mathcal{A}'$, as well as the drawing of samples.*

**Lemma 18.** *Suppose that $k \geq 1$, $r \leq O(\log k)$ and $d_0 \geq \max\{k, \Omega(k^{1-1/r}\log r)\}$. If an $r$-round learning algorithm $\mathcal{A}$ outputs a $0.02$-accurate classifier with probability $\geq 1 - \frac{1}{100k}$ on a random instance $\mathcal{I}'$ with parameters $(d_0, k, r)$, $\mathcal{A}$ must take $\Omega(d_0 k^{1/r}/(rk))$ samples in expectation.*

*Proof of Theorem 16 assuming Lemmas 17 and 18.* Let $d_0 = \Theta(d/\log k)$. The assumption that $d \geq \max\{\Omega(k \log k), \Omega(k^{1-1/r}\log(k)\log(r))\}$ ensures that $d_0 \geq k$ and $d_0 \geq \Omega(k^{1-1/r}\log r)$. Suppose that $\mathcal{A}$ is an $r$-round, $(0.01, 0.01)$-PAC MDL algorithm with sample complexity $M$ for $k$ distributions and hypothesis classes of VC dimension $\leq d$. By Lemma 17, there is another algorithm $\mathcal{A}'$ that, on a random instance $\mathcal{I}'$ from Definition 3, returns a $0.02$-accurate classifier with probability at least $1 - \frac{1}{100k}$ and takes $O((M/k)\log k)$ samples in expectation. By Lemma 18, we have

$$O\left(\frac{M}{k}\cdot \log k\right) \geq \Omega\left(\frac{d_0 k^{1/r}}{rk}\right) = \Omega\left(\frac{dk^{1/r}}{rk \log k}\right).$$

It follows that $M \geq \Omega\left(\frac{dk^{1/r}}{r\log^2 k}\right)$. $\qquad\square$

## B.3 Proof of Lemma 17

To prove Lemma 17, it suffices to show that we can solve a random single-distribution instance $\mathcal{I}'$ (from Definition 3) by running an MDL algorithm $\mathcal{A}$ in a black-box way. Naturally, we plant $\mathcal{I}'$ into an MDL instance $\mathcal{I}$ with $k$ distributions (from Definition 2), solve instance $\mathcal{I}$ using $\mathcal{A}$, and use the output of $\mathcal{A}$ to solve the actual instance $\mathcal{I}'$. While the idea is simple, some care needs to be taken to carry out this plan correctly.

First, we draw $i^\star$ uniformly at random from $[k]$, and let the $i^\star$-th data distribution in $\mathcal{I}$ correspond to the distribution in $\mathcal{I}'$. For each $i \in [k] \setminus \{i^\star\}$, we draw $\mathrm{diff}_i$ from $D^{\mathrm{diff}}$ independently. Let $V_{i^\star}$ denote the subspace of $\mathbb{F}_2^d$ corresponding to the task $\mathcal{I}'$, where $d := \Theta(d_0 \log k)$ is the ambient dimension. Intuitively, for each $i \neq i^\star$, we want to sample a $\mathrm{diff}_i$-dimensional subspace $V_i$ to construct the MDL instance. This ensures that $(V_1, V_2, \ldots, V_k)$ form a random MDL instance (from Definition 2), in which the $k$ distributions are symmetric, so that the identity of $i^\star$ would not be revealed. Here, an obstacle is that we cannot construct $V_i$ for $i \neq i^\star$ without knowing the subspace $V_{i^\star}$. In particular, knowing $V_{i^\star}$ or just the value of $\dim(V_{i^\star})$ would make the single-distribution version too easy.

Our workaround is to slightly modify the definition of the single-distribution task. We allow the algorithm for the single-distribution instance to specify an integer $m_0 \geq 0$ at the beginning. Then, the algorithm receives $m_0$ vectors in $\mathbb{F}_2^d$ chosen uniformly at random subject to that, along with any basis of $V_{i^\star}$, these $m_0 + \dim(V_{i^\star})$ vectors are linearly independent. Intuitively, the algorithm is provided with information about the "complement" of $V_{i^\star}$ in $\mathbb{F}_2^d$, which allows it to construct the $k-1$ fictitious subspaces $\{V_i\}_{i\neq i^\star}$ and thus an MDL instance on $k$ distributions. If $m_0 > d - \mathrm{diff}_{i^\star} = d - \dim(V_{i^\star})$, the learner fails the learning task immediately. Note that we cannot simply give all the $d - \mathrm{diff}_{i^\star}$ vectors to the learner directly—otherwise the learner would know $\mathrm{diff}_{i^\star}$, which makes the single-distribution instance easy.

*Proof of Lemma 17.* Let $\mathcal{I}'$ be a single-distribution instance generated randomly with parameters $k$ and $d_0$ according to Definition 3. Recall that the hypothesis class is $\mathcal{H}_d$, where $d = \Theta(d_0 \log k)$ is the ambient dimension. The data distribution $D$ is uniform over an unknown subspace $V \subseteq \mathbb{F}_2^d$ with an unknown dimension $\mathsf{diff} \sim D^{\mathsf{diff}}$.

We consider the following procedure for solving $\mathcal{I}'$ using the given algorithm $\mathcal{A}$:

- Draw $i^\star$ from $[k]$ uniformly at random. For each $i \in [k] \setminus \{i^\star\}$, draw $\mathsf{diff}_i$ independently from $D^{\mathsf{diff}}$.

- Set $m_0 := \sum_{i \in [k] \setminus \{i^\star\}} \mathsf{diff}_i$. Request $m_0$ random vectors in $\mathbb{F}_2^d$ that are linearly independent of the unknown subspace $V$. In this step, the algorithm might fail due to $m_0 > d - \mathsf{diff}_{i^\star}$, where $\mathsf{diff}_{i^\star}$ is the unknown difficulty level of the single-distribution instance $\mathcal{I}'$.

- Use these vectors to construct a $\mathsf{diff}_i$-dimensional subspace $V_i$ for each $i \in [k] \setminus \{i^\star\}$. Let $D_i$ be the uniform distribution over $V_i$.

- Simulate the MDL algorithm $\mathcal{A}$ on distributions $D_1, D_2, \dots, D_k$, where $D_{i^\star}$ is the alias of the data distribution $D$ in instance $\mathcal{I}'$. In addition, when $\mathcal{A}$ draws the first round of samples, we draw $\Theta(\log k)$ additional samples from $D_{i^\star}$ to form a validation dataset.

- When $\mathcal{A}$ terminates and outputs a classifier $\hat{h}$, check whether $\hat{h}$ has an error $\leq \frac{0.01+0.02}{2}$ on the validation dataset. If so, output $\hat{h}$ as the answer; otherwise, report "failure".

Let $\mathcal{A}^{\mathsf{sim}}$ denote the algorithm defined above. In the following, we will show that: (1) $\mathcal{A}^{\mathsf{sim}}$ *can* be implemented using $\mathcal{A}$ as a black box; (2) $\mathcal{A}^{\mathsf{sim}}$ solves instance $\mathcal{I}'$ with a good probability; (3) $\mathcal{A}^{\mathsf{sim}}$ does not draw too many samples from $D$.

**Details of the Simulation** When $\mathcal{A}^{\mathsf{sim}}$ simulates algorithm $\mathcal{A}$, it maintains a set $S_i \subseteq \mathbb{F}_2^d \times \mathbb{F}_2$ for each $i \neq i^\star$, which is empty at the beginning. Whenever $\mathcal{A}$ requests samples from $D_{i^\star}$, $\mathcal{A}^{\mathsf{sim}}$ draws from $D$ and forwards the labeled examples to $\mathcal{A}$. When $\mathcal{A}$ requests a sample from $D_i$ for some $i \neq i^\star$, $\mathcal{A}^{\mathsf{sim}}$ first draws $x \sim D_i$. If $x$ lies in $\mathrm{Span}(\{x \in \mathbb{F}_2^d : (x, y) \in S_i, \exists y \in \mathbb{F}_2\})$, $\mathcal{A}^{\mathsf{sim}}$ computes the unique label $y \in \mathbb{F}_2$ such that $(x, y)$ remains consistent with the labeled examples in $S_i$; otherwise, $\mathcal{A}^{\mathsf{sim}}$ draws a random $y \sim \mathrm{Bernoulli}(1/2)$. The labeled example $(x, y)$ is forwarded to $\mathcal{A}$, and then added to the dataset $S_i$. By doing so, we defer the randomness in the ground truth classifier $h^\star$, namely, we realize one bit of information of $h^\star$ whenever this information is needed to determine the label of an example.

**Equivalence** To analyze the performance of $\mathcal{A}^{\mathsf{sim}}$, the key observation is the following: Conditioning on that $\mathcal{A}^{\mathsf{sim}}$ does not fail when requesting the $m_0$ vectors, from the perspective of the simulated copy of $\mathcal{A}$, it is running on a random MDL instance $\mathcal{I}$ from Definition 2. This is because conditioning on that $\mathcal{A}^{\mathsf{sim}}$ does not fail has the same effect as the conditioning on $\sum_{i=1}^k \mathsf{diff}_i \leq d$ in Definition 2. Furthermore, both the marginals of $D_1$ through $D_k$ on $\mathbb{F}_2^d$ and the choice of the labels are identical to those in the definition of $\mathcal{I}$.

**Boost the Success Probability** Since $\mathcal{A}$ is $(0.01, 0.01)$-PAC, it holds with probability $\geq 0.99$ that its output $\hat{h}$ is a $0.01$-accurate classifier for each $D_i$. In particular, $\hat{h}$ is $0.01$-accurate for distribution $D = D_{i^\star}$, and is thus a valid answer for instance $\mathcal{I}'$. However, this only guarantees a success probability of $0.99$, falling short of the desired $1 - 1/(100k)$.

Fortunately, it suffices to run $l = \Theta(\log k)$ independent copies of algorithm $\mathcal{A}'$ to boost the success probability. Note that these $l$ copies must be simulated in parallel, so that the resulting algorithm still takes $r$ rounds of samples. Furthermore, we share the vectors requested at the beginning of different copies of $\mathcal{A}'$, so that the number of vectors that we actually request, $m_0$, is the maximum realization of $\sum_{i \neq i^\star} \mathsf{diff}_i$ over the $l$ simulations. Whenever one of the $l$ simulations outputs a classifier $\hat{h}$, we test it on the size-$\Theta(\log k)$ validation dataset and verify whether its empirical error is below $\frac{0.01+0.02}{2}$. We output $\hat{h}$ as the answer to $\mathcal{I}'$ only if it passes the test. Let $\mathcal{A}'$ denote the above algorithm.

**Correctness of $\mathcal{A}'$** Now, we analyze the probability that $\mathcal{A}'$ fails to output a $0.01$-accurate classifier for $\mathcal{I}'$. There are three possible reasons:

- **Reason 1:** $m_0 > d - \text{diff}_{i^\star}$. This happens when $\sum_{i=1}^k \text{diff}_i > d$ holds in one of the $l$ copies of $\mathcal{A}^{\text{sim}}$. Note that in each fixed copy of $\mathcal{A}^{\text{sim}}$, $\text{diff}_1, \text{diff}_2, \ldots, \text{diff}_k$ independently follow $D^{\text{diff}}$. Recall that $\mathbb{E}_{\text{diff} \sim D^{\text{diff}}}[\text{diff}] = O(r d_0/k)$ and $d = \Theta(d_0 \log k)$. It follows from a multiplicative Chernoff bound that $\sum_{i=1}^k \text{diff}_i > d$ happens with probability $1/\text{poly}(k)$, which is still $\ll 1/k$ after a union bound over the $l = \Theta(\log k)$ copies.

  In more detail, let random variable $X_i$ denote the value of $\text{diff}_i/d_0$. Since $D^{\text{diff}}$ is supported over $[d_0]$, $X_i \in [0, 1]$. Furthermore, $\mathbb{E}[X_i] = \mathbb{E}_{\text{diff} \sim D^{\text{diff}}}[\text{diff}/d_0] = O(r d_0/k)/d_0 = O(r/k)$. Then, $\sum_{i=1}^k X_i$ is the sum of $k$ random variables in $[0, 1]$ and has an expectation of $\mu := k \cdot O(r/k) = O(r)$.

  Assuming that $r = O(\log k)$ holds with a sufficiently small constant factor, we have $\mu = O(r) \leq \ln k$. Let $\delta = (5 \ln k)/\mu - 1 \geq 4$. Also recall that $d = \Theta(d_0 \log k)$, so we may assume $d/d_0 \geq 5 \ln k$. Then, we have

$$\mathbf{Pr}\left[\sum_{i=1}^k \text{diff}_i > d\right] = \mathbf{Pr}\left[\sum_{i=1}^k X_i > d/d_0\right] \leq \mathbf{Pr}\left[\sum_{i=1}^k X_i > 5 \ln k\right] = \mathbf{Pr}\left[\sum_{i=1}^k X_i > (1 + \delta)\mu\right].$$

  A multiplicative Chernoff bound gives

$$\mathbf{Pr}\left[\sum_{i=1}^k X_i > (1 + \delta)\mu\right] \leq \exp\left(-\frac{\delta^2 \mu}{2 + \delta}\right) \leq \exp\left(-\frac{2\delta\mu}{3}\right) \leq \exp\left(-\frac{8 \ln k}{3}\right) \ll \frac{1}{k},$$

  where the second step applies $\delta \geq 4$ and the third step applies $\delta\mu = 5 \ln k - \mu \geq 4 \ln k$.

- **Reason 2: None of the $l$ simulations succeeds.** Note that conditioning on the realization of instance $\mathcal{I}'$, the $l$ copies of the simulation are independent. Furthermore, since $\mathcal{A}$ is $(0.01, 0.01)$-PAC, each copy fails with probability $\leq 0.01$. The probability for all $l = \Theta(\log k)$ copies to fail is thus at most $0.01^l \ll 1/k$.

- **Reason 3: The validation procedure fails.** We need to validate at most $l = O(\log k)$ classifiers that are generated independently of the validation set. For each classifier $\hat{h}$, we need to distinguish the two cases $\text{err}(\hat{h}, D) \leq 0.01$ and $\text{err}(\hat{h}, D) > 0.02$. By a Chernoff bound and the union bound, the validation set of size $\Theta(\log k)$ is sufficient for upper bounding the failure probability by $l \cdot e^{-\Omega(\log k)} \leq 1/\text{poly}(k) \ll 1/k$.

Combining the three cases above, for all sufficiently large $k$, the total failure probability is smaller than $\frac{1}{100k}$. In other words, algorithm $\mathcal{A}'$ outputs a $0.01$-accurate classifier with probability $1 - \frac{1}{100k}$ on a random single-distribution instance $\mathcal{I}'$.

**Sample Complexity** Each of the $l = O(\log k)$ simulated copies of $\mathcal{A}$ draws at most $M$ samples from $D_1$ through $D_k$ in total, as each simulated copy of $\mathcal{A}$ effectively runs on a valid MDL instance. Furthermore, from the perspective of each simulated copy of $\mathcal{A}$, the $k$ distributions $D_1, \ldots, D_k$ are generated symmetrically, so the expected number of actual samples drawn from $D_{i^\star} = D$ is at most $M/k$. Therefore, the $l$ copies of $\mathcal{A}^{\text{sim}}$ together draw $(M/k) \cdot l = O((M/k) \log k)$ samples in expectation. Note that this dominates the $O(\log k)$ samples drawn for the purpose of validation, so the overall sample complexity of $\mathcal{A}'$ is $O((M/k) \log k)$. $\qquad\square$

### B.4 Proof of Lemma 18

It remains to prove a sample complexity lower bound for solving the single-distribution instance $\mathcal{I}'$ from Definition 3 within $r$ rounds. Recall that $\mathcal{I}'$ is defined as the task of learning linear functions over the hypercube of dimension $d = \Theta(d_0 \log k)$, and the data distribution is uniform over a random subspace of dimension $\text{diff} \sim D^{\text{diff}}$. Also recall that $\alpha = k^{1/r} \geq 2$, $D^{\text{diff}}(d_0) = \frac{1}{k}$ and $D^{\text{diff}}(d_0/\alpha^i) = (\alpha^i - \alpha^{i-1})/k$ for every $i \in [r]$.

*Proof of Lemma 18.* Suppose towards a contradiction that there exists an $r$-round learning algorithm $\mathcal{A}$ for the single-distribution instance $\mathcal{I}'$ defined in Definition 3, such that it outputs a 0.02-accurate classifier with probability $1 - \frac{1}{100k}$ and takes at most $\frac{\alpha d_0}{100rk}$ samples in expectation.

**Typical Events** Consider the execution of $\mathcal{A}$ on the random instance $\mathcal{I}'$. We introduce $2r + 1$ "typical events", which will be shown to happen simultaneously with high probability:

- Event $\mathcal{E}_0^{\mathsf{num}}$: Let random variable $M_0$ denote the value of $m_0$, i.e., the number of linearly independent vectors requested by $\mathcal{A}$ at the beginning. $\mathcal{E}_0^{\mathsf{num}}$ is defined as the event that $M_0 \leq d - d_0$.

- Events $\mathcal{E}_1^{\mathsf{num}}, \mathcal{E}_2^{\mathsf{num}}, \ldots, \mathcal{E}_r^{\mathsf{num}}$: For each $i \in [r]$, let random variable $M_i$ denote the number of samples that $\mathcal{A}$ draws in the $i$-th round. $\mathcal{E}_i^{\mathsf{num}}$ is defined as the event that $M_i \leq \frac{\alpha^i d_0}{4k}$.

- Events $\mathcal{E}_1^{\mathsf{ind}}, \mathcal{E}_2^{\mathsf{ind}}, \ldots, \mathcal{E}_r^{\mathsf{ind}}$: For each $i \in [r]$, $\mathcal{E}_i^{\mathsf{ind}}$ is defined as the event that, among the labeled examples that $\mathcal{A}$ draws in the first $i$ rounds, all the instances (namely, the vectors in $\mathbb{F}_2^d$) are linearly independent.

For $i \in \{0, 1, \ldots, r\}$, we use the shorthands
$$\mathcal{E}_{\leq i}^{\mathsf{num}} := \mathcal{E}_0^{\mathsf{num}} \cap \mathcal{E}_1^{\mathsf{num}} \cap \cdots \mathcal{E}_i^{\mathsf{num}} \quad \text{and} \quad \mathcal{E}_{\leq i}^{\mathsf{ind}} := \mathcal{E}_1^{\mathsf{ind}} \cap \mathcal{E}_2^{\mathsf{ind}} \cap \cdots \mathcal{E}_i^{\mathsf{ind}}.$$
Moreover, we shorthand $\mathcal{E}_i^{\mathsf{diff}}$ for the event diff $= \frac{\alpha^i d_0}{k}$.

**Induction Hypothesis** We will prove the following statement by induction on $i$: For every $i \in \{0, 1, \ldots, r\}$ and every $i^\star \in \{i, i+1, \ldots, r\}$, it holds that
$$\mathbf{Pr}\left[\mathcal{E}_{\leq i}^{\mathsf{num}} \cap \mathcal{E}_{\leq i}^{\mathsf{ind}} \mid \mathcal{E}_{i^\star}^{\mathsf{diff}}\right] \geq 0.99 - \frac{4i}{25r} \geq \frac{1}{2}. \tag{7}$$

We first show that the above leads to a contradiction and thus proves the lemma. When $i = i^\star = r$, Equation (7) reduces to
$$\mathbf{Pr}\left[\mathcal{E}_{\leq r}^{\mathsf{num}} \cap \mathcal{E}_{\leq r}^{\mathsf{ind}} \mid \mathsf{diff} = d_0\right] \geq \frac{1}{2}.$$
Recall that $\alpha = k^{1/r} \geq 2$. Thus, with probability $\geq 1/2$ conditioning on diff $= d_0$, $\mathcal{A}$ draws
$$\sum_{i=1}^{r} M_i \leq \frac{\alpha d_0}{4k} + \frac{\alpha^2 d_0}{4k} + \cdots + \frac{\alpha^r d_0}{4k} \leq 2 \cdot \frac{\alpha^r d_0}{4k} = \frac{d_0}{2}$$
samples in total. Then, over the remaining randomness in the ground truth classifier $h^\star \in \mathcal{H}_d$, the correct label of every instance outside the span of the observed instances is still uniformly distributed over $\{0, 1\}$. In particular, regardless of how $\mathcal{A}$ outputs a classifier, the error of the classifier is at least $(1 - 2^{\lfloor d_0/2 \rfloor - d_0})/2$ in expectation. By Markov's inequality, the conditional probability of outputting a classifier with error $\leq 0.02$ is at most
$$\frac{1 - (1 - 2^{\lfloor d_0/2 \rfloor - d_0})/2}{1 - 0.02} < \frac{4}{5}.$$
Therefore, the probability that $\mathcal{A}$ fails to output a classifier with error $\leq 0.02$ on a random single-distribution instance $\mathcal{I}'$ is at least
$$\mathbf{Pr}_{\mathsf{diff} \sim D^{\mathsf{diff}}}\left[\mathsf{diff} = d_0\right] \cdot \mathbf{Pr}\left[\mathcal{E}_{\leq r}^{\mathsf{num}} \cap \mathcal{E}_{\leq r}^{\mathsf{ind}} \mid \mathsf{diff} = d_0\right] \cdot \frac{1}{5} \geq \frac{1}{k} \cdot \frac{1}{2} \cdot \frac{1}{5} > \frac{1}{100k},$$
contradicting the assumption on $\mathcal{A}$.

The remainder of the proof will establish Equation (7) by induction.

**Base Case** We start by verifying the base case that $i = 0$, namely,
$$\mathbf{Pr}\left[M_0 \leq d - d_0 \mid \mathcal{E}_{i^\star}^{\mathsf{diff}}\right] \geq 0.99, \ \forall i^\star \in \{0, 1, \ldots, r\}.$$
Note that $\mathcal{A}$ chooses $M_0$ before drawing any samples, so $M_0$ is independent of diff. Suppose towards a contradiction that $\mathbf{Pr}\left[M_0 > d - d_0\right] > 0.01$. Then, we have
$$\mathbf{Pr}\left[M_0 > d - d_0 \wedge \mathsf{diff} = d_0\right] = \mathbf{Pr}\left[M_0 > d - d_0\right] \cdot \mathbf{Pr}_{\mathsf{diff} \sim D^{\mathsf{diff}}}\left[\mathsf{diff} = d_0\right] > 0.01 \cdot \frac{1}{k} = \frac{1}{100k}.$$
Moreover, when both $M_0 > d - d_0$ and diff $= d_0$ hold, $\mathcal{A}$ would fail due to $M_0 > d - \mathsf{diff}$. This shows that the failure probability of $\mathcal{A}$ is strictly higher than $\frac{1}{100k}$, a contradiction.

**Inductive Step: Event $\mathcal{E}_i^{\mathsf{num}}$**  Fix $i \in [r]$ and $i^\star \in \{i, i+1, \ldots, r\}$. Suppose that the induction hypothesis (Equation (7)) holds for $i - 1$: shorthanding $\mathcal{E}^{\mathsf{good}} := \mathcal{E}_{\leq i-1}^{\mathsf{num}} \cap \mathcal{E}_{\leq i-1}^{\mathsf{ind}}$, it holds for every $j \in \{i-1, i, \ldots, r\}$ that

$$\mathbf{Pr}\left[\mathcal{E}^{\mathsf{good}} \mid \mathcal{E}_j^{\mathsf{diff}}\right] \geq 0.99 - \frac{4(i-1)}{25r} \geq \frac{1}{2}.$$

In the following, we condition on $\mathcal{E}^{\mathsf{good}}$ and $\mathcal{E}_{i^\star}^{\mathsf{diff}}$ and aim to lower bound the conditional probabilities of events $\mathcal{E}_i^{\mathsf{num}}$ and $\mathcal{E}_i^{\mathsf{ind}}$.

We observe that, conditioning on $\mathcal{E}^{\mathsf{good}}$, the labeled examples that $\mathcal{A}$ draws in the first $i-1$ rounds are conditionally independent of diff. In particular, the conditional distribution of $M_i \mid \mathcal{E}^{\mathsf{good}}$ is identical to that of $M_i \mid (\mathcal{E}^{\mathsf{good}} \cap \mathcal{E}_{i^\star}^{\mathsf{diff}})$. Then, we note that $\mathbb{E}\left[M_i \mid \mathcal{E}^{\mathsf{good}}\right] \leq \frac{\alpha^i d_0}{50rk}$ must hold; otherwise, we have

$$\mathbb{E}\left[M_i\right] \geq \mathbb{E}\left[M_i \mid \mathcal{E}^{\mathsf{good}}\right] \cdot \mathbf{Pr}\left[\mathcal{E}^{\mathsf{good}}\right] > \frac{\alpha^i d_0}{50rk} \cdot \mathbf{Pr}\left[\mathcal{E}^{\mathsf{good}}\right].$$

Furthermore, the induction hypothesis implies

$$\mathbf{Pr}\left[\mathcal{E}^{\mathsf{good}}\right] \geq \sum_{j=i-1}^{r} \mathbf{Pr}\left[\mathcal{E}_j^{\mathsf{diff}}\right] \cdot \mathbf{Pr}\left[\mathcal{E}^{\mathsf{good}} \mid \mathcal{E}_j^{\mathsf{diff}}\right]$$

$$\geq \sum_{j=i-1}^{r} \mathbf{Pr}\left[\mathsf{diff} = \frac{\alpha^j d_0}{k}\right] \cdot \frac{1}{2}$$

$$= \frac{1}{2}\mathbf{Pr}\left[\mathsf{diff} \geq \frac{\alpha^{i-1} d_0}{k}\right] = \frac{1}{2\alpha^{i-1}}.$$

It would then follow that

$$\mathbb{E}\left[M_i\right] > \frac{\alpha^i d_0}{50rk} \cdot \frac{1}{2\alpha^{i-1}} = \frac{\alpha d_0}{100rk},$$

contradicting the assumption that $\mathcal{A}$ has an expected sample complexity of at most $\frac{\alpha d_0}{100rk}$.

Thus, we have $\mathbb{E}\left[M_i \mid \mathcal{E}^{\mathsf{good}} \cap \mathcal{E}_{i^\star}^{\mathsf{diff}}\right] = \mathbb{E}\left[M_i \mid \mathcal{E}^{\mathsf{good}}\right] \leq \frac{\alpha^i d_0}{50rk}$. Markov's inequality then gives

$$\mathbf{Pr}\left[M_i > \frac{\alpha^i d_0}{4k} \mid \mathcal{E}^{\mathsf{good}} \cap \mathcal{E}_{i^\star}^{\mathsf{diff}}\right] \leq \frac{\mathbb{E}\left[M_i \mid \mathcal{E}^{\mathsf{good}} \cap \mathcal{E}_{i^\star}^{\mathsf{diff}}\right]}{\alpha^i d_0/(4k)} \leq \frac{\alpha^i d_0/(50rk)}{\alpha^i d_0/(4k)} = \frac{2}{25r}.$$

Therefore, we have

$$\mathbf{Pr}\left[\mathcal{E}_i^{\mathsf{num}} \cap \mathcal{E}^{\mathsf{good}} \mid \mathcal{E}_{i^\star}^{\mathsf{diff}}\right] \geq \mathbf{Pr}\left[\mathcal{E}^{\mathsf{good}} \mid \mathcal{E}_{i^\star}^{\mathsf{diff}}\right] \cdot \mathbf{Pr}\left[\mathcal{E}_i^{\mathsf{num}} \mid \mathcal{E}^{\mathsf{good}} \cap \mathcal{E}_{i^\star}^{\mathsf{diff}}\right]$$

$$\geq \left[0.99 - \frac{4(i-1)}{25r}\right] \cdot \left(1 - \frac{2}{25r}\right) \geq 0.99 - \frac{4i - 2}{25r}.$$

**Inductive Step: Event $\mathcal{E}_i^{\mathsf{ind}}$**  Now, we condition on event $\mathcal{E}_i^{\mathsf{num}} \cap \mathcal{E}^{\mathsf{good}} \cap \mathcal{E}_{i^\star}^{\mathsf{diff}}$ and aim to lower bound the conditional probability of $\mathcal{E}_i^{\mathsf{ind}}$. After the conditioning, it holds that

$$M_1 + M_2 + \cdots + M_i \leq \frac{\alpha d_0}{4k} + \frac{\alpha^2 d_0}{4k} + \cdots + \frac{\alpha^i d_0}{4k} \leq 2 \cdot \frac{\alpha^i d_0}{4k} \leq \frac{\alpha^{i^\star} d_0}{2k} = \mathsf{diff}/2.$$

Regardless of the $N := M_1 + M_2 + \cdots + M_{i-1}$ samples that $\mathcal{A}$ draws in the first $i - 1$ rounds, the probability that the next instance falls into the subspace spanned by those $N$ instances is at most $2^{N-\mathsf{diff}}$. By the same argument, for every $i \in \{0, 1, \ldots, M_i - 1\}$, after $i$ samples have been drawn in the $i$-th round, the next instance is outside the span of the first $N + i$ instances except with probability $2^{(N+i)-\mathsf{diff}}$. By the union bound, the $M_i$ additional instances in the $i$-th round are linearly independent together with the previous $N$ instances, except with probability

$$2^{N-\mathsf{diff}} + 2^{(N+1)-\mathsf{diff}} + \cdots + 2^{(N+M_i-1)-\mathsf{diff}} \leq 2^{N+M_i-\mathsf{diff}} \leq 2^{\mathsf{diff}/2-\mathsf{diff}} = 2^{-\mathsf{diff}/2}.$$

Recall that diff $= \frac{\alpha^{i^\star} d_0}{k} \geq \frac{\alpha d_0}{k} = \frac{d_0}{k^{1-1/r}}$. Therefore, for all sufficiently large $d_0 = \Omega(k^{1-1/r} \log r)$, it holds that $2^{-\text{diff}/2} \leq \frac{2}{25r}$. It then follows that

$$\mathbf{Pr}\left[\mathcal{E}_i^{\text{ind}} \mid \mathcal{E}_i^{\text{num}} \cap \mathcal{E}^{\text{good}} \cap \mathcal{E}_{i^\star}^{\text{diff}}\right] \geq 1 - \frac{2}{25r},$$

which further implies

$$\mathbf{Pr}\left[\mathcal{E}_{\leq i}^{\text{num}} \cap \mathcal{E}_{\leq i}^{\text{ind}} \mid \mathcal{E}_{i^\star}^{\text{diff}}\right] \geq \mathbf{Pr}\left[\mathcal{E}_i^{\text{num}} \cap \mathcal{E}^{\text{good}} \mid \mathcal{E}_{i^\star}^{\text{diff}}\right] \cdot \mathbf{Pr}\left[\mathcal{E}_i^{\text{ind}} \mid \mathcal{E}_i^{\text{num}} \cap \mathcal{E}^{\text{good}} \cap \mathcal{E}_{i^\star}^{\text{diff}}\right]$$

$$\geq \left(0.99 - \frac{4i-2}{25r}\right) \cdot \left(1 - \frac{2}{25r}\right) \geq 0.99 - \frac{4i}{25r}.$$

This completes the inductive step and concludes the proof. $\qquad \square$

# C  Upper Bounds for OODS

We analyze the correctness and the round complexity of the LazyHedge algorithm (Algorithm 3). We then show how the ideas in both the algorithm and its analysis can be easily transferred to the agnostic MDL setting, resulting in an algorithm with $\widetilde{O}(\sqrt{k})$ round complexity and near-optimal sample complexity.

---

**Algorithm 3:** LazyHedge: Hedge with Lazy Updates

---

**Input:** Number of dimensions $k$, number of iterations $T = \Theta\left(\frac{\log k}{\varepsilon^2}\right)$, step size $\eta = \Theta(\varepsilon)$,
        margin parameter $C > 1$, and access to a first-order oracle of $f$.

1 Set $w^{(1)} = (1/k, 1/k, \ldots, 1/k)$ and $\overline{w}^{(0)} = (0, 0, \ldots, 0)$.
2 **for** $t = 1, 2, \ldots, T$ **do**
3     **if** $w^{(t)} \in \mathcal{O}(\overline{w}^{(t-1)})$ **then**
4         Set $\overline{w}^{(t)} = \overline{w}^{(t-1)}$.
5     **else**
6         Set $\overline{w}_i^{(t)} = C \cdot \max\{w_i^{(1)}, w_i^{(2)}, \ldots, w_i^{(t)}\}$ for every $i \in [k]$.
7         Start a new OODS round with cap $\overline{w}^{(t)}$.

8     Query the first-order oracle to obtain supergradient $r^{(t)} \in \nabla f(w^{(t)})$.
9     Compute $w^{(t+1)} \in \Delta^{k-1}$ such that for every $i \in [k]$,

$$w_i^{(t+1)} = \frac{w_i^{(t)} \cdot e^{\eta r_i^{(t)}}}{\sum_{j=1}^{k} w_j^{(t)} \cdot e^{\eta r_j^{(t)}}}.$$

**Output:** Average weight vector $\frac{1}{T}\sum_{t=1}^{T} w^{(t)}$.

---

## C.1  Proof of Lemma 5

Recall that Lemma 5 states that LazyHedge finds an $O(\varepsilon)$-approximate maximizer of $f$ on $\Delta^{k-1}$.

*Proof of Lemma 5.* The standard regret analysis of the Hedge algorithm (e.g., [MRT18, Theorem 8.6]) gives

$$\sum_{t=1}^{T}\langle r^{(t)}, w^{(t)}\rangle \geq \max_{i \in [k]} \sum_{t=1}^{T}\langle r^{(t)}, e_i\rangle - R,$$

where $R = \frac{\ln k}{\eta} + \frac{\eta T}{8} = O\left(\frac{\log k}{\varepsilon}\right)$. Let $w^* \in \Delta^{k-1}$ be a maximizer of $f$ over $\Delta^{k-1}$. Then,

$$\sum_{t=1}^{T}\langle r^{(t)}, w^*\rangle \leq \max_{i \in [k]} \sum_{t=1}^{T}\langle r^{(t)}, e_i\rangle \leq \sum_{t=1}^{T}\langle r^{(t)}, w^{(t)}\rangle + R.$$

Rearranging gives $\sum_{t=1}^{T} \langle r^{(t)}, w^* - w^{(t)} \rangle \leq R$. Since $f$ is concave and $r^{(t)} \in \nabla f(w^{(t)})$ for every $t \in [T]$, we have $f(w^*) \leq f(w^{(t)}) + \langle r^{(t)}, w^* - w^{(t)} \rangle$. It follows that

$$\sum_{t=1}^{T} [f(w^*) - f(w^{(t)})] \leq \sum_{t=1}^{T} \langle r^{(t)}, w^* - w^{(t)} \rangle \leq R.$$

Dividing both sides by $T$ gives

$$f(w^*) - \frac{1}{T} \sum_{t=1}^{T} f(w^{(t)}) \leq \frac{R}{T} = O(\varepsilon).$$

Finally, applying the concavity of $f$ again gives

$$f(\hat{w}) = f\left( \frac{1}{T} \sum_{t=1}^{T} w^{(t)} \right) \geq \frac{1}{T} \sum_{t=1}^{T} f(w^{(t)}) \geq f(w^*) - O(\varepsilon) = \max_{w \in \Delta^{k-1}} f(w) - O(\varepsilon).$$

$\square$

## C.2 Upper Bound for the Box Setting

We formally prove Lemma 7, which states an upper bound of $\min\left\{ k, O((k/C) \cdot \log^8(k/\varepsilon)) \right\} \cdot O(\log_C k)$ on the round complexity of LazyHedge in the box setting. The proof proceeds in the following three steps. First, whenever the cap is updated in LazyHedge, we find an index $i$ that leads to this update and call it the *culprit* of this cap update. Then, we show that every an index can become the culprit of at most $O(\log_C k)$ cap updates. Finally, we control the number of indices that become the culprit at least once by $\min\{k, O((k/C) \cdot \log^8(k/\varepsilon))\}$, so the lemma immediately follows.

*Proof of Lemma 7.* If the cap is updated in the $t$-th iteration of LazyHedge, i.e., $\overline{w}^{(t)} \neq \overline{w}^{(t-1)}$, by definition of Algorithm 3, it must hold that $w^{(t)} \notin \mathcal{O}(\overline{w}^{(t-1)})$. By definition of the box setting, there exists $i \in [k]$ such that $w_i^{(t)} > \overline{w}_i^{(t-1)}$. We call the smallest such index $i$ the *culprit* of the cap update in iteration $t$. For each $i \in [k]$, we define

$$\mathcal{T}_i := \{t \in [T] : i \text{ is the culprit of the cap update in iteration } t\}.$$

Then, the round complexity of LazyHedge is exactly $\sum_{i=1}^{k} |\mathcal{T}_i|$.

**Every $\mathcal{T}_i$ is Small** We show that $|\mathcal{T}_i| \leq O(\log_C k)$ holds for every $i \in [k]$. Fix $i \in [k]$ and let $t_1 < t_2 < \cdots < t_m$ be the $m = |\mathcal{T}_i|$ elements of $\mathcal{T}_i$ in increasing order. For each $j \in [m]$, let $a_j := \max_{1 \leq t \leq t_j} w_i^{(t)}$ denote the maximum weight on the $i$-th coordinate among all weight vectors up to time $t_j$. Fix $j \in \{2, 3, \ldots, m\}$. Since $i$ is the culprit at time $t_j$, it holds that $a_j \geq w_i^{(t_j)} > \overline{w}_i^{(t_j-1)}$. Recall that, at iteration $t_{j-1}$, the cap $\overline{w}^{(t_{j-1})}$ is updated such that

$$\overline{w}_i^{(t_{j-1})} = C \cdot \max\left\{ w_i^{(1)}, \ldots, w_i^{(t_{j-1})} \right\} = C \cdot a_{j-1}.$$

Moreover, LazyHedge ensures that $\overline{w}_i^{(t)}$ is non-decreasing in $t$. Therefore, we have

$$C \cdot a_{j-1} = \overline{w}_i^{(t_{j-1})} \leq \overline{w}_i^{(t_j-1)} \leq a_j.$$

It follows that $a_m \geq C^{m-1} a_1$. Since $a_1 \geq w_i^{(1)} = 1/k$ and $a_m \leq 1$, we have $m - 1 \leq \log_C(a_m/a_1) \leq \log_C k$, which gives $|\mathcal{T}_i| = m = O(\log_C k)$.

**Only a Few $\mathcal{T}_i$s are Non-empty** The number of indices $i \in [k]$ such that $\mathcal{T}_i \neq \emptyset$ is trivially upper bounded by $k$. Next, we give another upper bound of $O((k/C) \cdot \log^8(k/\varepsilon))$. Fix $i \in [k]$. Suppose that $\mathcal{T}_i$ is non-empty and let $t_1$ be the smallest element in $\mathcal{T}_i$. Either one of the following must be true:

- $t_1 = 1$. This can only hold for $i = 1$.

- $t_1 > 1$. By definition of Algorithm 3, the cap is set to $\overline{w}^{(1)} = (C/k, C/k, \ldots, C/k)$ in the first iteration. For index $i$ to be the culprit at time $t_1 > 1$, we must have

$$w_i^{(t_1)} > \overline{w}_i^{(t_1-1)} \geq \overline{w}_i^{(1)} = C/k,$$

which implies $\max_{1 \leq t \leq T} w_i^{(t)} \geq C/k$. By Lemma 3, at most

$$\frac{\sum_{i=1}^{k} \max_{1 \leq t \leq T} w_i^{(t)}}{C/k} \leq \frac{k}{C} \cdot O(\log^8(k/\varepsilon))$$

different indices $i$ can satisfy this condition.

Thus, the number of non-empty sets $\mathcal{T}_i$ is at most $O((k/C) \cdot \log^8(k/\varepsilon))$.

Putting everything together, the round complexity of LazyHedge is given by

$$\sum_{i=1}^{k} |\mathcal{T}_i| \leq \sum_{i \in [k]: \mathcal{T}_i \neq \emptyset} O(\log_C k) \leq \min\left\{k, O((k/C) \cdot \log^8(k/\varepsilon))\right\} \cdot O(\log_C k).$$

$\square$

### C.3  Upper Bound for the Ellipsoid Setting

In the ellipsoid setting, we have an improved upper bound $\widetilde{O}(\sqrt{k})$ on the round complexity of LazyHedge.

**Lemma 19.** *For any $C \in [4, k]$, LazyHedge takes $O(\sqrt{k/C} \cdot \log^8(k/\varepsilon))$ rounds in the ellipsoid setting.*

Again, combining Lemmas 5, 6 and 19 immediately shows that the ellipsoid setting of OODS can be solved with an $\widetilde{O}(C)$ sample overhead in $\widetilde{O}(\sqrt{k/C})$ rounds.

**Theorem 20.** *For any $C \in [4, k]$, in the ellipsoid setting, LazyHedge finds an $O(\varepsilon)$-approximate maximum with an $O(C \log^8(k/\varepsilon))$ sample overhead in $O(\sqrt{k/C} \cdot \log^8(k/\varepsilon))$ rounds.*

The proof of Lemma 19 is significantly more technical than that of Lemma 7. We classify the cap updates (except the one in the first iteration) into two types: A "Type I" update is when some coordinate $w_i$ exceeds $1/\sqrt{k/C}$, and a "Type II" update is one without a significant increase in any of the $k$ coordinates. We bound the number of Type I and Type II updates by $\widetilde{O}(\sqrt{k/C})$ separately.

The upper bound for Type I updates is a simple consequence of the $\mathrm{polylog}(k/\varepsilon)$ upper bound on the Hedge trajectory shown by [ZZC$^+$24] (Lemma 3). This bound implies that there are at most $\widetilde{O}(1) \cdot \sqrt{k/C}$ Type I updates where the coordinate reaches $\approx 1/\sqrt{k/C}$, at most $\widetilde{O}(1) \cdot \sqrt{k/C}/2$ Type I updates where the coordinate reaches $\approx 2/\sqrt{k/C}$, and so on. These upper bounds sum up to $\widetilde{O}(\sqrt{k/C})$.

The analysis for Type II updates is more involved. Roughly speaking, we say that a coordinate $i \in [k]$ gains a *potential* of $a^2/b$ when it increases from $b$ to $a$ through the Hedge dynamics. Then, we prove the following two claims: (1) Each Type II update may happen only if an $\Omega(1)$ potential is accrued over all $k$ coordinates; (2) The total amount of potential that the $k$ coordinates may contribute to Type II updates is at most $\widetilde{O}(\sqrt{k/C})$. Combining these two claims proves the $\widetilde{O}(\sqrt{k/C})$ upper bound on the number of Type II updates.

*Proof of Lemma 19.* Consider an iteration $t \in \{2, 3, \ldots, T\}$ of LazyHedge in which the cap is updated, i.e., $w^{(t)} \notin \mathcal{O}(\overline{w}^{(t-1)})$. By definition of the ellipsoid setting (Definition 1), we have

$$\sum_{i=1}^{k} \frac{[w_i^{(t)}]^2}{\overline{w}_i^{(t-1)}} > 1.$$

Let $\mathcal{I}_t^{\mathrm{sig}} := \{i \in [k] : w_i^{(t)} \geq \overline{w}_i^{(t-1)}/2\}$ denote the set of *significant* indices on which the weight $w_i^{(t)}$ exceeds half of the cap $\overline{w}_i^{(t-1)}$. Let $\mathcal{I}_t^{\mathrm{insig}} := \{i \in [k] : w_i^{(t)} < \overline{w}_i^{(t-1)}/2\} = [k] \setminus \mathcal{I}^{\mathrm{sig}}$ be the set

of *insignificant* indices. Then, the contribution from insignificant indices to the left-hand side can be upper bounded:

$$\sum_{i \in \mathcal{I}_t^{\text{insig}}} \frac{[w_i^{(t)}]^2}{\overline{w}_i^{(t-1)}} \leq \sum_{i \in \mathcal{I}_t^{\text{insig}}} \frac{[w_i^{(t)}]^2}{2w_i^{(t)}} = \frac{1}{2} \sum_{i \in \mathcal{I}_t^{\text{insig}}} w_i^{(t)} \leq \frac{1}{2},$$

where the first step applies $i \in \mathcal{I}_t^{\text{insig}} \implies \overline{w}_i^{(t-1)} > 2w_i^{(t)}$, and the last step applies $w^{(t)} \in \Delta^{k-1}$. Therefore, the contribution from the significant indices is at least

$$\sum_{i \in \mathcal{I}_t^{\text{sig}}} \frac{[w_i^{(t)}]^2}{\overline{w}_i^{(t-1)}} = \sum_{i=1}^{k} \frac{[w_i^{(t)}]^2}{\overline{w}_i^{(t-1)}} - \sum_{i \in \mathcal{I}_t^{\text{insig}}} \frac{[w_i^{(t)}]^2}{\overline{w}_i^{(t-1)}} \geq 1 - \frac{1}{2} = \frac{1}{2}.$$

We say that the cap update in iteration $t$ is *Type I* if there exists an index $i \in \mathcal{I}_t^{\text{sig}}$ such that

$$\max\left\{ w_i^{(1)}, w_i^{(2)}, \ldots, w_i^{(t)} \right\} \geq \frac{1}{\sqrt{k/C}};$$

otherwise, the update is *Type II*. Let $\mathcal{T}_1 \subseteq [T]$ and $\mathcal{T}_2 \subseteq [T]$ denote the iterations in which a Type I and a Type II cap update happens, respectively. Then, the round complexity is simply $|\mathcal{T}_1| + |\mathcal{T}_2| + 1$, where the "+1" accounts for the cap update at $t = 1$, which is neither Type I nor Type II.

**Number of Type I Updates**   We further classify the Type I cap updates (in $\mathcal{T}_1$) into $O(\log k)$ *sub-types*. For each integer $j \in [0, \log_2 \sqrt{k/C}]$, let $\mathcal{T}_{1,j}$ denote the set of pairs $(t, i) \in \mathcal{T}_1 \times [k]$ such that: (1) A Type I cap update happens in iteration $t$; (2) $i \in \mathcal{I}_t^{\text{sig}}$ is the smallest index such that

$$\max\left\{ w_i^{(1)}, w_i^{(2)}, \ldots, w_i^{(t)} \right\} \geq 1/\sqrt{k/C};$$

(3) It holds that

$$\max\left\{ w_i^{(1)}, w_i^{(2)}, \ldots, w_i^{(t)} \right\} \in \left[ \frac{2^j}{\sqrt{k/C}}, \frac{2^{j+1}}{\sqrt{k/C}} \right).$$

Then, it remains to control the size of each $\mathcal{T}_{1,j}$.

We first argue that only a few indices $i \in [k]$ may appear as the second coordinate of a $(t, i)$-pair in $\mathcal{T}_{1,j}$. Indeed, if $(t_0, i) \in \mathcal{T}_{1,j}$, we must have $\max_{1 \leq t \leq T} w_i^{(t)} \geq \max_{1 \leq t \leq t_0} w_i^{(t)} \geq \frac{2^j}{\sqrt{k/C}}$. Recall from Lemma 3 that $\sum_{i=1}^{k} \max_{1 \leq t \leq T} w_i^{(t)} \leq O(\log^8(k/\varepsilon))$, so the aforementioned condition hold for at most $2^{-j} \cdot O(\sqrt{k/C} \log^8(k/\varepsilon))$ different values of $i \in [k]$.

Then, we argue that no $\mathcal{T}_{1,j}$ may contain two pairs $(t_1, i)$ and $(t_2, i)$ for $t_1 \neq t_2$. In other words, no index $i$ can contribute to the same sub-type $j$ twice. Suppose towards a contradiction that for some $t_1 < t_2$ and $i \in [k]$, $(t_1, i), (t_2, i) \in \mathcal{T}_{1,j}$. By definition of $\mathcal{T}_{1,j}$, $\max_{1 \leq t \leq t_1} w_i^{(t)} \geq 2^j/\sqrt{k/C}$. Then, by the cap update in LazyHedge, we have

$$\overline{w}_i^{(t_2-1)} \geq \overline{w}_i^{(t_1)} = C \cdot \max_{1 \leq t \leq t_1} w_i^{(t)} \geq C \cdot \frac{2^j}{\sqrt{k/C}}.$$

On the other hand, since $(t_2, i) \in \mathcal{T}_{1,j}$, we have $i \in \mathcal{I}_{t_2}^{\text{sig}}$, which further implies

$$w_i^{(t_2)} \geq \frac{1}{2}\overline{w}_i^{(t_2-1)} \geq \frac{C}{2} \cdot \frac{2^j}{\sqrt{k/C}} \geq \frac{2^{j+1}}{\sqrt{k/C}},$$

where the last step applies $C \geq 4$. We then have $\max_{1 \leq t \leq t_2} w_i^{(t)} \geq w_i^{(t_2)} \geq 2^{j+1}/\sqrt{k/C}$, which contradicts $(t_2, i) \in \mathcal{T}_{1,j}$.

Therefore, the number of Type I updates is at most

$$|\mathcal{T}_1| = \sum_{j=0}^{\lfloor \log_2 \sqrt{k/C} \rfloor} |\mathcal{T}_{1,j}| \leq \sum_{j=0}^{\lfloor \log_2 \sqrt{k/C} \rfloor} 2^{-j} \cdot O(\sqrt{k/C} \log^8(k/\varepsilon)) = O(\sqrt{k/C} \log^8(k/\varepsilon)).$$

**Number of Type II Updates: Overview** Recall that if a cap update happens in iteration $t \geq 2$, we have $\sum_{i \in \mathcal{I}_t^{\text{sig}}} \frac{[w_i^{(t)}]^2}{w_i^{(t-1)}} \geq \frac{1}{2}$. Summing over $t \in \mathcal{T}_2$ gives

$$\sum_{t \in \mathcal{T}_2} \sum_{i \in \mathcal{I}_t^{\text{sig}}} \frac{[w_i^{(t)}]^2}{\overline{w}_i^{(t-1)}} \geq \frac{|\mathcal{T}_2|}{2}. \tag{8}$$

Next, we upper bound the double summation on the left-hand side of (8) by changing the order of summation. We will show that, for every $i \in [k]$, it holds that

$$\sum_{t \in \mathcal{T}_2 : i \in \mathcal{I}_t^{\text{sig}}} \frac{[w_i^{(t)}]^2}{\overline{w}_i^{(t-1)}} \leq \min\left\{ \frac{k}{C} \left( \max_{1 \leq t \leq T} w_i^{(t)} \right)^2, 1 \right\}. \tag{9}$$

Furthermore, we will upper bound the sum of the right-hand side of (9) as follows:

$$\sum_{i=1}^{k} \min\left\{ \frac{k}{C} \left( \max_{1 \leq t \leq T} w_i^{(t)} \right)^2, 1 \right\} \leq O(\sqrt{k/C} \log^8(k/\varepsilon)). \tag{10}$$

Then, combining Equations (8) through (10) immediately gives $|\mathcal{T}_2| \leq O(\sqrt{k/C} \log^8(k/\varepsilon))$ and completes the proof. We prove Equations (9) and (10) in the remainder of the proof.

**Proof of Equation** (9) Towards proving Equation (9), we fix $i \in [k]$ and list the elements in $\mathcal{T}_2$ that satisfies $i \in \mathcal{I}_t^{\text{sig}}$ in increasing order: $2 \leq t_1 < t_2 < \cdots < t_m \leq T$. We write $t_0 = 1$ and $a_j := \max_{1 \leq t \leq t_j} w_i^{(t)}$ for every $j \in \{0, 1, \ldots, m\}$.

We first upper bound the left-hand side of (9) in terms of $(a_j)_{j=0}^{m}$. For each $j \in [m]$, we have

$$w_i^{(t_j)} \leq \max_{1 \leq t \leq t_j} w_i^{(t)} = a_j \quad \text{and} \quad \overline{w}_i^{(t_j - 1)} \geq \overline{w}_i^{(t_{j-1})} = C \cdot \max_{1 \leq t \leq t_{j-1}} w_i^{(t)} = C \cdot a_{j-1}.$$

Then, each $j \in [m]$ contributes a term of

$$\frac{[w_i^{(t_j)}]^2}{\overline{w}_i^{(t_j - 1)}} \leq \frac{a_j^2}{C \cdot a_{j-1}} = \frac{1}{C} \cdot \frac{a_j^2}{a_{j-1}}$$

to the left-hand side of (9). Thus, it suffices to upper bound $\sum_{j=1}^{m} \frac{a_j^2}{a_{j-1}}$.

Next, we examine the sequence $(a_j)_{j=0}^{m}$. Fix $j \in [m]$. Since $i \in \mathcal{I}_{t_j}^{\text{sig}}$, we have

$$a_j \geq w_i^{(t_j)} \geq \frac{1}{2}\overline{w}_i^{(t_j - 1)} \geq \frac{1}{2}\overline{w}_i^{(t_{j-1})}.$$

The cap update at time $t_{j-1}$ ensures that

$$\overline{w}_i^{(t_{j-1})} = C \cdot \max_{1 \leq t \leq t_{j-1}} w_i^{(t)} = C \cdot a_{j-1}.$$

Combining the above and applying the assumption that $C \geq 4$ gives $a_j \geq \frac{C}{2} a_{j-1} \geq 2 a_{j-1}$.

The sequence $(a_j)_{j=0}^{m}$ starts with $a_0 = w_i^{(1)} = 1/k$ and ends with $a_m = \max_{1 \leq t \leq t_m} w_i^{(t)}$. Moreover, since the cap update in iteration $t_m$ is Type II and $i \in \mathcal{I}_{t_m}^{\text{sig}}$, it holds that

$$\max_{1 \leq t \leq t_m} w_i^{(t)} < \frac{1}{\sqrt{k/C}}.$$

Therefore, we have the upper bound $a_m \leq \min\left\{ \max_{1 \leq t \leq T} w_i^{(t)}, \frac{1}{\sqrt{k/C}} \right\}$. In summary, $(a_0, a_1, \ldots, a_m)$ is a sequence of positive numbers such that:

- $a_j \geq 2 a_{j-1}$.

- $a_0 = 1/k$ and $a_m \le \min\left\{\max_{1\le t\le T} w_i^{(t)}, \frac{1}{\sqrt{k/C}}\right\}$.

For any positive numbers $a, b, c$ that satisfy $a/b \ge 2$ and $b/c \ge 2$, we have

$$\frac{a^2}{b} \le \frac{3}{4} \cdot \frac{a^2}{c} = \frac{a^2}{c} - \frac{(a/2)^2}{c} \le \frac{a^2}{c} - \frac{b^2}{c}.$$

Rearranging gives $\frac{a^2}{b} + \frac{b^2}{c} \le \frac{a^2}{c}$. Therefore, starting from the sequence $(a_0, a_1, \ldots, a_m)$, we may repeatedly remove the elements $a_{m-1}, a_{m-2}, \ldots, a_1$ one by one. By doing so, the invariant $a_j/a_{j-1} \ge 2$ is always maintained, and the value of $\sum_{j=1}^m a_j^2/a_{j-1}$ never decreases. Therefore, we have

$$\sum_{j=1}^m \frac{a_j^2}{a_{j-1}} \le \frac{a_m^2}{a_0} \le \frac{\left(\min\left\{\max_{1\le t\le T} w_i^{(t)}, \frac{1}{\sqrt{k/C}}\right\}\right)^2}{1/k} = \min\left\{k\left(\max_{1\le t\le T} w_i^{(t)}\right)^2, C\right\}.$$

Recalling that the left-hand side of (9) is at most $\frac{1}{C} \cdot \sum_{j=1}^m \frac{a_j^2}{a_{j-1}}$, we have proved the inequality.

**Proof of Equation** (10)   For each integer $j \in [0, \log_2 k]$, let

$$\mathcal{I}_j := \left\{i \in [k] : \max_{1\le t\le T} w_i^{(t)} \in \left[\frac{2^j}{k}, \frac{2^{j+1}}{k}\right)\right\}$$

denote the set of indices $i \in [k]$ on which the maximum weight over the $T$ iterations is roughly $2^j/k$. Then, the upper bound $\sum_{i=1}^k \max_{1\le t\le T} w_i^{(t)} \le O(\log^8(k/\varepsilon))$ from Lemma 3 implies $|\mathcal{I}_j| \le 2^{-j} \cdot O(k \log^8(k/\varepsilon))$. Note that $\mathcal{I}_0, \mathcal{I}_1, \ldots, \mathcal{I}_{\lceil \log_2 k \rceil}$ form a partition of $[k]$, so we have

$$\sum_{i=1}^k \min\left\{\frac{k}{C}\left(\max_{1\le t\le T} w_i^{(t)}\right)^2, 1\right\} = \sum_{j=0}^{\lceil \log_2 k \rceil} \sum_{i\in \mathcal{I}_j} \min\left\{\frac{k}{C}\left(\max_{1\le t\le T} w_i^{(t)}\right)^2, 1\right\}$$

$$\le \sum_{j=0}^{\lceil \log_2 k \rceil} |\mathcal{I}_j| \cdot \min\left\{\frac{k}{C}\left(\frac{2^{j+1}}{k}\right)^2, 1\right\}$$

$$\le O(k\log^8(k/\varepsilon)) \cdot \sum_{j=0}^{\lceil \log_2 k \rceil} 2^{-j} \cdot \min\left\{\frac{k}{C}\left(\frac{2^{j+1}}{k}\right)^2, 1\right\},$$

where the second step follows from $i \in \mathcal{I}_j \implies \max_{1\le t\le T} w_i^{(t)} < 2^{j+1}/k$, and the third step applies the upper bound $|\mathcal{I}_j| \le 2^{-j} \cdot O(k\log^8(k/\varepsilon))$.

The summation $\sum_{j=0}^{\lceil \log_2 k \rceil} 2^{-j} \cdot \min\left\{\frac{k}{C}\left(\frac{2^{j+1}}{k}\right)^2, 1\right\}$ is further upper bounded by

$$O(1) \cdot \sum_{j=0}^{+\infty} \min\left\{\frac{2^j}{Ck}, 2^{-j}\right\}.$$

The summand $\min\left\{\frac{2^j}{Ck}, 2^{-j}\right\}$ is given by the first term and thus geometrically increasing when $4^j \le Ck$. The summand is geometrically decreasing when $4^j \ge Ck$. Therefore, the summation is dominated by the term at $j^* = \lceil \log_4(Ck) \rceil$, namely, $2^{-j^*} = O(1/\sqrt{Ck})$. Therefore, we have

$$\sum_{i=1}^k \min\left\{\frac{k}{C}\left(\max_{1\le t\le T} w_i^{(t)}\right)^2, 1\right\} \le O(k\log^8(k/\varepsilon)) \cdot O(1/\sqrt{Ck}) = O(\sqrt{k/C}\log^8(k/\varepsilon)).$$

This proves Equation (10) and completes the proof. $\qquad\square$

## C.4 Applications to Multi-Distribution Learning

We sketch how our proofs of Theorem 8 and Theorem 20 can be easily adapted to give MDL algorithms with near-optimal sample complexities that run in either $\widetilde{O}(k)$ or $\widetilde{O}(\sqrt{k})$ rounds. We prove these results by modifying an algorithm of [ZZC+24], which we briefly describe below. We refer the reader to [ZZC+24, Algorithm 1] for the full pseudocode description.

**The MDL Algorithm of [ZZC+24]**  Let $\mathcal{H}$ be the hypothesis class with VC dimension $d$. For brevity, we treat the failure probability $\delta$ as a constant, and use the $\widetilde{O}(\cdot)$ and $\widetilde{\Theta}(\cdot)$ notations to suppress $\mathrm{polylog}(kd/\varepsilon)$ factors. The algorithm maintains $k$ datasets $S_1, S_2, \ldots, S_k$, where each $S_i$ contains training examples drawn from the $i$-th data distribution $D_i$. The algorithm runs the Hedge dynamics for $T = \Theta((\log k)/\varepsilon^2)$ iterations starting at $w^{(1)} = (1/k, 1/k, \ldots, 1/k)$. Each iteration $t \in [T]$ consists of the following two steps:

- **ERM step:** For each $i \in [k]$, draw additional samples from $D_i$ and add them to $S_i$ until

$$|S_i| \geq w_i^{(t)} \cdot \widetilde{\Theta}((d+k)/\varepsilon^2).$$

  Then, find a hypothesis $h^{(t)} \in \mathcal{H}$ that approximate minimizes the empirical error

$$\hat{L}(h) := \sum_{i=1}^{k} w_i^{(t)} \cdot \frac{1}{|S_i|} \sum_{(x,y) \in S_i} \mathbb{1}\left[h(x) \neq y\right],$$

  which is an estimate of the error of $h$ on $\sum_{i=1}^{k} w_i^{(t)} D_i$.

- **Hedge update step:** For each $i \in [k]$, draw $w_i^{(t)} \cdot \Theta(k)$ *fresh* samples from $D_i$ to obtain an estimate $r_i^{(t)}$ of the error of $h^{(t)}$ on $D_i$. Compute $w^{(t+1)}$ from $w^{(t)}$ and $r^{(t)}$ via a Hedge update.

The crux of the analysis of [ZZC+24] is to show that the dataset sizes in the two steps above are sufficient for finding a sufficiently accurate ERM $h^{(t)}$ as well as computing the loss vector $r^{(t)}$ that is sufficiently accurate for the Hedge update.

Note that a straightforward implementation of the algorithm needs $T = \Theta((\log k)/\varepsilon^2)$ rounds of sampling. While this round complexity is logarithmic in $k$, it has a polynomial dependence in $1/\varepsilon$. In the following, we apply the ideas behind our OODS upper bounds to improve this round complexity to $\widetilde{O}(k)$ and then $\widetilde{O}(\sqrt{k})$, which is lower than $(\log k)/\varepsilon^2$ in the $\varepsilon \ll 1/k^{1/4}$ regime.

**$\widetilde{O}(k)$-Round MDL from Theorem 8**  The LazyHedge algorithm (Algorithm 3) suggests a natural modification to the algorithm of [ZZC+24]: To ensure $|S_i| \geq w_i^{(t)} \cdot \widetilde{\Theta}((d+k)/\varepsilon^2)$, instead of drawing additional samples at every round, we maintain a cap vector $\overline{w}^{(t)} \in [0,1]^k$. We ensure the invariant that, at the end of every iteration $t$, it holds for every $i \in [k]$ that $|S_i| \geq \overline{w}_i^{(t)} \cdot \widetilde{\Theta}((d+k)/\varepsilon^2)$.

At each round $t$, if $w_i^{(t)} \leq \overline{w}_i^{(t-1)}$ already holds for every $i \in [k]$, we use the current datasets to perform the ERM step. Since $|S_i| \geq \overline{w}_i^{(t-1)} \cdot \widetilde{\Theta}((d+k)/\varepsilon^2) \geq w_i^{(t)} \cdot \widetilde{\Theta}((d+k)/\varepsilon^2)$ holds for every $i \in [k]$, the uniform convergence result of [ZZC+24, Lemma 1] guarantees that $h^{(t)}$ is an $O(\varepsilon)$-accurate ERM. Otherwise, we update the cap vector to $\overline{w}^{(t)}$ according to LazyHedge, and then add fresh samples to each $S_i$ until $|S_i| \geq \overline{w}_i^{(t)} \cdot \widetilde{\Theta}((d+k)/\varepsilon^2)$.

It remains to handle the sampling in the Hedge update step, which requires $w_i^{(t)} \cdot \Theta(k)$ fresh samples from each $D_i$. If we draw these samples in each round, the MDL algorithm would still need $T = \Omega((\log k)/\varepsilon^2)$ rounds of sampling. Instead, we maintain a dataset $S_{i,t}$ for each pair $(i, t) \in [k] \times [T]$ that is reserved for the Hedge update step for distribution $D_i$ in the $t$-th iteration. We ensure the invariant that, at any iteration $t$,

$$|S_{i,t'}| \geq \overline{w}_i^{(t)} \cdot \Theta(k)$$

holds for every $i \in [k]$ and $t' \geq t$. To this end, whenever the cap vector $\overline{w}^{(t)}$ is updated, we add fresh samples to each $S_{i,t'}$ (where $t' \geq t$) so that the invariant above still holds. By doing so, we ensure

that in the Hedge update step at any iteration $t$, we have $|S_{i,t}| \geq \overline{w}_i^{(t)} \cdot \Theta(k) \geq w_i^{(t)} \cdot \Theta(k)$ samples. Furthermore, we only need one round of sampling whenever the cap vector is updated.

The above exactly corresponds to the LazyHedge algorithm in the box setting of OODS: Whenever the Hedge dynamics reaches a point $w^{(t)}$, the algorithm must update the cap $\overline{w}^{(t)}$ to ensure that $\overline{w}^{(t)} \geq w^{(t)}$. Every cap update in LazyHedge corresponds to a round of adaptive sampling in MDL. By Lemma 7, the resulting MDL algorithm has an $O(k \log k)$ round complexity. Moreover, the sample complexity of the MDL algorithm is at most

$$\sum_{i=1}^{k} \overline{w}_i^{(T)} \cdot \widetilde{\Theta}((d+k)/\varepsilon^2) + T \cdot \sum_{i=1}^{k} \overline{w}_i^{(T)} \cdot \Theta(k) = \widetilde{O}\left(\frac{C(d+k)}{\varepsilon^2}\right),$$

where we apply $T = \widetilde{O}(1/\varepsilon^2)$ and the bound $\sum_{i=1}^{k} \overline{w}_i^{(T)} = O(C \log^8(k/\varepsilon)) = \widetilde{O}(C)$ from Lemma 6, which in turn follows from [ZZC$^+$24, Lemma 3]. Setting $C = O(1)$ gives the following result for MDL.

**Corollary 21.** *There is an $O(k \log k)$-round MDL algorithm with an $\widetilde{O}((d+k)/\varepsilon^2)$ sample complexity.*

$\widetilde{O}(\sqrt{k})$**-Round MDL from Theorem 20** We further improve the round complexity to $\widetilde{O}(\sqrt{k})$ using our results for the ellipsoid setting (Theorem 20). The key observation is that the analysis of [ZZC$^+$24] does not require the "box" constraint $w_i^{(t)} \leq \overline{w}_i^{(t)}$ for every $i \in [k]$. Instead, the weaker "ellipsoid" constraint $\sum_{i=1}^{k} [w_i^{(t)}]^2 / \overline{w}_i^{(t)} \leq 1$ suffices.

To see this, we note that the analysis of [ZZC$^+$24] uses the lower bounds on the sample size at two different places, both of which go through under the ellipsoid constraint. The first place is the proof of the uniform convergence result [ZZC$^+$24, Lemma 1]. For fixed $w \in \Delta^{k-1}$, $n \in \mathbb{N}^k$ and $h \in \mathcal{H}$, the analysis boils down to showing that, if every dataset $S_i$ contains $n_i$ independent samples from $D_i$,

$$\sum_{i=1}^{k} \frac{w_i}{n_i} \sum_{(x,y) \in S_i} \mathbb{1}\left[h(x) \neq y\right]$$

is an estimate for the population error of $h$ on the mixture distribution $\sum_{i=1}^{k} w_i D_i$ with sub-Gaussian parameter $\sigma^2 \leq \sum_{i=1}^{k} \frac{w_i^2}{n_i}$, and thus concentrates around the population error up to an error of $O(\sigma \sqrt{\log(1/\delta)})$ except with probability $\delta$. In particular, assuming that $n_i \geq w_i \cdot \widetilde{\Theta}((d+k)/\varepsilon^2)$, we have

$$\sigma^2 \leq \sum_{i=1}^{k} \frac{w_i^2}{w_i \cdot \widetilde{\Theta}((d+k)/\varepsilon^2)} = \widetilde{\Theta}\left(\frac{\varepsilon^2}{d+k}\right) \sum_{i=1}^{k} w_i = \widetilde{\Theta}\left(\frac{\varepsilon^2}{d+k}\right).$$

We note that the same bound hold under the weaker assumption that, for some cap vector $\overline{w}$, it holds that $\sum_{i=1}^{k} w_i^2 / \overline{w}_i \leq 1$ and $n_i \geq \overline{w}_i \cdot \widetilde{\Theta}((d+k)/\varepsilon^2)$:

$$\sigma^2 \leq \sum_{i=1}^{k} \frac{w_i^2}{n_i} \leq \sum_{i=1}^{k} \frac{w_i^2}{\overline{w}_i \cdot \widetilde{\Theta}((d+k)/\varepsilon^2)} = \widetilde{\Theta}\left(\frac{\varepsilon^2}{d+k}\right) \cdot \sum_{i=1}^{k} w_i^2 / \overline{w}_i = \widetilde{\Theta}\left(\frac{\varepsilon^2}{d+k}\right).$$

The second place is Step 3 in the proof of [ZZC$^+$24, Lemma 17]. This step uses the fact that $w_i^{(t)} \cdot \Theta(k)$ samples are used to compute the estimate $r_i^{(t)}$, so that the estimate has a variance of $O\left(\frac{1}{kw_i^{(t)}}\right)$. Fortunately, this variance bound is only used in aggregate over all $i \in [k]$ in their Equation (111), which shows that the weighted average $\sum_{i=1}^{k} w_i^{(t)} \cdot r_i^{(t)}$ has a variance of at most

$$\sum_{i=1}^{k} [w_i^{(t)}]^2 \cdot O\left(\frac{1}{kw_i^{(t)}}\right) = O(1/k) \cdot \sum_{i=1}^{k} w_i^{(t)} = O(1/k).$$

Again, this step would still go through under the ellipsoid constraint $\sum_{i=1}^{k} [w_i^{(t)}]^2 / \overline{w}_i^{(t)}$: As long as at least $\overline{w}_i^{(t)} \cdot \Theta(k)$ fresh samples are used to estimate $r_i^{(t)}$, the weighted average $\sum_{i=1}^{k} w_i^{(t)} \cdot r_i^{(t)}$ has

a variance of at most

$$\sum_{i=1}^{k}[w_i^{(t)}]^2 \cdot O\left(\frac{1}{k\overline{w}_i^{(t)}}\right) = O(1/k) \cdot \sum_{i=1}^{k}\frac{[w_i^{(t)}]^2}{\overline{w}_i^{(t)}} = O(1/k).$$

Therefore, Theorem 20 implies the following result for MDL.

**Corollary 22.** *There is an $\widetilde{O}(\sqrt{k})$-round MDL algorithm with an $\widetilde{O}((d+k)/\varepsilon^2)$ sample complexity.*

## D   Lower Bounds for OODS

We prove lower bounds on the sample-adaptivity tradeoff in the OODS model using two different hard instances: one for the "large-$\varepsilon$" regime where $\varepsilon \le O(1/k)$, and the other for the "small-$\varepsilon$" regime where $\varepsilon \le e^{-\Omega(k)}$.

### D.1   Hard Instance for Large $\varepsilon$

**Definition 4** (Hard instance for large $\varepsilon$). Given $k \ge m \ge 1$, draw $m$ different indices $i_1^\star, i_2^\star, \ldots, i_m^\star \in [k]$ uniformly at random. The objective function $f : \Delta^{k-1} \to [0,2]$ is

$$f(w) := \min\left\{w_{i_1^\star} + \frac{1}{m^2}, w_{i_2^\star} + \frac{2}{m^2}, \ldots, w_{i_m^\star} + \frac{m}{m^2}\right\}.$$

Note that we allow the co-domain of $f$ to be $[0,2]$ instead of $[0,1]$ for brevity; scaling everything down by a factor of 2 gives a hard instance and thus lower bounds for the formulation in Definition 1.

Before we formally state and prove the lower bounds, we make a few simple observations and then sketch the intuition behind the lower bound proof.

**Characterization of Approximate Maxima**   If we set $w_{i_j^\star} = \frac{3m+1}{2m^2} - \frac{j}{m^2}$ for every $j \in [m]$ and $w_i = 0$ for every $i \in [k] \setminus \{i_1^\star, i_2^\star, \ldots, i_m^\star\}$, we have

$$w_{i_1^\star} + \frac{1}{m^2} = w_{i_2^\star} + \frac{2}{m^2} = \cdots = w_{i_m^\star} + \frac{m}{m^2} = \frac{3m+1}{2m^2},$$

which gives $f(w) = \frac{3m+1}{2m^2}$. Furthermore, it can be easily verified that $w \in \Delta^{k-1}$. We note that $w$ is the maximizer of $f$ over $\Delta^{k-1}$, since achieving an objective strictly higher than $\frac{3m+1}{2m^2}$ requires a strict increase in each of $w_{i_1^\star}, \ldots, w_{i_m^\star}$, which would violate the constraint that $w \in \Delta^{k-1}$.

More generally, we have a "robust" version of this observation: As long as $\varepsilon \le O(1/k)$, every $\varepsilon$-approximate maximum of $f$ must put a significant weight of $\Omega(1/m)$ on at least half of the *critical indices* $i_1^\star, \ldots, i_m^\star$.

**Lemma 23.** *For every $\varepsilon \le 1/(2k)$ and every $w \in \Delta^{k-1}$ that satisfies $f(w) \ge \frac{3m+1}{2m^2} - \varepsilon$, we have*

$$|\{i \in [k] : w_i \ge 1/(2m)\} \cap \{i_1^\star, i_2^\star, \ldots, i_m^\star\}| \ge m/2.$$

*Proof.* Suppose towards a contradiction that strictly fewer than $m/2$ entries among $w_{i_1^\star}, w_{i_2^\star}, \ldots, w_{i_m^\star}$ are at least $1/(2m)$. Then, there exists $j \in \{1, 2, \ldots, \lceil m/2 \rceil\}$ such that $w_{i_j^\star} < 1/(2m)$. It follows that

$$f(w) \le w_{i_j^\star} + \frac{j}{m^2} < \frac{1}{2m} + \frac{(m+1)/2}{m^2} = \frac{2m+1}{2m^2}.$$

On the other hand, since $m \le k$ and $\varepsilon \le 1/(2k)$, we have

$$\frac{3m+1}{2m^2} - \varepsilon \ge \frac{3m+1}{2m^2} - \frac{1}{2m} = \frac{2m+1}{2m^2} > f(w),$$

a contradiction.                                                                                    □

**Limited Information from a First-Order Oracle**  Therefore, it remains to argue that an OODS algorithm cannot identify $\Omega(m)$ critical indices using $\ll m$ rounds while incurring a $\mathrm{polylog}(k)$ sample overhead. To this end, we observe that a first-order oracle provides little information on the critical indices, unless we have already found many such indices.

By definition of $f$ from Definition 4, the following is a valid first-order oracle: Given $w \in \Delta^{k-1}$, find the minimum $j \in [m]$ such that $f(w) = w_{i_j^\star} + j/m^2$. Return the value of $f(w)$ and the supergradient $e_{i_j^\star}$. In other words, via a first-order oracle, the optimization algorithm only gets to know the critical index that accounts for the value of $f(w)$.

Our next lemma suggests that, unless we put a weight of $> 1/m^2$ on each of $i_1^\star, i_2^\star, \ldots, i_j^\star$, the value of $f(w)$ is determined by the first $j$ terms in the minimum. In other words, unless we already "know" the first $j$ critical indices, we cannot learn the value of $i_{j+1}^\star, \ldots, i_m^\star$ from the oracle.

**Lemma 24.** *For every $w \in \Delta^{k-1}$ and $j \in [m]$, assuming $\min\{w_{i_1^\star}, w_{i_2^\star}, \ldots, w_{i_j^\star}\} \le 1/m^2$, we have*

$$f(w) = \min\left\{ w_{i_1^\star} + \frac{1}{m^2}, w_{i_2^\star} + \frac{2}{m^2}, \ldots, w_{i_j^\star} + \frac{j}{m^2} \right\}.$$

*Proof.* Suppose towards a contradiction that some $w \in \Delta^{k-1}$ satisfies $w_{i_{j_1}^\star} \le 1/m^2$ for some $j_1 \le j$, but $f(w) \ne \min\left\{ w_{i_1^\star} + \frac{1}{m^2}, w_{i_2^\star} + \frac{2}{m^2}, \ldots, w_{i_j^\star} + \frac{j}{m^2} \right\}$. Then, there exists $j_2 > j$ such that

$$w_{i_{j_2}^\star} + \frac{j_2}{m^2} < \min\left\{ w_{i_1^\star} + \frac{1}{m^2}, w_{i_2^\star} + \frac{2}{m^2}, \ldots, w_{i_j^\star} + \frac{j}{m^2} \right\} \le w_{i_{j_1}^\star} + \frac{j_1}{m^2},$$

which implies $w_{i_{j_1}^\star} > w_{i_{j_2}^\star} + (j_2 - j_1)/m^2 \ge 1/m^2$, a contradiction. $\square$

**Intuition behind Lower Bound**  There is a natural $m$-round algorithm for solving the instance from Definition 4: In the first round, we query the uniform weight vector $w = (1/k, 1/k, \ldots, 1/k)$ to learn the value of $i_1^\star$. In the second round, we query $f$ on some $w$ with $w_{i_1^\star} \gg 1/m^2$, thereby learning the value of $i_2^\star$. Repeating this gives a $m$-round algorithm with a low sample overhead.

One might hope to be "more clever" and learn more than one critical index in each round. For example, the algorithm might put a cap of $\gg 1/m^2$ on several coordinates in the first round, in the hope of hitting more than one indices in $i_1^\star, i_2^\star, \ldots$. However, if the algorithm has a sample overhead of $s$, only $O(m^2 s)$ such guesses can be made. In particular, if $m^2 s \ll k$, the guesses only cover a tiny fraction of the indices. Thus, over the uniform randomness in $i^\star$, the algorithm can still only learn $O(1)$ critical indices in expectation within each round.

### D.2 Lower Bounds for Large $\varepsilon$

Now, we formally state and prove the lower bounds in the $\varepsilon \le O(1/k)$ regime, for both the box and the ellipsoid settings.

**Theorem 25.** *The following holds for all sufficiently large $k$, $\varepsilon \le 1/(2k)$ and $s \ge 1$: (1) In the box setting, for $m := \lfloor \frac{1}{2}\sqrt{k/s} \rfloor$, no OODS algorithm with $r \le m/24$ rounds and sample overhead $s$ can find an $\varepsilon$-approximate maximum with probability at least $9/10$; (2) In the ellipsoid setting, for $m := \lfloor \frac{1}{2}(k/s)^{1/4} \rfloor$, no OODS algorithm with $r \le m/24$ rounds and sample overhead $s$ can find an $\varepsilon$-approximate maximum with probability at least $9/10$.*

In particular, to have a sample overhead $s \le \mathrm{polylog}(k)$, any OODS algorithm must take $\widetilde{\Omega}(\sqrt{k})$ rounds in the box setting, and $\widetilde{\Omega}(k^{1/4})$ rounds in the ellipsoid setting.

*Proof.* We will focus on the box setting; the ellipsoid setting follows easily with only a few changes in the proof.

Suppose towards a contradiction that an OODS algorithm $\mathcal{A}$ for the box setting takes $r \le m/24$ rounds and succeeds with probability at least $9/10$. Consider the execution of $\mathcal{A}$ on a random OODS instance defined in Definition 4 with $m := \lfloor \frac{1}{2}\sqrt{k/s} \rfloor$.

For each round $t \in [r]$, let $I^{(t)} := \{i \in [m] : \overline{w}_i^{(t)} > 1/m^2\}$ denote the indices on which algorithm $\mathcal{A}$ sets a cap strictly higher than $1/m^2$. Since $\mathcal{A}$ has a sample overhead of $s$, we have $\|\overline{w}^{(t)}\|_1 \leq s$ and thus $|I^{(t)}| \leq m^2 s$. By definition of the box setting, within each round $t$, $\mathcal{A}$ cannot query the oracle on $w$ if $w_i > 1/m^2$ holds for some $i \in [k] \setminus I^{(t)}$. Therefore, Lemma 24 implies the following: Unless $\{i_1^\star, i_2^\star, \ldots, i_j^\star\} \subseteq I^{(t)}$, for every point $w$ that $\mathcal{A}$ queries in round $t$, it holds that

$$f(w) = \min \left\{ w_{i_1^\star} + \frac{1}{m^2}, w_{i_2^\star} + \frac{2}{m^2}, \ldots, w_{i_j^\star} + \frac{j}{m^2} \right\}.$$

**Deferring Randomness**   Rather than drawing all the $m$ critical indices at the beginning, in our analysis, we choose these indices "on demand" as the OODS algorithm expands the observable region. We do this carefully, so that the distribution of $i^\star$ is not biased.

In the first round, after $\overline{w}^{(1)}$ (and thus the set $I^{(1)}$) has been determined by $\mathcal{A}$, we draw $i_1^\star$ from $[k]$ uniformly at random. If $i_1^\star \notin I^{(1)}$, we stop the process; otherwise, we draw $i_2^\star$ from $[k] \setminus \{i_1^\star\}$ uniformly at random. We keep doing this, until either all the $m$ critical indices are determined, or we draw some $i_j^\star \notin I^{(1)}$. Let $n^{(1)}$ denote the number of critical indices drawn in the first round. By Lemma 24, for every $w \in \mathcal{O}(\overline{w}^{(1)})$ observable to $\mathcal{A}$ in the first round, $f(w)$ is determined by the first $n^{(1)}$ critical indices. In particular, the remaining $m - n^{(1)}$ critical indices ($i_{n^{(1)}+1}^\star$ through $i_m^\star$) are still uniformly distributed among $[k] \setminus \{i_1^\star, \ldots, i_{n^{(1)}}^\star\}$, conditioned on any information that $\mathcal{A}$ obtains from the first-order oracle.

More generally, at the beginning of each round $t \in [r]$, we observe the set $I^{(t)}$ is determined by algorithm $\mathcal{A}$. We keep sampling $i_{n^{(t-1)}+1}^\star, i_{n^{(t-1)}+2}^\star, \ldots$, until either all $m$ indices are chosen or we encounter an index outside $I^{(t)}$. Let $n^{(t)}$ denote the total number of critical indices that have been sampled, including those sampled in the first $t-1$ rounds. Then, all queries made by $\mathcal{A}$ during the $t$-th round can be answered solely based on the values of $i_1^\star$ through $i_{n^{(t)}}^\star$, as they do not depend on the $m - n^{(t)}$ indices that have not been decided.

**Control the Progress Measure**   Let random variable $N^{(t)}$ denote the value of $n^{(t)}$ in round $t$, over the randomness in both algorithm $\mathcal{A}$ and the random drawing of the critical indices. Next, we upper bound the expectation of $N^{(r)}$ after all $r$ rounds.

In each round $t \in [r]$, whenever we sample a critical index $i_j^\star$ ($j \in [m]$), there are $k - (j-1) > k - m$ possible choices (namely, $[k] \setminus \{i_1^\star, \ldots, i_{j-1}^\star\}$). Among these choices, at most $|I^{(t)}| \leq m^2 s$ fall into the set $I^{(t)}$. Recall our choice of $m := \left\lfloor \frac{1}{2}\sqrt{k/s} \right\rfloor$, which ensures $m^2 s \leq k/4$ and $m \leq k/2$. Thus, the probability of not stopping after drawing $i_j^\star$ is at most $\frac{m^2 s}{k-m} \leq \frac{k/4}{k/2} = \frac{1}{2}$. It follows that the number of critical indices sampled in each round $t$ is stochastically dominated by a geometric random variable with parameter $1/2$. Therefore, we conclude that $\mathbb{E}\left[N^{(r)}\right] \leq 2r$.

**Control the Success Probability**   Finally, we derive a contradiction by arguing that the probability that $\mathcal{A}$ finds an $\varepsilon$-approximate maximum of $f$ is below $9/10$. Let $\hat{w}$ denote the output of $\mathcal{A}$, and $\hat{I} := \{i \in [k] : \hat{w}_i \geq 1/(2m)\}$ be the indices on which $\hat{w}$ puts a weight of at least $1/(2m)$. Since the entries of $\hat{w} \in \Delta^{k-1}$ sum up to 1, $|\hat{I}| \leq 2m$. By Lemma 23, for $\hat{w}$ to be an $\varepsilon$-approximate maximum, $\hat{I} \cap \{i_1^\star, i_2^\star, \ldots, i_m^\star\}$ must have a size $\geq m/2$.

Conditioning on the event that $N^{(r)} \leq m/4$, at least $3m/4$ critical indices have not been chosen, and they are uniformly distributed among the $k - N^{(r)}$ remaining indices. For the condition $|\hat{I} \cap \{i_1^\star, i_2^\star, \ldots, i_m^\star\}| \geq m/2$ to hold, we must have $|\hat{I} \cap \{i_{N^{(r)}+1}^\star, i_{N^{(r)}+2}^\star, \ldots, i_m^\star\}| \geq m/2 - N^{(r)} \geq m/4$. For any choice of $\hat{I}$, over the remaining randomness in $\{i_{N^{(r)}+1}^\star, i_{N^{(r)}+2}^\star, \ldots, i_m^\star\}$, it holds that

$$\mathbb{E}\left[ |\hat{I} \cap \{i_{N^{(r)}+1}^\star, i_{N^{(r)}+2}^\star, \ldots, i_m^\star\}| \mid N^{(r)} \right] \leq |\hat{I}| \cdot \frac{m - N^{(r)}}{k - N^{(r)}} \leq 2m \cdot \frac{m}{k - m/4} \leq \frac{4m^2}{k}.$$

Recall that $m \leq \frac{1}{2}\sqrt{k/s} \leq \sqrt{k}/2$. Markov's inequality gives

$$\mathbf{Pr}\left[|\hat{I} \cap \{i^\star_{N^{(r)}+1}, i^\star_{N^{(r)}+2}, \ldots, i^\star_m\}| \geq m/4 \mid N^{(r)} \leq m/4\right]$$

$$\leq \frac{\mathbb{E}\left[|\hat{I} \cap \{i^\star_{N^{(r)}+1}, i^\star_{N^{(r)}+2}, \ldots, i^\star_m\}| \mid N^{(r)} \leq m/4\right]}{m/4}$$

$$\leq \frac{4m^2/k}{m/4} \leq \frac{8}{\sqrt{k}} \leq \frac{1}{3},$$

where the last step holds for all sufficiently large $k \geq 576$. In other words, conditioning on $N^{(r)} \leq m/4$, the probability of finding an $\varepsilon$-approximate maximum is at most $1/3$. On the other hand, by Markov's inequality and the assumption that $r \leq m/24$,

$$\mathbf{Pr}\left[N^{(r)} > m/4\right] \leq \frac{\mathbb{E}\left[N^{(r)}\right]}{m/4} \leq \frac{2r}{m/4} \leq \frac{1}{3}.$$

Therefore, the overall probability for $\mathcal{A}$ to output an $\varepsilon$-approximate maximum is at most $2/3 < 9/10$, a contradiction.

**The Ellipsoid Setting**    In this setting, we consider the execution of $\mathcal{A}$ on a random OODS instance defined in Definition 4 with $m := \left\lfloor \frac{1}{2}(k/s)^{1/4} \right\rfloor$. Different from the analysis for the box setting, we define $I^{(t)} := \{i \in [m] : \overline{w}^{(t)}_i > 1/m^4\}$ using threshold $1/m^4$ instead of $1/m^2$. Then, we have an analogous implication of Lemma 24: Unless $\{i^\star_1, i^\star_2, \ldots, i^\star_j\} \subseteq I^{(t)}$, for every point $w$ that $\mathcal{A}$ queries in round $t$, it holds that

$$f(w) = \min\left\{w_{i^\star_1} + \frac{1}{m^2}, w_{i^\star_2} + \frac{2}{m^2}, \ldots, w_{i^\star_j} + \frac{j}{m^2}\right\}.$$

To see this, note that if $i^\star_{j_0} \notin I^{(t)}$ holds for some $j_0 \in [j]$, we have $\overline{w}^{(t)}_{i^\star_{j_0}} \leq 1/m^4$. Then, for any $w \in \mathcal{O}(\overline{w}^{(t)})$, we must have

$$\frac{w^2_{i^\star_{j_0}}}{\overline{w}^{(t)}_{i^\star_{j_0}}} \leq \sum_{i=1}^{k} \frac{w^2_i}{\overline{w}^{(t)}_i} \leq 1,$$

which implies $w_{i^\star_{j_0}} \leq \sqrt{\overline{w}^{(t)}_{i^\star_{j_0}}} \leq 1/m^2$. Then, Lemma 24 implies

$$f(w) = \min\left\{w_{i^\star_1} + \frac{1}{m^2}, w_{i^\star_2} + \frac{2}{m^2}, \ldots, w_{i^\star_j} + \frac{j}{m^2}\right\}.$$

To control the expectation of $N^{(r)}$, we note that the definition of $I^{(t)}$ and the assumption on $\mathcal{A}$ having a sample overhead of $s$ together imply $|I^{(t)}| \leq m^4 s$. Then, the choice of $m \leq \frac{1}{2}(k/s)^{1/4} \leq k/2$ guarantees that the sampling process stops at each step except with probability $\frac{m^4 s}{k-m} \leq \frac{k/16}{k/2} \leq 1/2$. The rest of the proof goes through. □

### D.3  Hard Instance for Small $\varepsilon$

When the accuracy parameter $\varepsilon$ is exponentially small, we give a slightly different instance on which any OODS algorithm must take $\Omega(k)$ rounds in the box setting and $\Omega(\sqrt{k})$ rounds in the ellipsoid setting, matching the exponents in Theorems 8 and 20.

**Definition 5** (Hard instance for small $\varepsilon$). Given $k \geq m \geq 1$, draw $m$ different indices $i^\star_1, i^\star_2, \ldots, i^\star_m \in [k]$ uniformly at random. The objective function $f : \Delta^{k-1} \to [0,2]$ is

$$f(w) := \min_{j \in [m]}(a_j \cdot w_{i^\star_j} + b_j),$$

where $a_j = 2^{-j}$ and $b_j = (1 - 2^{-j})/m$.

Again, we allow the co-domain of $f$ to be $[0,2]$ instead of $[0,1]$ for brevity. We start with a few simple observations and the intuition behind the lower bound proof.

**Characterization of Approximate Maxima**  We first note that the maximizer of $f$ over $\Delta^{k-1}$ is the following vector $w$: $w_i = 1/m$ for every $i \in \{i_1^\star, \ldots, i_m^\star\}$ and $w_i = 0$ for $i \in [k] \setminus \{i_1^\star, \ldots, i_m^\star\}$. Indeed, such $w$ ensures that

$$a_j \cdot w_{i_j^\star} + b_j = 2^{-j} \cdot \frac{1}{m} + \frac{1 - 2^{-j}}{m} = \frac{1}{m}$$

for every $j \in [m]$ and thus $f(w) = 1/m$. Furthermore, to achieve an objective strictly higher than $1/m$, we must strictly increase each of $w_{i_1^\star}, \ldots, w_{i_m^\star}$, which would violate $w \in \Delta^{k-1}$.

Analogous to Lemma 23, the following lemma states that for every $\varepsilon \le e^{-\Omega(k)}$, every $\varepsilon$-approximate maximum of $f$ must put a significant weight of $\Omega(1/m)$ on at least half of the *critical indices* $i_1^\star, \ldots, i_m^\star$.

**Lemma 26.** *For every $\varepsilon \le 2^{-(k+1)/2}/(2k)$ and every $w \in \Delta^{k-1}$ that satisfies $f(w) \ge \frac{1}{m} - \varepsilon$, we have*

$$\left|\{i \in [k] : w_i \ge 1/(2m)\} \cap \{i_1^\star, i_2^\star, \ldots, i_m^\star\}\right| \ge m/2.$$

*Proof.* Suppose towards a contradiction that strictly fewer than $m/2$ entries among $w_{i_1^\star}, w_{i_2^\star}, \ldots, w_{i_m^\star}$ are at least $1/(2m)$. Then, there exists $j \in \{1, 2, \ldots, \lceil m/2 \rceil\}$ such that $w_{i_j^\star} < 1/(2m)$. It follows that

$$f(w) \le a_j \cdot w_{i_j^\star} + b_j < \frac{2^{-j}}{2m} + \frac{1 - 2^{-j}}{m} = \frac{1}{m} - \frac{2^{-j}}{2m} \le \frac{1}{m} - \frac{2^{-(m+1)/2}}{2m}.$$

On the other hand, since $m \le k$ and $\varepsilon \le 2^{-(k+1)/2}/(2k)$, we have

$$\frac{1}{m} - \varepsilon \ge \frac{1}{m} - \frac{2^{-(k+1)/2}}{2k} = \frac{1}{m} - \frac{2^{-(m+1)/2}}{2m}.$$

This contradicts the assumption that $f(w) \ge \frac{1}{m} - \varepsilon$. $\qquad\square$

**Limited Information from a First-Order Oracle**  Again, the lower bound proof amounts to showing that an OODS algorithm cannot identify $\Omega(m)$ critical indices using $\ll m$ rounds while having a low sample overhead. To this end, we note that the following is a valid first-order oracle for $f$: Given $w \in \Delta^{k-1}$, find the minimum $j \in [m]$ such that $f(w) = a_j \cdot w_{i_j^\star} + b_j$. Return the value of $f(w)$ and the supergradient $a_j \cdot e_{i_j^\star}$. Again, the optimization algorithm only gets to know the critical index that accounts for the value of $f(w)$.

The following lemma is an analogue of Lemma 24: We must put a weight of $\Omega(1/m)$ on each of $i_1^\star, i_2^\star, \ldots, i_j^\star$ to learn the values of $i_{j+1}^\star, \ldots, i_m^\star$ from the first-order oracle.

**Lemma 27.** *For every $w \in \Delta^{k-1}$ and $j \in [m]$, assuming $\min\{w_{i_1^\star}, w_{i_2^\star}, \ldots, w_{i_j^\star}\} \le 1/(2m)$, we have*

$$f(w) = \min\left\{a_1 \cdot w_{i_1^\star} + b_1, a_2 \cdot w_{i_2^\star} + b_2, \ldots, a_j \cdot w_{i_j^\star} + b_j\right\}.$$

*Proof.* Suppose towards a contradiction that some $w \in \Delta^{k-1}$ satisfies $w_{i_{j_1}^\star} \le 1/(2m)$ for some $j_1 \le j$, but $f(w) \ne \min\left\{a_1 \cdot w_{i_1^\star} + b_1, a_2 \cdot w_{i_2^\star} + b_2, \ldots, a_j \cdot w_{i_j^\star} + b_j\right\}$. Then, there exists $j_2 > j$ such that

$$a_{j_2} \cdot w_{i_{j_2}^\star} + b_{j_2} < \min\left\{a_1 \cdot w_{i_1^\star} + b_1, a_2 \cdot w_{i_2^\star} + b_2, \ldots, a_j \cdot w_{i_j^\star} + b_j\right\} \le a_{j_1} \cdot w_{i_{j_1}^\star} + b_{j_1}.$$

Plugging $a_j = 2^{-j}$ and $b_j = (1 - 2^{-j})/m$ into the above gives

$$2^{-j_2} \cdot w_{i_{j_2}^\star} + \frac{1 - 2^{-j_2}}{m} < 2^{-j_1} \cdot w_{i_{j_1}^\star} + \frac{1 - 2^{-j_1}}{m}.$$

Recalling the assumption that $w_{i_{j_1}^\star} \le 1/(2m)$, we have

$$w_{i_{j_2}^\star} < \frac{1}{m} + 2^{j_2 - j_1} \cdot \left(w_{i_{j_1}^\star} - \frac{1}{m}\right) \le \frac{1}{m} - \frac{1}{2m} \cdot 2^{j_2 - j_1} \le 0,$$

a contradiction. $\qquad\square$

**Intuition behind Lower Bound** As in the large-$\varepsilon$ regime, there is a natural $m$-round algorithm for solving the instance from Definition 5, so the proof amounts to arguing that the OODS algorithm cannot be "more clever" and learn more than one critical index in each round. For example, the algorithm might put a cap of $\gg 1/m$ on several coordinates in the first round, in the hope of hitting more than one indices in $i_1^\star, i_2^\star, \ldots$. However, any algorithm with a sample overhead of $s$ can make only $O(ms)$ such guesses. Then, assuming $ms \ll k$, the guesses only cover a tiny fraction of the indices. Thus, over the uniform randomness in $i^\star$, the algorithm can still only learn $O(1)$ critical indices in expectation within each round.

### D.4 Lower Bounds for Small $\varepsilon$

Finally, we state and prove the lower bounds in the $\varepsilon \leq e^{-\Omega(k)}$ regime, for both the box and the ellipsoid settings.

**Theorem 28.** *The following holds for all sufficiently large $k$, $\varepsilon \leq 2^{-(k+1)/2}/(2k)$ and $s \geq 1$: (1) In the box setting, for $m := \min\{\lfloor k/(8s) \rfloor, \lfloor k/48 \rfloor\}$, no OODS algorithm with $r \leq m/24$ rounds and sample overhead $s$ can find an $\varepsilon$-approximate maximum with probability at least $9/10$; (2) In the ellipsoid setting, for $m := \min\{\lfloor \frac{1}{4}\sqrt{k/s} \rfloor, \lfloor k/48 \rfloor\}$, no OODS algorithm with $r \leq m/24$ rounds and sample overhead $s$ can find an $\varepsilon$-approximate maximum with probability at least $9/10$.*

In particular, to have a sample overhead $s \leq \text{polylog}(k)$, any OODS algorithm must take $\widetilde{\Omega}(k)$ rounds in the box setting, and $\widetilde{\Omega}(\sqrt{k})$ rounds in the ellipsoid setting. These match the exponents in the upper bounds (Theorems 8 and 20).

The proof is analogous to the one for Theorem 25, so we will be brief.

*Proof.* Again, we focus on the box setting; the ellipsoid setting follows easily with only a few changes in the proof.

Suppose towards a contradiction that an OODS algorithm $\mathcal{A}$ for the box setting takes $r \leq m/24$ rounds and succeeds with probability at least $9/10$. Consider the execution of $\mathcal{A}$ on a random OODS instance defined in Definition 4 with $m := \min\{\lfloor k/(8s) \rfloor, \lfloor k/48 \rfloor\}$. For each round $t \in [r]$, let $I^{(t)} := \{i \in [m] : \overline{w}_i^{(t)} > 1/(2m)\}$ denote the indices on which algorithm $\mathcal{A}$ sets a cap of $\Omega(1/m)$. Since $\mathcal{A}$ has a sample overhead of $s$, we have $\|\overline{w}^{(t)}\|_1 \leq s$ and thus $|I^{(t)}| \leq 2ms$. By definition of the box setting, within each round $t$, $\mathcal{A}$ cannot query the oracle on $w$ if $w_i > 1/(2m)$ holds for some $i \in [k] \setminus I^{(t)}$. Therefore, Lemma 27 implies the following: Unless $\{i_1^\star, i_2^\star, \ldots, i_j^\star\} \subseteq I^{(t)}$, for every point $w$ that $\mathcal{A}$ queries in round $t$, it holds that

$$f(w) = \min\left\{ a_1 \cdot w_{i_1^\star} + b_1, a_2 \cdot w_{i_2^\star} + b_2, \ldots, a_j \cdot w_{i_j^\star} + b_j \right\}.$$

**Deferring Randomness** Again, we draw the $m$ critical indices "on demand" in our analysis. We start with $n^{(0)} = 0$. In each round $t \in [r]$, after $\mathcal{A}$ decides on $\overline{w}^{(t)}$ and $I^{(t)}$, we keep sampling $i_{n^{(t-1)}+1}^\star, i_{n^{(t-1)}+2}^\star, \ldots$, until either all $m$ indices are chosen or we encounter an index outside $I^{(t)}$. Let $n^{(t)}$ denote the total number of critical indices that have been sampled by the end of round $t$. By the implication of Lemma 27, all queries made by $\mathcal{A}$ during the $t$-th round can be answered solely based on the values of $i_1^\star$ through $i_{n^{(t)}}^\star$, as they do not depend on the $m - n^{(t)}$ indices that have not been decided.

**Control the Progress Measure** Let random variable $N^{(t)}$ denote the value of $n^{(t)}$ over the randomness in both algorithm $\mathcal{A}$ and the critical indices. In each round $t \in [r]$, whenever we sample a critical index $i_j^\star$ ($j \in [m]$), there are $k - (j-1) > k - m$ possible choices (namely, $[k] \setminus \{i_1^\star, \ldots, i_{j-1}^\star\}$). Among these choices, at most $|I^{(t)}| \leq 2ms$ fall into the set $I^{(t)}$. Recall our choice of $m \leq k/(8s)$, which ensures both $2ms \leq k/4$ and $m \leq k/2$. Thus, the probability of not stopping after drawing $i_j^\star$ is at most $\frac{2ms}{k-m} \leq \frac{k/4}{k/2} = \frac{1}{2}$. It follows that at most 2 critical indices are sampled in each round $t$ in expectation, so we have $\mathbb{E}\left[N^{(r)}\right] \leq 2r$.

**Control the Success Probability** Finally, we derive a contradiction by arguing that the probability that $\mathcal{A}$ finds an $\varepsilon$-approximate maximum of $f$ is below $9/10$. Let $\hat{w}$ denote the output of $\mathcal{A}$, and

$\hat{I} \coloneqq \{i \in [k] : \hat{w}_i \geq 1/(2m)\}$ be the indices on which $\hat{w}$ puts a weight of at least $1/(2m)$. Since the entries of $\hat{w} \in \Delta^{k-1}$ sum up to 1, $|\hat{I}| \leq 2m$. By Lemma 26, for $\hat{w}$ to be an $\varepsilon$-approximate maximum, $\hat{I} \cap \{i_1^\star, i_2^\star, \dots, i_m^\star\}$ must have a size $\geq m/2$.

Conditioning on the event that $N^{(r)} \leq m/4$, at least $3m/4$ critical indices have not been chosen, and they are uniformly distributed among the $k - N^{(r)}$ remaining indices. For the condition $|\hat{I} \cap \{i_1^\star, i_2^\star, \dots, i_m^\star\}| \geq m/2$ to hold, we must have $|\hat{I} \cap \{i_{N^{(r)}+1}^\star, i_{N^{(r)}+2}^\star, \dots, i_m^\star\}| \geq m/2 - N^{(r)} \geq m/4$. For any choice of $\hat{I}$, over the remaining randomness in $\{i_{N^{(r)}+1}^\star, i_{N^{(r)}+2}^\star, \dots, i_m^\star\}$, it holds that

$$\mathbb{E}\left[|\hat{I} \cap \{i_{N^{(r)}+1}^\star, i_{N^{(r)}+2}^\star, \dots, i_m^\star\}| \mid N^{(r)}\right] \leq |\hat{I}| \cdot \frac{m - N^{(r)}}{k - N^{(r)}} \leq 2m \cdot \frac{m}{k - m/4} \leq \frac{4m^2}{k}.$$

Recall that $m \leq \frac{1}{2}\sqrt{k/s} \leq \sqrt{k}/2$. Markov's inequality gives

$$\mathbf{Pr}\left[|\hat{I} \cap \{i_{N^{(r)}+1}^\star, i_{N^{(r)}+2}^\star, \dots, i_m^\star\}| \geq m/4 \mid N^{(r)} \leq m/4\right]$$

$$\leq \frac{\mathbb{E}\left[|\hat{I} \cap \{i_{N^{(r)}+1}^\star, i_{N^{(r)}+2}^\star, \dots, i_m^\star\}| \mid N^{(r)} \leq m/4\right]}{m/4}$$

$$\leq \frac{4m^2/k}{m/4} = \frac{16m}{k} \leq \frac{1}{3},$$

where the last step follows from our choice of $m \leq k/48$. In other words, conditioning on $N^{(r)} \leq m/4$, the probability of finding an $\varepsilon$-approximate maximum is at most $1/3$. On the other hand, by Markov's inequality and the assumption that $r \leq m/24$,

$$\mathbf{Pr}\left[N^{(r)} > m/4\right] \leq \frac{\mathbb{E}\left[N^{(r)}\right]}{m/4} \leq \frac{2r}{m/4} \leq \frac{1}{3}.$$

Therefore, the overall probability for $\mathcal{A}$ to output an $\varepsilon$-approximate maximum is at most $2/3 < 9/10$, a contradiction.

**The Ellipsoid Setting** Finally, for the ellipsoid setting, we instead consider the execution of $\mathcal{A}$ on a random OODS instance defined in Definition 4 with $m \coloneqq \min\{\lfloor\frac{1}{4}\sqrt{k/s}\rfloor, \lfloor k/48\rfloor\}$, and define $I^{(t)} \coloneqq \{i \in [m] : \overline{w}_i^{(t)} > 1/(4m^2)\}$ using threshold $1/(4m^2)$ instead of $1/(2m)$. Then, we have an analogous implication of Lemma 24: Unless $\{i_1^\star, i_2^\star, \dots, i_j^\star\} \subseteq I^{(t)}$, for every point $w$ that $\mathcal{A}$ queries in round $t$, it holds that

$$f(w) = \min\left\{w_{i_1^\star} + \frac{1}{m^2}, w_{i_2^\star} + \frac{2}{m^2}, \dots, w_{i_j^\star} + \frac{j}{m^2}\right\}.$$

To see this, note that if $i_{j_0}^\star \notin I^{(t)}$ holds for some $j_0 \in [j]$, we have $\overline{w}_{i_{j_0}^\star}^{(t)} \leq 1/(4m^2)$. Then, for any $w \in \mathcal{O}(\overline{w}^{(t)})$, we must have

$$\frac{w_{i_{j_0}^\star}^2}{\overline{w}_{i_{j_0}^\star}^{(t)}} \leq \sum_{i=1}^{k} \frac{w_i^2}{\overline{w}_i} \leq 1,$$

which implies $w_{i_{j_0}^\star} \leq \sqrt{\overline{w}_{i_{j_0}^\star}^{(t)}} \leq 1/(2m)$. Then, Lemma 27 implies

$$f(w) = \min\left\{w_{i_1^\star} + \frac{1}{m^2}, w_{i_2^\star} + \frac{2}{m^2}, \dots, w_{i_j^\star} + \frac{j}{m^2}\right\}.$$

To control the expectation of $N^{(r)}$, we note that the definition of $I^{(t)}$ and the assumption on $\mathcal{A}$ having a sample overhead of $s$ together imply $|I^{(t)}| \leq 4m^2 s$. Then, the choice of $m \leq \frac{1}{4}\sqrt{k/s}$ guarantees that the sampling process stops at each step except with probability $\frac{4m^2 s}{k-m} \leq \frac{k/4}{k/2} = 1/2$. The rest of the proof goes through. $\qquad\square$

