# OpenReview forum: "Sample-Adaptivity Tradeoff in On-Demand Sampling"
_NeurIPS.cc/2025/Conference — NeurIPS 2025 spotlight_

### Official Review · Reviewer_7PVT · 2025-06-18

**Clarity:** 2
**Significance:** 3
**Originality:** 3
**Rating:** 5
**Confidence:** 2

**Summary:**

The paper studies the problem of adaptive data collection for multi-distribution binary classification. Multi-distribution refers to the learner being evaluated on the error for data coming from across $k$ distributions. The adaptive data collection is formalised by a procedure over $r$ rounds where in each round the learner can choose the number of samples to collect from each of the distributions (and this can be based on samples collected in previous rounds). The aim of the paper is to understand the trade-off between $r$ and the total number of samples needed for $\epsilon$ accuracy (on the multi-distribution error in a PAC sense). In the realizable case (where a perfect predictor exists), they provide upper and lower bounds that characterise the (near) optimal trade-off. In particular, O(log k) adaptive rounds is optimal to achieve the optimal sample complexity. In the more challenging agnostic case (where a perfect predictor does not exist), an algorithm is provided that achieves the near-optimal sample-complexity in a number of rounds that is independent of $\epsilon$. The optimization via on-demand sampling framework is introduced whose connection to the multi-distribution problem allows to motivate the difficulties in improving the round-complexity of the agnostic case, as well as establish general results on the trade-off between sample complexity and adaptivity.

**Questions:**

1. In the discussion following Theorem 1, I am not sure about the claim at l.130 that $r = \log k$ is the “most sample-efficient”. Given the upper-bound in Theorem 1, the most sample efficient is when $r \rightarrow \infty$ - I guess what was meant is that $r = \log k$ already allows to recover the order of the optimal sample complexity. But really if we ignore what we know to be the optimal sample-complexity, the bound in Theorem 1 suggests that we should set $r$ to match the second term of the bound. And at a higher level, what it says is that the adaptivity level (the value of $r$) should be set as a function of k and d and in particular will behave somewhat differently when k>d or d>k. Have you thought about this ? And also how this might be relevant to settings where $k$ may be unknown, how hard would it be to extend these results ?
2. In l.132, when $r = 5$ is chosen in Theorem 1, why is it $\sqrt{k}$ in the sample complexity and not $k^{2/5}$, is this a typo ? I think there is also a typo l.157: it should be $\sqrt{k}d$ not $\sqrt{d}k$, right?
3. In weakness 4., it was mentioned that no lower-bound is provided for the round-complexity in the agnostic case. In section 5.3, it was discussed the improving the round-complexity required a different approach. Could you comment on whether you believe the $\sqrt{k}$ round-complexity could be improved and if so what the optimal round-complexity could be?

**Ethical Concerns:**

["NO or VERY MINOR ethics concerns only"]

**Final Justification:**

The paper is well-written and presents a substantial set of results, significantly improving upon prior work. There are a few weaknesses to the work but they are sufficiently minor to not be an obstacle to acceptance.

**Limitations:**

yes

**Quality:**

3

**Strengths And Weaknesses:**

Strengths:
1. The paper is generally well-written and easy to follow (except a few parts – see weaknesses). The algorithms are well explained and the proof sketches provided in the main text are nice and provide some useful intuitions.
2. Realizable case: An upper-bound is established that captures a trade-off between adaptivity and sample-complexity. In particular, I believe the main point that can be concluded from the upper and lower bound is about the optimality (sufficiency + necessity) of $\log k$ rounds to achieve the optimal sample-complexity, completing the characterization of optimality for adaptive sample-complexity.
3. Agnostic case: An upper-bound is established that achieves the near-optimal sample-complexity with a round-complexity that is independent of the accuracy $\epsilon$, improving on prior work whose algorithms had round-complexity that depended polynomially on $\epsilon$.
4. The algorithms are non-trivial extensions of algorithms from prior work.

Weaknesses:
1. Some of the technical description is a bit unclear. There seems to be some missing notation in the description of Algorithm 1, e.g. $q_t^m$ is not defined (I guess this means m samples from $q_t$ ?), $q_{t+1}$ / $q_t$ is not defined (only $q_{t+1}(j)$ is). Additionally, it is not clear where $p$ and $\varepsilon_A$ are used in the algorithm, which then makes the following discussion of the technical overview a bit unclear. In particular, my understanding is that the learner $A$ is called to to learn a predictor $q_t$ that minimizes the fraction of distributions (as weighted by $q_t$) where the weights $q_t(j)$ have been adjusted in previous rounds according to the errors being more or less than $\tau/2$ but I don’t see how the equation below l.142 fits in.
The proof sketch of Theorem 2 for $r=2$ provides some nice intuition but the explanation of the way $diff_i$ as uniformly random permutations is not clear. It is not clear to me what is meant by a uniformly random permutation of a copy of $\Theta(d)$. Is $diff_1$ a sample from a permutation of $[1,…,d]$ ? Is $diff_1$ itself a permutation, though my understanding is that $diff_i$ is the dimension of subspaces that more or less partition the space, I.e.\ an integer. Also, if possible I think it would be clearer to avoid using $\Theta$ notation here and explicitly put the values.
2. The lower-bound in Theorem 2 has a $k^{1/r}$ dependence while the upper bound is $k^{2/r}$. However, this is enough to establish the necessity of the $\log k$ rounds, which in my understanding is the main contribution this result allows to establish for the realizable case (and perhaps should be stressed more ?), especially since known algorithms with optimal sample-complexity already had a round complexity of $\log k$. Indeed, for the case where $r$ is a small constant, the upper-bound from Theorem 1 is an interesting result, but does not match the lower-bound.
3. The high-level connection between optimization via on-demand sampling and MDL is discussed in lines 260-265 (and again in Section 5.3), however it is unclear how the results from each section are connected. What is the connection between Theorem 4 and proposition 1 ? And with this absence of connection, it is a bit unclear how the results from Section 5 fit in with the rest of the paper. In particular, I understand how Theorem 9 provides some insight into how well lazy-hedge can perform for MDL but it is unclear how Theorem 4 does this from an upper-bound perspective. More generally, while Section 3 has a clear structure that is easy to follow, it becomes a bit harder to follow the narrative through Sections 4 and 5. Also, in Section 4 a bit more discussion could be provided as currently the results are simply stated without much discussion.
4. Proposition 1 provides an algorithm with near-optimal sample-complexity with a poly(k) round-complexity but no lower-bound is provided.

---

> ### Author Rebuttal · Authors · 2025-07-29
>
> Thank you for your detailed review! We start by clarifying the notations that you mentioned in the "weaknesses" part:
>
> - We will define the notations $q_t$ and $q_t^m$ in Algorithm 1. Thank you for your comment!
>
> - Parameters $p$ and $\epsilon_{\mathbb{A}}$ are used for setting the learning rate $\alpha$ (Lines 1 and 8) and the sample size $m$ for each round (Lines 2 and 5).
>
> - The equation below Line 142 is a consequence of Markov's inequality when $h_t$ has an $O(\tau p)$ error on the mixture distribution $q_t$.
>
> - The notation $\mathrm{diff}$ stands for a length-$k$ vector $(\mathrm{diff}_1, \mathrm{diff}_2, \ldots, \mathrm{diff}_k)$, which is in turn a uniformly random permutation of the following $k$ numbers: one copy of $c_1 \cdot d$, $\sqrt{k}$ copies of $c_2 \cdot d/\sqrt{k}$, and $(k - \sqrt{k} - 1)$ copies of $c_3 \cdot d / k$. (The constant factors $c_1, c_2, c_3 > 0$ are chosen such that these $k$ numbers sum up to at most $d$, and they are suppressed by the $\Theta(\cdot)$ notation on Lines 161--162.)
>
> We answer your questions in the following:
>
> 1. Thanks for catching this! Yes, we meant that setting $r = \log k$ is sufficient for the $k^{1/r}$ factor to be $O(1)$, so larger values of $r$ will not help (modulo constant factors).
>
> Regarding setting $r$ so that the two terms match, that is a very nice observation and thanks for suggesting it! For example, when $d = O(1)$, setting $r = 3$ would be sufficient for the first term to be dominated by the second. What we had in mind was the $d \gg k$ regime (e.g., the setup of Theorem 16 (formal version of Theorem 2)), in which case the first term would dominate.
>
> We were unsure about the setting with an unknown $k$: how would the learner sample from the distributions if the number is unknown?
>
> 2. Yes, it is a typo (or at least, a loose bound) on Line 132: $r = 4$ rounds should be sufficient. And yes, it should have been $d\sqrt{k}$ on Line 157---thank you catching that!
>
> 3. We believe that the right round complexity for the agnostic setting should be $O(\log k)$, mirroring the realizable setting. That is, our lower bounds for the OODS framework suggest a barrier in achieving this round complexity if the learning algorithm treats MDL as an optimization problem in a black-box way. A concrete starting point towards breaking this barrier would be the alternative MDL algorithms (e.g., the one of [Peng, COLT'24]) that use the samples in a more sophisticated way (beyond running ERM or evaluating empirical errors) and sidestep the OODS barrier.

---

> ### Comment · Reviewer_7PVT · 2025-08-02
>
> Thank you for your detailed response. Most of my points have been cleared up. I still don't fully understand the eq. below line 142, is it saying the weights distribution errors should be less than p rather than just minimized ? I would encourage the authors to add some more detail explaining what it means / represents in the final version. I also would encourage the authors to think about clarifying the part mentioned in the third weakness. For the unknown $k$, I guess you are right it does not make sense. Otherwise, I am happy to keep my score.

---

> > ### Author Response · Authors · 2025-08-03
> >
> > Thank you for your suggestions! For the equation below Line 142, the following is a more detailed explanation, which we will incorporate into the final version:
> >
> > Algorithm 1 calls the learner $\mathbb{A}$ to minimize the error of predictor $h_t$ on a mixture of distribution $D_1, D_2, \ldots, D_k$, which can be viewed as a weighted average of the errors of $h_t$ on the $k$ distributions. This would then imply that, on most of the $k$ distributions, the error of $h_t$ is low.
> >
> > In more detail, the learner $\mathbb{A}$ learns predictor $h_t$ for the mixture $q_t = \sum_{j=1}^{k}q_t(j) \cdot D_j$. By our choice of the sample size $m$ (on Line 2), $h_t$ has an error $\le \epsilon_{\mathbb{A}} < \tau p$ on $q_t$ (except with probability $\delta_{\mathbb{A}}$). Equivalently, if we sample a random index $j \in [k]$ according to $(q_t(1), q_t(2), \ldots, q_t(k))$, the expected error of $h_t$ on $D_j$---$\mathbb{E}_j[\mathrm{err}(h_t, D_j)]$---is at most $\tau p$. By Markov's inequality, we have $\Pr_j[\mathrm{err}(h_t, D_j) > \tau] \le \frac{\tau p}{\tau} = p$, which is equivalent to the inequality below Line 142.

---

### Official Review · Reviewer_npZF · 2025-06-30

**Clarity:** 3
**Significance:** 3
**Originality:** 3
**Rating:** 4
**Confidence:** 4

**Summary:**

This paper works on reducing the round complexity for multi-distributional learning problem. Multi-distribution learning (MDL) focuses on learning multiple distributions with on-demand sampling. Recent work established near-optimal sample complexity for MDL. In this work, the authors try to reduce the round complexity with lazy updates. They provide a trade-off between the round complexity and the sample complexity for the realizable case, and prove a sub-linear round complexity for the agnostic case.

**Questions:**

1. Can you provide some intuition for why the lower bound is not polynomial in $\epsilon$?  Is there any method to combine your arguments with the classical $\tilde{\Omega}( \frac{d+k}{\epsilon^2})$ lower bound (without limitation on round complexity) to obtain a better rate?
2. I wonder whether there is substantial difficulty in extending the upper bounds for realizable setting to the agnostic setting. In particular, fix $r\leq \sqrt{k}$, could you prove a sample complexity upper bound better than the trivial bound $dk/\epsilon^2$ for the agnostic setting?

**Ethical Concerns:**

["NO or VERY MINOR ethics concerns only"]

**Final Justification:**

My major concerns are resolved, so I would like to keep my evaluation as 4.

**Limitations:**

The theoretical results only apply to the specific parameter regimes.

**Paper Formatting Concerns:**

I did not find any significant formatting issues.

**Quality:**

3

**Strengths And Weaknesses:**

Strengths:
1. The major contribution of this work is to establish the trade-off between the adaptivity and the sample complexity. This represents a valuable contribution to related fields.
2. In technique, the major contribution is  a novel application of a variant of the AdaBoost algorithm with a particular notion of margin.  The construct of the lower bound is also a significant technical contribution.

Weakness:
1. The results in the agnostic setting are relatively weak due to the absence of statistical lower bounds.

---

> ### Author Rebuttal · Authors · 2025-07-29
>
> Thank you for your review! We answer your questions in the following:
>
> 1. While the lower bound in Theorem 2 is stated for constant $\epsilon$ (and thus does not have an $\epsilon$ dependence), it is easy to derive an $\Omega(\frac{1}{\epsilon} \cdot \frac{dk^{1/r}}{r\log^2 k})$ lower bound for general $\epsilon$ via the standard argument below. (The dependence would be $1/\epsilon$ instead of $1/\epsilon^2$ since the hard instance that we use is realizable.) We focused on the constant-$\epsilon$ regime for brevity. We will add the following discussion to our paper and point to it after the statement of Theorem 2.
>
> We start with the construction for constant accuracy parameter $\epsilon_0 = 0.01$. Then, we obtain a new MDL instance by "diluting" each data distribution by a factor of $100\epsilon$: we scale the probability mass on each example by a factor of $100\epsilon$, and put the remaining mass of $1 - 100\epsilon$ on a "trivial" example $(\bot, 0)$, where $\bot$ is a dummy instance that is labeled with $0$ by every hypothesis. Learn this new instance up to error $\epsilon$ is equivalent to learning the original instance (before diluting) up to error $\epsilon / (100\epsilon) = 0.01 = \epsilon_0$.  Intuitively, the example $(\bot, 0)$ provides no information, and the learning algorithm has to draw $\Omega(1 / \epsilon)$ samples in expectation to see an informative example. This leads to a lower bound with a $1/\epsilon$ factor.
>
> 2. Yes, Theorem 20 and the discussion in Appendix C.4 imply an $\widetilde O(\sqrt{k/C})$-round algorithm for agnostic MDL with sample complexity $\widetilde O(C \cdot (d + k) / \epsilon^2)$ for any $C \in [4, k]$. For example, setting $C = k^{0.8}$ gives an $\widetilde O(k^{0.1})$-round algorithm with sample complexity $\widetilde O(k^{0.8} \cdot (d + k) / \epsilon^2)$, which is generally better than the trivial bound of $dk/\epsilon^2$.

---

> > ### Comment · Reviewer_npZF · 2025-08-02
> > **Discussion**
> >
> > Thank you for the detailed response. I would like to keep my score at the moment.

---

### Official Review · Reviewer_EyU1 · 2025-07-02

**Clarity:** 4
**Significance:** 3
**Originality:** 3
**Rating:** 5
**Confidence:** 4

**Summary:**

This paper studies an adaptive learning process in the Multi-Distribution Learning (MDL) setting, and characterizes a tradeoff between the sample complexity and round complexity. Specifically, it studies to what extent an adaptive sample collection can achieve a near-optimal sample complexity while maintaining a beneficial round complexity. Under the realizable setting, it characterizes the sample complexity as $dk^{\Theta(1/r)}/\epsilon$, where $r$ is the round complexity, using a variant of the classical AdaBoost algorithm. In the agnostic case, it shows that with $\tilde{O}(\sqrt{k})$ rounds, a sample complexity of $\tilde{O}((d+k)/\epsilon^2)$ can be achieved through an revised algorithm from that of [ZZC+24] (by allowing laze sampling with a cap). From a theoretical perspective, the paper also introduces the framework of Optimization via On-Demand Sampling that naturally connects to the multi-distributional learning problems, which implies both the upper and lower bounds for the round complexity with specified sample overhead.

**Questions:**

Did authors consider other boosting algorithms that might further reduce the sample complexity?

**Ethical Concerns:**

["NO or VERY MINOR ethics concerns only"]

**Final Justification:**

The paper is of high quality. During the discussion period, the authors provided further clarification on its related work. I have checked other reviews, and all reviewers showed support for acceptance. Hence, I believe the paper is definitely above the acceptance bar.

**Limitations:**

Yes.

**Paper Formatting Concerns:**

No concerns.

**Quality:**

4

**Strengths And Weaknesses:**

The paper considers an important problem of multi-distributional learning under the adaptive setting, and provides a theoretical foundation for studying the tradeoff between the sample and round complexity. It studies the problem in both realizable and agnostic settings, providing both upper (through well-designed algorithms) and lower bounds (through rigorous analysis). The paper is very well presented with original and insightful discussions. The work is very influential in adaptive learning settings, and could benefit in this area of research, including on-demanding sampling, multi-arm bandit problems, continual learning, to name a few.

---

> ### Author Rebuttal · Authors · 2025-07-29
>
> Thank you for your review! For the realizable setting, in Appendix A (Lines 584--607), we discuss using other boosting algorithms based on recursive boosting [Schapire, 1990] and boost-by-majority [Schapire and Freund, 2012]. They yield similar quantitative bounds in terms of tradeoff between sample complexity and round complexity, but can be favorable in the regime of constant rounds. For example, as mentioned in Lines 597--598,  only 3 rounds of adaptive sampling suffice to achieve a sample complexity of $\widetilde{O}(d\sqrt{k}/\epsilon)$.

---

### Official Review · Reviewer_ajYw · 2025-07-10

**Clarity:** 3
**Significance:** 4
**Originality:** 4
**Rating:** 4
**Confidence:** 3

**Summary:**

This work studies on-demand sampling, where there are $K$ distributions and the goal of Learner is to learn a hypothesis that minimizes the worst-case error. They consider two classical settings in statistical learning: the realizable setting and the agnostic setting. They present both the upper and lower bounds results. For this work, they study the fundamental trade-off between sample complexity and round complexity, which is very important.

**Questions:**

1. What is adaptive sampling, or how to decide in each round, whether it is adaptive sampling or non-adaptive sampling? And what is full adaptivity and full non-adaptivity? I hope to see more background knowledge of adaptive sampling.

2. Is my understanding right the provided results in the realizable setting can only improve the existing non-adaptive sampling results? From the content below Theorem 1, it seems it cannot improve the results shown in [BHPQ17, CZZ18, NZ18].

3. Can you give more explanations and intuitions about the agnostic results such as how to get rid of the $\epsilon$ in your new results? And why the round complexity does not need to be dependent on $\epsilon$? For me, the existing results make more sense.

4. For Algorithm 1, how to decide the number of samples that are distributed according to each of the K distribution in each round? Can it be varied among distributions?

5. The discussion below Theorem 1, it will make it look nicer if, the two special cases, the case where $r = \log k$ and  $r = O(1)$, are presented in Table 1.

6. Also, for the discussion below Theorem 1, it seems that when $r = \log K$, it cannot recover the existing results.  Why claim it is almost recovered? I think it only improves the non-adaptive sampling results.

7. Algorithm 1 inputs round $r$, can it be chosen adaptively, for example, based on the past collected data?

8. Maybe I missed something in  Algorithm~1. what are $q_t^m$ and $D_j^n$  ?

9. In algorithm 1, in step 3, it seems we do not know $D_i$ right? So, how to compute $q_1$?

10. For Theorem 2, it is a minimax lower bound or a lower bound for this specific algorithm? If it is a minimax lower bound, it seems to be contradicted with the existing results in [BHPQ17, CZZ18, NZ18] in the realizable setting.

**Ethical Concerns:**

["NO or VERY MINOR ethics concerns only"]

**Final Justification:**

There are some space for the improvement in terms of presentation. So, I keep my current score.

**Quality:**

3

**Strengths And Weaknesses:**

Strength: the studied trade-off is very fundamental.

Weakness: for some part about the presented Algorithm 1, it is not clear. The details are in the box below.

---

> ### Author Rebuttal · Authors · 2025-07-29
>
> Thank you for your review! We answer your questions in the following:
>
> 1. The term *adaptive sampling* refers to that the sampling is done in multiple rounds (see Lines 104--107). The learner does not get to "decide" whether each round is adaptive or not---every round counts toward the round complexity.
>
> Put in a different way: If the learner wants the second round of sampling to be non-adaptive (i.e., independent of the samples drawn in the first round), they could have merged the first two rounds into a single round.
>
> "Full adaptivity" means that there is no limit on the number of rounds. "Full non-adaptivity" means that there is only one round.
>
> 2. The algorithms of [BHPQ17, CZZ18, NZ18] all require $r = \Theta(\log k)$ rounds. In comparison, our results for the realizable setting (Theorems 1 and 2) apply to a general value of $r$ (between $1$ and $\Theta(\log k)$) and not just the fully non-adaptive case. As mentioned on Lines 47--50, little was known in the regime that $2 \le r \ll \log k$.
>
> 3. To see why the $\epsilon$-dependence can be removed, it would be the most informative to check Proposition 2 and its proof sketch, which we outline in the following. The round complexity of the algorithm in [ZZC+24] has an $\epsilon$-dependence because the Hedge algorithm needs $(\log k) / \epsilon^2$ iterations, and each iteration requires one round of sampling. In comparison, the box version of our $\mathsf{LazyHedge}$ algorithm only samples when the weight assigned to one of the $k$ distributions substantially increases (say, by a factor of $2$). Since the weight of each distribution starts at $1/k$ and is at most $1$, each weight can double at most $\log k$ times. There are $k$ distributions in total, so the round complexity of the algorithm is at most $k \log k$, which is independent of $\epsilon$.
>
> 4. In each round $t$, the numbers of samples to be drawn from the $k$ distributions are computed from the mixing weights $q_t(1), q_t(2), \ldots, q_t(k)$. On Line 5 of Algorithm 1, $m$ samples are drawn from the mixture distribution $q_t = \sum_{i=1}^{k}q_t(i) \cdot D_i$. Drawing a single sample from $q_t$ is equivalent to the following two steps: (1) Sample $j \in [k]$ according to the distribution specified by $(q_t(1), q_t(2), \ldots, q_t(k))$, i.e., $j$ is set to each $j' \in [k]$ with probability $q_t(j')$; (2) Draw a sample from $D_j$. By repeating this $m$ times, Algorithm 1 obtains the number of samples that need to be drawn from each $D_i$.
>
> 5. Thank you for this comment!
>
> 6. Setting $r = \log k$ in Theorem 1 gives a sample complexity of $O(\frac{d\log k}{\epsilon} + \frac{k\log(k)\log(k/\delta)}{\epsilon}) = \widetilde O((d + k) / \epsilon)$, recovering the existing results in the first row of Table 1. (Note that the $k^{2/r}$ factor reduces to $O(1)$ when $r = \log k$.)
>
> 7. As stated, $r$ is a parameter of the algorithm and is specified before drawing any samples. It is an interesting direction for future work to design learning algorithms that set $r$ adaptively, and thereby using as few rounds as possible on each MDL instance.
>
> 8. $q_t$ is the mixture distribution specified by the mixing weights $(q_t(1), q_t(2), \ldots, q_t(k))$, i.e., $q_t = \sum_{i=1}^{k}q_t(i) \cdot D_i$. For a distribution $D$ and positive integer $m$, $D^m$ denotes the product distribution formed by $m$ copies of $D$. That is, $S \sim D^m$ is a shorthand for drawing $m$ independent samples from $m$. We will clarify these notations---thank you for catching these!
>
> 9. Note that $q_1$ is only used on Line 5, where Algorithm 1 draws $m$ independent samples from $q_1$. As mentioned in the answer to Question 4, the algorithm does not need to know $D_i$ to sample from $q_1$---to sample from $q_1 = \frac{1}{k}\sum_{i=1}^{k}D_i$, it suffices to draw an index $j$ from $\{1, 2, \ldots, k\}$ uniformly at random and sample from $D_j$.
>
> 10. Theorem 2 is a minimax lower bound that holds for all learning algorithms. It does not contradict the results of [BHPQ17, CZZ18, NZ18], since the algorithms in these work all require at least $\log k$ rounds. At $r = \log k$, the lower bound in Theorem 2 reduces to $\Omega(d / \log^3 k)$, which is consistent with the prior results (namely, the $\widetilde O(d + k)$ sample complexity for constant $\epsilon$).

---

### Decision · Program_Chairs · 2025-09-17

**Decision:**

Accept (spotlight)

**Comment:**

The reviewers agree that the paper studies a fundamental tradeoff. Moreover, the paper is well-written and presents a substantial set of results. The Algorithms are well explained as well as the proof sketches.
And the results altogether for the realizable and agnostic case make an interesting and deep contribution, which is very relevant to the field.
I will therefore accept.